# The LRRK2 G2019S mutation alters astrocyte-to-neuron communication via extracellular vesicles and induces neuron atrophy in a human iPSC-derived model of Parkinson's disease

Aurelie de Rus Jacquet*, Jenna L Tancredi, Andrew L Lemire, Michael C DeSantis, Wei-Ping Li, Erin K O'Shea

Janelia Research Campus, Howard Hughes Medical Institute, Ashburn, United States

**Abstract** Astrocytes are essential cells of the central nervous system, characterized by dynamic relationships with neurons that range from functional metabolic interactions and regulation of neuronal firing activities, to the release of neurotrophic and neuroprotective factors. In Parkinson's disease (PD), dopaminergic neurons are progressively lost during the course of the disease, but the effects of PD on astrocytes and astrocyte-to-neuron communication remain largely unknown. This study focuses on the effects of the PD-related mutation LRRK2 G2019S in astrocytes generated from patient-derived induced pluripotent stem cells. We report the alteration of extracellular vesicle (EV) biogenesis in astrocytes and identify the abnormal accumulation of key PD-related proteins within multivesicular bodies (MVBs). We found that dopaminergic neurons internalize astrocyte-secreted EVs and that LRRK2 G2019S EVs are abnormally enriched in neurites and fail to provide full neurotrophic support to dopaminergic neurons. Thus, dysfunctional astrocyte-to-neuron communication via altered EV biological properties may participate in the progression of PD.

*For correspondence: aureliederus@gmail.com

## Introduction

The loss of dopaminergic neurons in the *substantia nigra pars compacta* is associated with the severe and debilitating motor dysfunction observed in Parkinson's disease (PD) patients (*Massano and Bhatia, 2012*). The cause of dopaminergic neuron degeneration has been under intense investigation and has revealed the roles played by cellular stressors such as oxidative stress, mitochondrial dysfunction, and disruption of protein degradation pathways (*Dawson and Dawson, 2003*; *Lynch-Day et al., 2012*). The dysregulation of these pathways within the neurons themselves reflects cell-autonomous mechanisms of neurodegeneration, but neurons exist in a highly dynamic multicellular environment in which non-neuronal cells may also contribute to neuron loss (*Allen and Lyons, 2018*; *Liddelow and Barres, 2017*). Among these non-neuronal populations, astrocytes are specialized in detecting and responding to neuronal signals to support neuronal health and function (*Allen and Eroglu, 2017*; *Banker, 1980*), suggesting that impaired astrocytic functions could initiate or accelerate neuron loss. Neurons rely on astrocytes for regulation of their firing activities (*Deemyad et al., 2018*), for trophic and metabolic support (*Allen and Lyons, 2018*), and detoxification of harmful factors (*Ioannou et al., 2019*). However, astrocytes can change from a resting to a reactive state in response to injury (*Zamanian et al., 2012*) or normal aging (*Clarke et al., 2018*). In fact, depending on the nature of the extracellular signals released after injury, astrocytes can protect neurons or induce neurotoxicity (*Liddelow et al., 2017*; *Zamanian et al., 2012*). The role of

astrocytes in promoting neuron loss has been the focus of a number of studies across multiple neurodegenerative diseases including amyotrophic lateral sclerosis (ALS) (*Di Giorgio et al., 2008*; *Haidet-Phillips et al., 2011*), Huntington disease (HD) (*Valenza et al., 2015*), and PD (*di Domenico et al., 2019*; *Gu et al., 2010*). For example, studies showed that astrocytes with the ALS-causing mutation SOD1 G93A produced factors toxic to motor neurons (*Di Giorgio et al., 2008*), as well as failed to secrete factors that support the viability of neurons (*Basso et al., 2013*). Similarly, astrocytes with the LRRK2 G2019S mutation were found to alter neuronal morphology in a non-cell-autonomous fashion and shuttle detrimental astrocyte-derived alpha-synuclein (αSyn) to co-cultured neurons (*di Domenico et al., 2019*). These observations suggest that disease astrocytes undergo important functional changes that ultimately affect the viability of neurons. Further studies of the effects of PD-related mutations on the functionality of astrocytes and their ability to support neuronal survival are greatly needed as much remains to be discovered to fully appreciate the unique contributions of astrocytes to PD pathogenesis.

Cell-to-cell communication is mediated by several highly regulated mechanisms including the secretion of signaling factors directly into the extracellular space or enclosed within extracellular vesicles (EVs). While astrocyte paracrine signaling via the secretion of soluble factors into the extracellular space has been the focus of many studies (*Allen and Eroglu, 2017*; *Spampinato et al., 2019*), astrocyte communication via secreted EVs has not been well characterized. EVs represent a heterogeneous population of vesicles that can be further classified into different subtypes according to their size, biogenesis pathways, and cargo. Microvesicles (~100 nm to 1 μm diameter) are formed by budding of the plasma membrane directly into the extracellular space, and their cargo resembles the cytoplasmic signature of the cell of origin (*Pegtel and Gould, 2019*). Exosomes (~40–120 nm) are formed by a regulated inward budding of early endosomes to form exosome-containing multivesicular bodies (MVBs), as well as by budding of the plasma membrane (*Pegtel and Gould, 2019*). The sorting and loading of exosome cargo is an active and regulated process (*Temoche-Diaz et al., 2019*), and the regulatory factors involved in EV/exosome biogenesis are just beginning to be identified. Among the well-known factors, Rab proteins are essential mediators of MVB trafficking and they regulate endosomal MVB formation/maturation as well as microvesicle budding directly from the plasma membrane (*Pegtel and Gould, 2019*; *Wang et al., 2014*). In addition, membrane remodeling is an essential aspect of MVB/EV formation that appears to be regulated, at least in part, by the endosomal sorting complex required for transport (ESCRT) machinery (*Pegtel and Gould, 2019*; *Schöneberg et al., 2017*). Accumulating evidence suggests that the PD-related protein LRRK2 plays a role in vesicle trafficking, possibly via phosphorylation of Rab GTPase substrates (*Alessi and Sammler, 2018*; *Steger et al., 2016*). The PD-causing mutation G2019S occurs within the kinase domain of LRRK2, resulting in a hyper-active kinase form (*West et al., 2005*), which could affect the MVB/EV pathway. Furthermore, the nature of the EV cargo varies in health and disease (*Chaudhuri et al., 2018*; *Lamontagne-Proulx et al., 2019*). The composition of EVs isolated from the biofluids of PD patients is different from that of healthy controls (*Fraser et al., 2016b*; *Lamontagne-Proulx et al., 2019*; *Shi et al., 2014*) and contains proteins associated with PD, such as LRRK2 and αSyn. The clinical observation that the EV profile is altered in PD patients, and the proposed role of LRRK2 in vesicle trafficking, opens exciting opportunities to further investigate the relationships between EVs and PD pathogenesis in disease-relevant cell types.

Here we made use of human-based genetic models of PD using induced pluripotent stem cells (iPSCs) derived from PD patients with the LRRK2 G2019S mutation and its gene-corrected isogenic control (*Reinhardt et al., 2013*), as well as a second independent pair of iPSCs prepared from a LRRK2 G2019S PD patient and a sex- and age-matched control. The LRRK2 G2019S mutation is one of the most common genetic determinants associated with an increased risk to develop sporadic or familial PD (*Bonifati, 2006*; *Kumari and Tan, 2009*). We show that the LRRK2 G2019S mutation alters the phenotype of MVBs and secreted EVs by affecting the morphology and distribution of MVBs, the morphology of secreted EVs, and by overaccumulating the PD-related proteins LRRK2 and phospho-S129 αSyn in MVBs. Importantly, we show that astrocyte-derived EVs are internalized by dopaminergic neurons, and WT but not LRRK2 G2019S EVs supports neuronal survival. Overall, this study implicates a non-cell-autonomous contribution in dopaminergic neuron loss in PD and suggests pathological dysregulation of EV-mediated astrocyte-to-neuron communication by the PD-related mutation LRRK2 G2019S.

# Results

## Differentiation of LRRK2 G2019S iPSCs and gene-corrected isogenic controls into NPCs and astrocytes

We obtained iPSCs reprogrammed from the dermal fibroblasts of two patients with the LRRK2 G2019S mutation, and one line was gene-corrected to produce an isogenic control (*Reinhardt et al., 2013*) while the second line was paired with iPSCs derived from a sex- and age-matched healthy individual (*Figure 1—figure supplement 1A-C*). This enables the direct comparison of two independent pairs of iPSCs to take into consideration the inherent variability that exists between different patient-derived cell lines. Of particular importance, one of our disease and control iPSC pair differs only in a single-point mutation in *LRRK2*, which is essential to ensure that the experimental observations are a consequence of the mutation of interest, and do not result from differences in genetic background, or from differences that could arise during fibroblast reprogramming of each non-isogenic pair (*Gore et al., 2011*; *Soldner et al., 2009*). Since PD pathology significantly affects the region of the midbrain, we prepared astrocytes from neural progenitor cells (NPCs) patterned toward a midbrain fate. We adapted the protocol developed by Kriks et al. to generate a population of renewable midbrain-patterned NPCs that can differentiate into astrocytes or dopaminergic neurons, depending on exogenous cues (*de Rus Jacquet, 2019*; *Kriks et al., 2011*; *Tcw et al., 2017*). The midbrain and floor plate identity of the NPCs was systematically validated for each iPSC-to-NPC differentiation by RT-qPCR (*Figure 1—figure supplement 1B*), and the differentiated NPCs expressed characteristic NPC markers such as vimentin, SOX1, nestin, and notch1 (*Figure 1—figure supplement 1C*). Midbrain-patterned NPCs carrying the LRRK2 G2019S mutation or its isogenic control were differentiated into astrocytes as described previously (*de Rus Jacquet, 2019*; *Tcw et al., 2017*). As expected, astrocytes expressed the markers GFAP, vimentin, and CD44 as demonstrated by immunofluorescence (*Figure 1A*) and flow cytometry analyses (*Figure 1—figure supplement 2A*). Differentiation was equally effective in WT and LRRK2 G2019S cells, with 100% of the differentiated astrocytes expressing GFAP (*Figure 1—figure supplement 2Bi*). To further demonstrate the successful differentiation of iPSCs into astrocytes, we analyzed gene expression using RNA-sequencing analysis (RNA-seq), including primary human midbrain astrocyte samples in the RNA-seq study to serve as a positive control for human astrocyte identity. iPSC-derived and human midbrain astrocytes expressed similar levels of genes markers of astrocyte identity, including *SOX9* and *SLC2A1* (*Figure 1—figure supplement 2B*). In addition, principal component and unsupervised cluster analyses separated undifferentiated iPSCs, iPSC-derived NPCs, and iPSC-derived astrocytes into independent clusters, demonstrating that our differentiation strategy produces distinct cell types (*Figure 1—figure supplement 2C* and D). Importantly, the transcriptome of iPSC-derived astrocytes showed more similarities to fetal human midbrain astrocytes than to NPCs or iPSCs, further validating their astrocyte identity (*Figure 1—figure supplement 2D*). Lastly, control and LRRK2 G2019S astrocytes showed classic astrocytic functional phenotypes such as spontaneous and transient calcium signaling and synaptosome uptake (*Figure 1—figure supplement 2E* and F).

## Expression of exosome components in iPSC-derived astrocytes is altered by the LRRK2 G2019S mutation

We analyzed the effects of the LRRK2 G2019S mutation on global gene expression patterns in iPSC-derived astrocytes using RNA-seq analysis. Differential gene expression analysis revealed that the isogenic pair differed by a total of 348 genes (*Figure 1B*), and the non-isogenic pair by 563 genes. Classification of these 348 genes using k-means clustering showed that they are distributed between transcripts expressed at high, moderate, and low levels based on their $\log_{10}(1 + \text{normalized sequencing counts})$ value, suggesting that LRRK2 G2019S affects a broad range of targets (*Figure 1B*). Gene ontology (GO) analysis revealed that components of the extracellular compartment are highly upregulated and include the extracellular region, extracellular matrix, and extracellular exosomes (*Figure 1D and F*). The exosome component is one of the most significantly upregulated GO terms in both isogenic and non-isogenic astrocytes, and comprises a total of 67 (isogenic pair) or 95 (non-isogenic pair) genes (*Supplementary files 1 and 2*). The large majority (~98%) of these gene products are described to be enclosed in exosomes (e.g., CBR1) but do not perform specific functions related to EV formation or secretion. Only a few genes are associated with exosome biogenesis and trafficking,

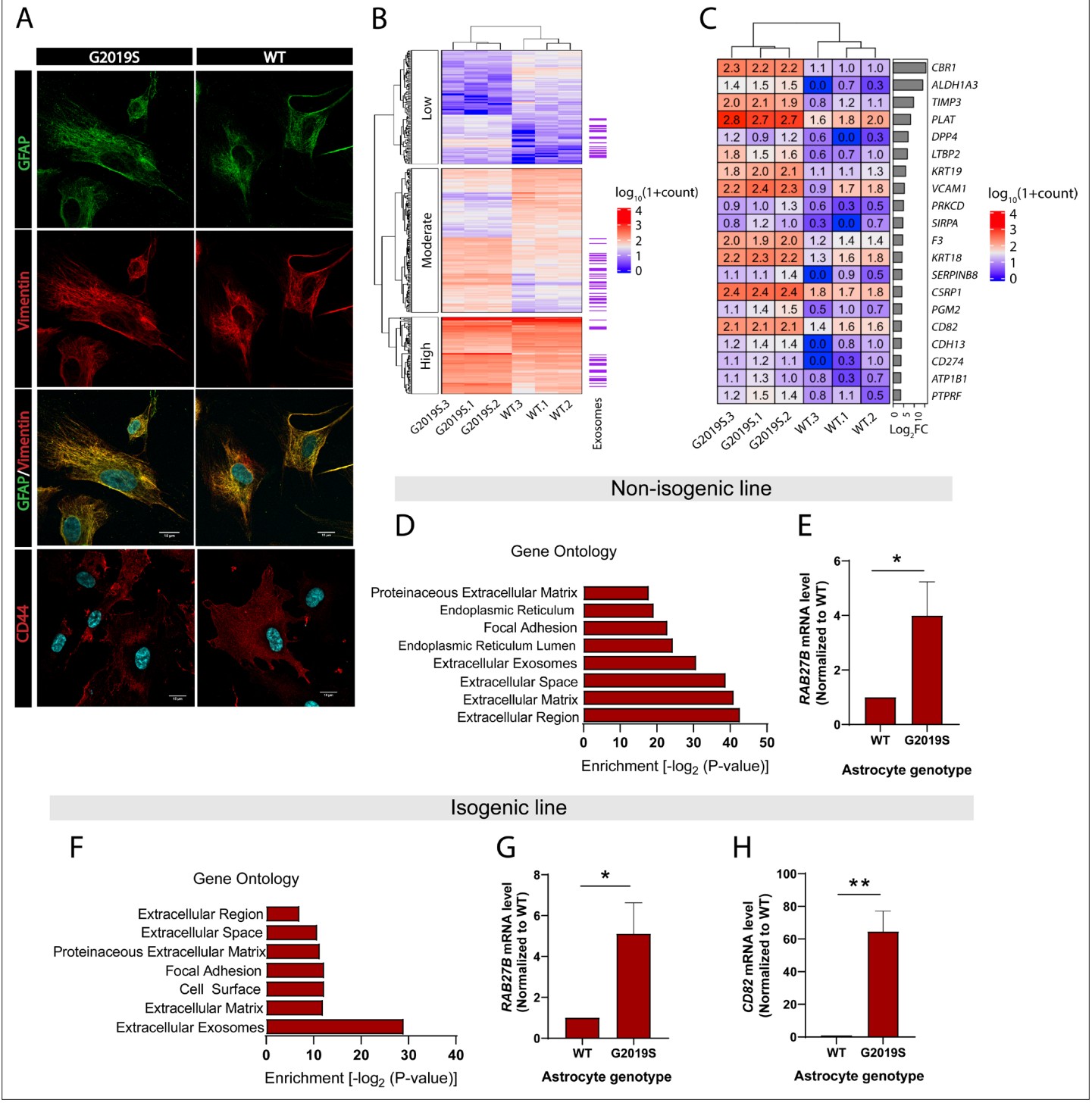

**Figure 1.** Expression of exosome components is dysregulated in LRRK2 G2019S astrocytes. (**A**) Astrocytes were prepared from induced pluripotent stem cells (iPSCs) carrying the Parkinson's disease (PD) mutation LRRK2 G2019S or its isogenic control. Confocal images of immunostained iPSC-derived astrocytes show expression of astrocyte markers GFAP (green), vimentin (red), merged GFAP (green) and vimentin (red) with the nuclear marker DAPI (blue), and merged astrocyte marker CD44 (red) with DAPI (blue). (**B**) Heatmap representing the hierarchical clustering of significantly upregulated and downregulated genes in LRRK2 G2019S vs. WT astrocytes using a 1.4-fold threshold for upregulated genes and a 0.7-fold threshold for downregulated genes, and a false discovery rate of 0.05. Sequencing counts were normalized using the median of ratios method and calculated by the EBseq package in R, as described in Materials and methods, then transformed using $\log_{10}(1 + $ normalized sequencing counts). The genes are separated into three categories based on their $\log_{10}(1 + $ normalized sequencing counts) value using k-means clustering with k = 3 to reveal groups of low-, moderate-, or high-expression genes. Genes encoding exosome-related components are indicated on the right (purple lines, with each indicating one gene). (**C**)

*Figure 1 continued on next page*

*Figure 1 continued*

Heatmap representing the differential expression of 20 genes encoding exosome components in WT and LRRK2 G2019S astrocytes, across three independent biological replicates (labeled 1–3). The values represent the $log_{10}$(1 + normalized sequencing counts) transformation as described in Materials and methods, $log_2$FC represents LRRK2 G2019S vs. WT fold change in gene expression. Genes are sorted by their $log_2$FC value in order of descending fold change. (D–H) Gene ontology analysis of non-isogenic and isogenic lines showing upregulated components identified by RNA sequencing, and Benjamini–Hochberg adjusted p-values were obtained from the Database for Annotation, Visualization and Integrated Discovery (DAVID) tool (D, F). Gene expression validation by qPCR of two exosome components, Rab27b (E, G) and CD82 (H) in non-isogenic and isogenic lines. Data are from three (B, C, D, F) four (G, H), or five (E) independent biological replicates; error bars represent mean + standard error of the mean (SEM). Statistical analysis was performed using two-tailed unpaired Student's t-test with equal standard deviation (s.d.) (*p≤0.05, **p<0.01).

The online version of this article includes the following figure supplement(s) for figure 1:

**Figure supplement 1.** Quality control of induced pluripotent stem cells (iPSCs) and iPSC-derived neural progenitor cells (NPCs).

**Figure supplement 2.** Quality control of induced pluripotent stem cell (iPSC)-derived astrocytes.

and we did not detect differences in the expression of canonical factors that regulate MVB formation (e.g., *VTA1*, *VPS4A*, or *PDCD6IP*). Quantification of gene expression by RT-qPCR showed a 5-fold increase in *RAB27B* (associated with exosome trafficking) and a 64-fold increase in *CD82* (associated with exosome biogenesis) (*Andreu and Yáñez-Mó, 2014*; *Chiasserini et al., 2014*; *Ostrowski et al., 2010*), in LRRK2 G2019S astrocytes compared to WT astrocytes (*Figure 1E, G and H*).

## LRRK2 G2019S affects the size of MVBs in iPSC-derived astrocytes

The exosome pathway is characterized by the formation of MVBs, endosome-derived structures that form exosomes via inward budding of the endosomal membrane (*Pegtel and Gould, 2019*). Using transmission electron microscopy (TEM), we first confirmed that iPSC-derived astrocytes produce MVB-like structures. We found that while both WT and LRRK2 G2019S astrocytes produce MVBs (*Figure 2A*), their average size is genotype-dependent. The area of MVBs in LRRK2 G2019S astrocytes is smaller than in WT astrocytes, with an average of 0.26 μm² (SEM = 0.02079) in LRRK2 G2019S and 0.44 μm² (SEM = 0.03472) in WT cells (*Figure 2B*). MVBs with an area smaller than 0.2 μm² are over-represented in LRRK2 G2019S compared to WT astrocytes, which range in area from 0.1 to 1.2 μm² (*Figure 2C*). Tetraspanins are transmembrane proteins, and the tetraspanin CD63 is enriched in exosomes and widely used as an exosomal marker (*Escola et al., 1998*; *Men et al., 2019*). However, cell-type specificities in the expression of exosomal markers such as CD63 have been documented (*Jørgensen et al., 2013*; *Yoshioka et al., 2013*). We therefore confirmed the presence of CD63-positive MVBs in iPSC-derived isogenic astrocytes by immunofluorescence (*Figure 2D*) and immuno-gold electron microscopy (IEM) (*Figure 2E*). Analysis of IEM images showed an abundance and similar level of CD63 localized within MVBs in WT and LRRK2 G2019S astrocytes (*Figure 2E*, *Figure 3— figure supplement 1A* and B), confirming that CD63 can be used as a marker of MVBs and exosomes in iPSC-derived astrocytes. These results demonstrate morphological defects in the MVBs produced by astrocytes derived from PD patients. Antibodies that target the CD63 protein accurately label astrocytic MVB and are therefore valuable markers to further characterize LRRK2 G2019S-dependent alterations in MVB-related pathways.

## Profiling WT and LRRK2 G2019S EVs secreted by iPSC-derived astrocytes

The alteration of MVB size and the dysregulated expression of key components involved in MVB formation (such as *RAB27B*) suggests that astrocytic EV signaling may be affected by the G2019S mutation. We therefore investigated if the LRRK2 G2019S mutation could affect the release and morphology of astrocyte-secreted EVs. We first analyzed iPSC-derived astrocytes by TEM and confirmed the presence of EVs that appear to be secreted by exocytosis (*Figure 3A*), and by a mechanism that resembles membrane budding or shedding (*Figure 3—figure supplement 1C*). We then collected and processed astrocyte conditioned media (ACM) by ultracentrifugation. Successive rounds of ultracentrifugation resulted in the isolation of an EV-enriched pellet, immediately prepared for cryogenic electron microscopy (cryo-EM) and nanoparticle tracking analysis (NTA), or stored for later use (*Figure 3B*). EVs form a heterogeneous population differing in their biogenesis pathway (endosomal or plasma membrane budding) and cargo (*Pegtel and Gould, 2019*; *Temoche-Diaz et al., 2019*). The unique characteristics of different EV populations are not fully understood, but the documentation of their size range and

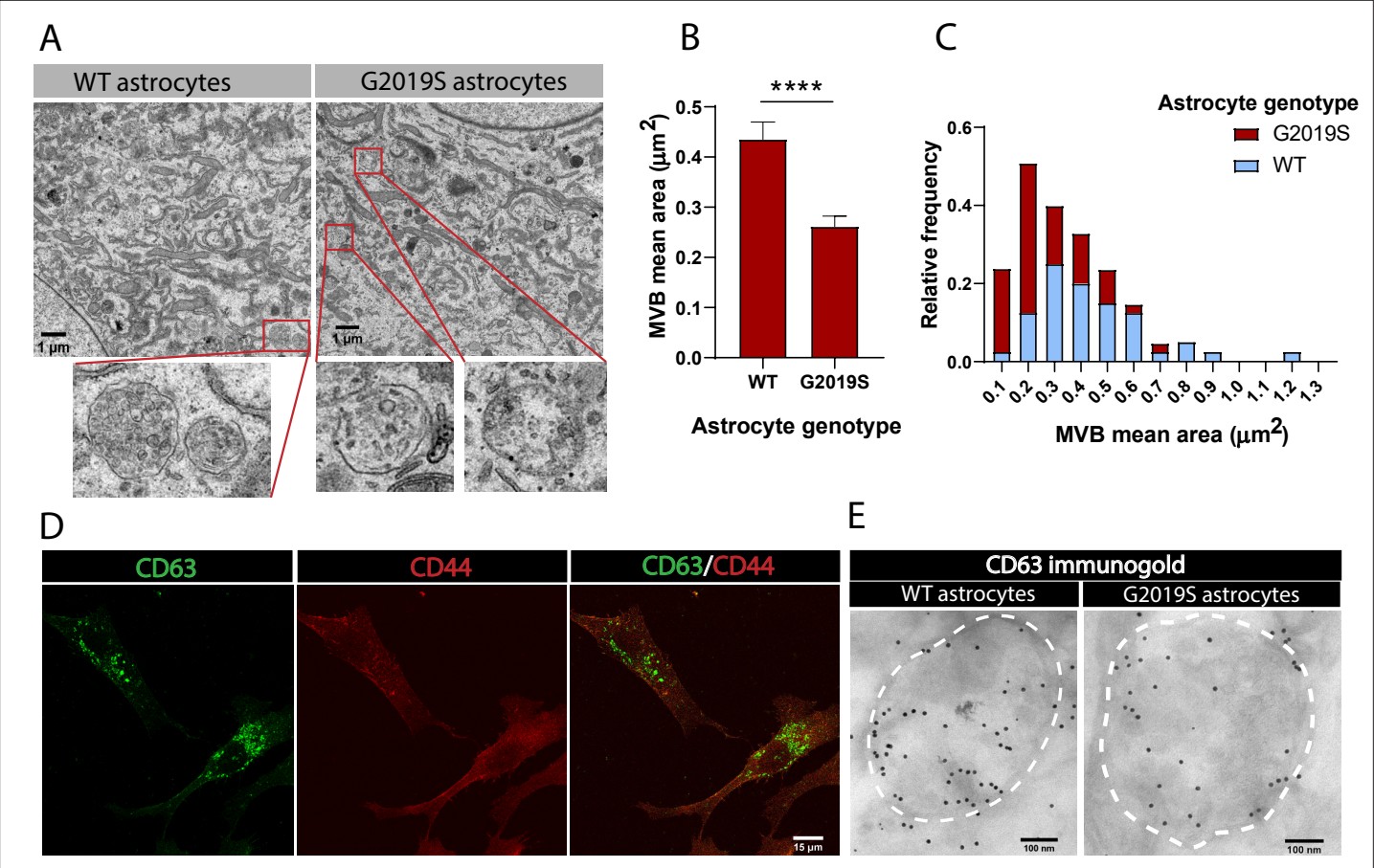

**Figure 2.** Multivesicular bodies (MVBs) in LRRK2 G2019S astrocytes are smaller than in WT astrocytes. (**A**) Transmission electron microscopy (TEM) images of MVBs in WT and LRRK2 G2019S astrocytes. For illustration purposes, the red boxes in the top panel indicate MVBs in the cytoplasm of astrocytes and the lower panel shows a zoomed-in view of the MVBs. (**B, C**) Quantification of mean area (**B**) and size distribution (**C**) of MVBs identified in TEM images of WT and LRRK2 G2019S astrocytes. Data are sampled from at least 20 cells (≥40 MVBs) in each experimental condition; error bars represent mean + SEM for two independent biological samples (**B**). Statistical analysis was performed using two-tailed unpaired Student's t-test with equal s.d. (****p<0.0001). (**D**) Representative confocal images of astrocytes labeled by immunofluorescence with the exosome marker CD63 (green) and the astrocyte marker CD44 (red). (**E**) Electron microscopy image showing immunogold labeling of CD63 (large gold) in astrocytes. Dashed lines delineate MVB membranes.

molecular signatures is commonly used to describe EV populations. The astrocyte-derived EV pellet is enriched in exosomes, as demonstrated by the expression of eight exosomal markers and the absence of cellular contamination (*Figure 3—figure supplement 1D*). NTA quantification showed that the number of secreted EVs does not differ between LRRK2 G2019S and isogenic control (*Figure 3C*), and it appears that LRRK2 G2019S particles have a slightly different size distribution compared to WT particles (*Figure 3D*). It should be noted that TEM and NTA are methods traditionally used to estimate the size distribution of EVs, but their accuracy is often challenged by sample processing artifacts and technical biases (*Pegtel and Gould, 2019*). To overcome these limitations, we complemented the NTA results with cryo-EM analysis of the size of EVs secreted by WT and LRRK2 G2019S isogenic astrocytes. EVs mostly displayed a circular morphology (as opposed to the cup-shaped morphology observed by TEM; *Figure 3E*), but a variety of other shapes were also observed (*Figure 3—figure supplement 1E*). Cryo-EM analysis confirmed that WT astrocyte-secreted EVs display a large range of sizes, from 80 nm to greater than 600 nm in diameter, with differences between WT and mutant populations (*Figure 3F*). The cryo-EM data suggested that mutant astrocytes secreted fewer particles in the 0–120 nm size range. The discrepancy with the NTA results could be explained by the following: (1) in contrast to cryo-EM, NTA does not discriminate EVs from cell debris, which could bias the quantification and increase the number of small particles quantified (*Noble et al., 2020*); and (2)

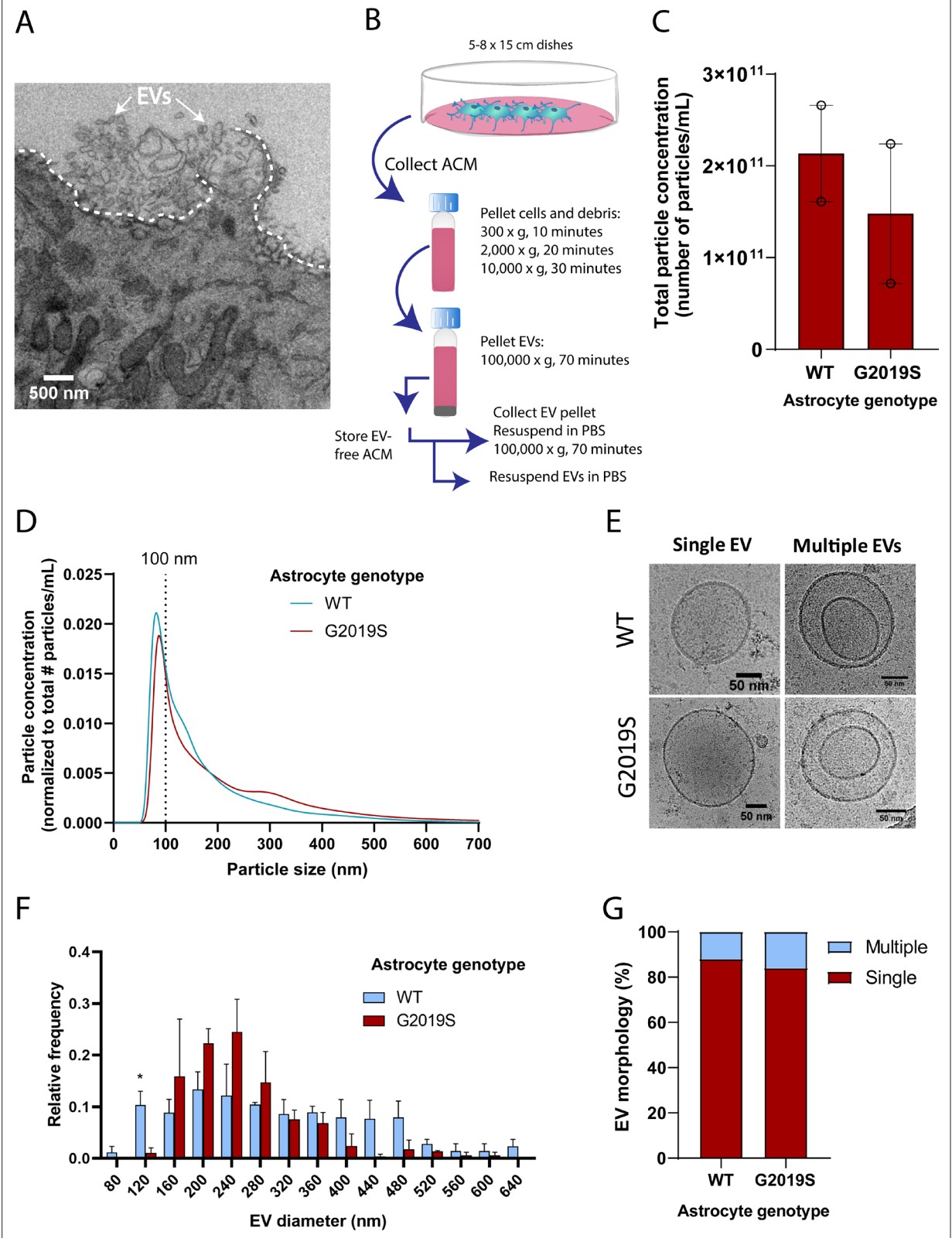

**Figure 3.** The size distribution of extracellular vesicles (EVs) secreted by astrocytes is altered in the LRRK2 G2019S mutant. (**A**) Transmission electron microscopy (TEM) images demonstrate that induced pluripotent stem cell (iPSC)-derived astrocytes actively produce and secrete EVs by exocytosis. The dashed line delineates the plasma membrane. The white arrows indicate EVs. (**B**) Overview of the procedure to isolate EVs by ultracentrifugation. ACM: astrocyte conditioned medium; Cryo-EM: cryogenic electron microscopy. (**C**) Nanoparticle tracking analysis (NTA) quantification of the number

*Figure 3 continued on next page*

*Figure 3 continued*

of particles (i.e., EVs) isolated in WT or LRRK2 G2019S ACM by ultracentrifugation as described in (**B**). (**D**) Graph showing the distribution of isolated EVs by particle size. Data in (**C, D**) are from two independent biological replicates; error bars represent mean + SEM. Statistical analysis was performed using two-tailed unpaired Student's t-test with equal s.d. (ns: not significant). (**E–G**) Secreted EVs were imaged by cryo-EM (**E**) and their diameter (**F**) and morphology (**G**) was analyzed. Data are from ≥177 EVs isolated from ACM taken from $30.4 \times 10^6$ plated astrocytes for each experimental condition; error bars are mean + SEM for three independent biological replicates. Statistical analysis was performed using two-tailed unpaired Student's t-test with equal s.d. (*p≤0.05).

The online version of this article includes the following figure supplement(s) for figure 3:

**Figure supplement 1.** Analysis of astrocyte-secreted extracellular vesicles (EVs).

studies showed that the size distributions between NTA and cryo-EM differ, the latter enabling the identification of larger particles (*Noble et al., 2020*). However, cryo-EM is a low-throughput methodology that limits data collection to a small sample size and has therefore a lower statistical power than NTA. Quantification of the number of simple vs. multiple EV structures did not reveal differences between the mutant and WT, and represent up to 16% of the EV population (*Figure 3G*). We then sought to complement our EV profiling experiments with an analysis of secreted CD63+ particles, which form one of the known exosomal subpopulations. We previously showed that WT and LRRK2 G2019S MVBs contain similar levels of the CD63 tetraspanin (*Figure 3—figure supplement 1A* and B), and an ELISA-based quantification confirmed that the number of CD63+ EVs remained unchanged between the two genotypes (*Figure 3—figure supplement 1F*). We conclude from these results that the total number and morphology of EVs produced by WT and LRRK2 G2019S astrocytes are similar, but mutant EVs may have a different size distribution compared to WT.

## LRRK2 is associated with MVBs and EVs in iPSC-derived astrocytes

In light of our observations that mutations in LRRK2 result in altered astrocytic MVB and EV phenotypes, we asked if LRRK2 is directly associated with MVBs in astrocytes and if this association is altered by the LRRK2 G2019S mutation. We analyzed and quantified the colocalization of LRRK2 with CD63 (*Figure 4A*), a marker for MVBs (*Beatty, 2008*; *Edgar et al., 2014*), and found that the proportion of LRRK2+/CD63+ structures remains unchanged between WT and LRRK2 G2019S isogenic astrocytes (*Figure 4B*). However, this immunofluorescence analysis does not discriminate between LRRK2 signal localized at the periphery or within CD63+ structures. To reveal the localization of LRRK2 in finer detail, we analyzed IEM images, which demonstrated that LRRK2 (small gold) is localized both in the vicinity of and inside CD63+ MVBs (large gold) (*Figure 4C*). Importantly, small gold LRRK2 was more frequently localized inside MVBs in LRRK2 G2019S astrocytes compared to WT astrocytes – 65% of MVBs in LRRK2 G2019S astrocytes are LRRK2+/CD63+ vs. only 44% in WT astrocytes. LRRK2 G2019S astrocytes contained an average of 1.8 LRRK2 small gold particles per MVB compared to only 0.9 small gold particles in WT astrocytes, and MVB populations containing more than five LRRK2 small gold particles were only observed in LRRK2 G2019S astrocytes (*Figure 4D*). We then performed biochemical assays to determine if astrocytes secrete LRRK2 in EVs, and ELISA analysis confirmed the presence of LRRK2 in EV-enriched fractions isolated from ACM (*Figure 4E*). We conclude that LRRK2 is associated with MVBs in astrocytes and is secreted in astrocyte-derived EVs, and the PD-related LRRK2 G2019S mutation results in the abnormal accumulation of LRRK2 inside MVBs and EVs.

## The LRRK2 mutation G2019S alters MVB distribution in iPSC-derived astrocytes

The proposed role of LRRK2 in vesicle trafficking (*Alessi and Sammler, 2018*; *Steger et al., 2016*), in addition to the dysregulation of factors associated with exosome/MVBs in mutant astrocytes (*Figure 1C–H*), and accumulation of LRRK2 in mutant MVBs (*Figure 4D*) suggested that the spatial distribution of vesicles could be disrupted in LRRK2 G2019S astrocytes. To gain further insight into the functional role of LRRK2 in the MVB pathway, we used immunofluorescence to probe the spatial distribution of CD63+ MVBs in WT and LRRK2 G2019S astrocytes (*Figure 5A*). We measured the distance of CD63+ structures to the nucleus and found that CD63+ MVBs accumulated primarily in the perinuclear region of WT astrocytes in both isogenic and non-isogenic pairs (*Figure 5B and D*). In contrast, CD63+ MVBs were less clustered near the nucleus in LRRK2 G2019S cells. The median distance of CD63+ MVBs to the nuclear membrane was 9.2 μm in LRRK2 G2019S astrocytes compared to 4.8 μm

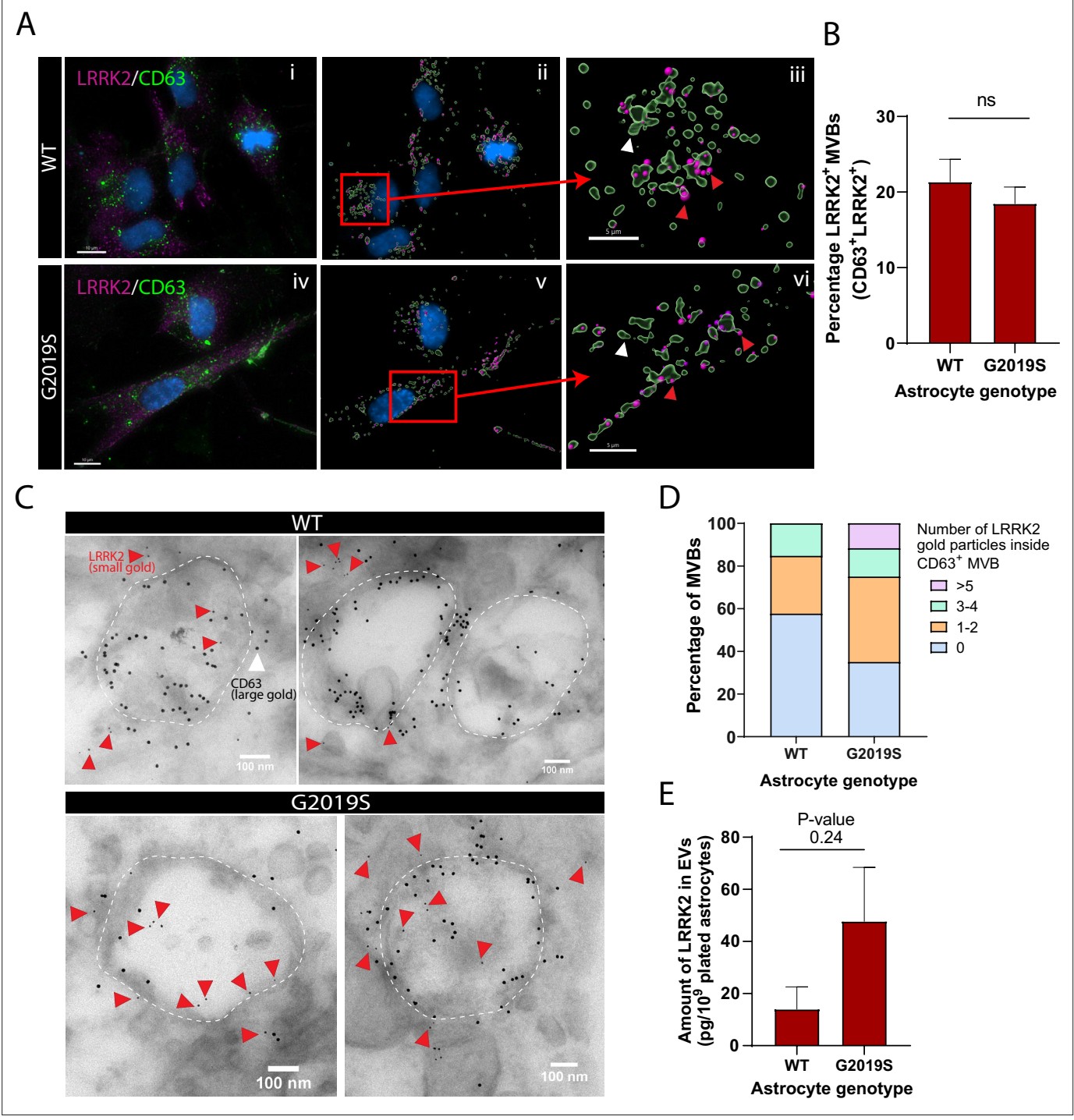

**Figure 4.** LRRK2 is associated with multivesicular bodies (MVBs) in astrocytes and accumulates in the MVBs of LRRK2 G2019S mutant. (**A**) Representative images of WT and LRRK2 G2019S isogenic astrocytes labeled by immunofluorescence with CD63 (green), LRRK2 (purple), and DAPI nuclear stain (blue) (**Ai, Aiv**). The images were analyzed using Imaris software to identify CD63+ MVBs (green surfaces) colocalized with LRRK2 (purple dots) and show the localization of the nucleus (DAPI, blue) (**Aii, Av**). Zoomed-in images shows two populations of CD63+ surfaces: CD63+/LRRK2+ (red arrowhead), and CD63+/LRRK2- (white arrowhead) (**Aiii, Avi**). (**B**) Percentage of CD63-labeled surfaces that are also LRKK2 positive in WT and LRRK2 G2019S astrocytes, quantified with Imaris software using object-based colocalization. Data are from three independent biological replicates, and ≥40 astrocytes (>3000 MVBs) were analyzed per experimental condition; error bars represent mean + SEM. Statistical analysis was performed using two-tailed unpaired Student's t-test with equal s.d. (ns: not significant). (**C**) Immunogold electron microscopy shows the presence of LRRK2 (small gold, red arrowheads) inside and in the vicinity of CD63+ MVBs (large gold) in WT and LRRK2 G2019S astrocytes. The dashed lines indicate the contour of MVBs. (**D**) Distribution of CD63+ MVBs according to their number of internal LRRK2 gold particles. Data are sampled from at least 20 astrocytes (≥59 MVBs) in each

*Figure 4 continued on next page*

Figure 4 continued

experimental condition. The distribution is significantly different in LRRK2 G2019S astrocytes compared to WT astrocytes (p-value = 0.0084, chi-square test). (**E**) Quantification of the amount of LRRK2 in WT or LRRK2 G2019S EV-enriched fractions by ELISA. Data are from at least three independent biological replicates; error bars represent mean + SEM; statistical analysis was performed using two-tailed unpaired Student's t-test with equal s.d. (ns: not significant).

in isogenic controls, and 11.5 μm in LRRK2 G2019S astrocytes compared to 7.5 μm in the non-isogenic control. Furthermore, 75% of the isogenic control CD63[+] MVBs were localized at a distance to the nucleus smaller than 11.23 μm compared to 19.34 μm in LRRK2 G2019S astrocytes (**Figure 5B and C**). In non-isogenic lines, 75% of the control CD63+ MVBs were localized at a distance to the nucleus smaller than 15.82 μm compared to 23.79 μm in LRRK2 G2019S astrocytes (**Figure 5D and E**). The observation that LRRK2 G2019S disrupts the perinuclear distribution of CD63[+] MVBs suggests that their localization is influenced by the LRRK2 protein, and this cellular organization is altered in the astrocytes of PD patients.

## The LRRK2 G2019S mutation increases the amount of phosphorylated alpha synuclein (Ser129) in MVBs

EVs isolated from the biofluids of PD patients exhibit accumulation of αSyn (**Lamontagne-Proulx et al., 2019**; **Shi et al., 2014**; **Zhao et al., 2018**), a hallmark protein whose phosphorylation at the serine residue 129 (p-αSyn) is correlated with PD pathogenesis (**Anderson et al., 2006**; **Fujiwara et al., 2002**). Since the MVB/EV secretion pathway is altered in our LRRK2 G2019S model of PD, we reasoned that mutant astrocytes might produce αSyn-enriched EVs by accumulating the protein in its native or phosphorylated form in MVBs or EVs. IEM analysis revealed an abundance of p-αSyn (small gold) inside and in the vicinity of MVBs of LRRK2 G2019S iPSC-derived astrocytes, but not isogenic control astrocytes (**Figure 6A**). We observed that 55% of the CD63[+] (large gold) MVBs in LRRK2 G2019S astrocytes are also p-αSyn[+] (small gold), compared to only 16% in WT MVBs. LRRK2 G2019S astrocytes contained on average 1.3 p-αSyn small gold particles per MVB compared to only 0.16 small gold particles in isogenic control astrocytes, and MVB populations containing more than 3 p-αSyn small gold particles were only observed in LRRK2 G2019S astrocytes (**Figure 6B**). When we analyzed the content of EVs by ELISA, we found that total αSyn levels (phosphorylated and non-phosphorylated) in EV-enriched fractions are similar between isogenic controls and LRRK2 G2019S (**Figure 6C**). These results suggest that astrocytes secrete αSyn-containing EVs, and the LRRK2 G2019S mutation appears to alter the ratio of p-αSyn/total αSyn in MVB-related astrocyte secretory pathways.

## Exosomes secreted by iPSC-derived astrocytes are internalized by iPSC-derived dopaminergic neurons

EVs have been suggested to mediate astrocyte paracrine signaling and affect neuronal health (**Basso et al., 2013**; **Chaudhuri et al., 2018**; **Pascua-Maestro et al., 2018**); it is therefore important to understand the pattern of astrocyte-derived EV uptake by dopaminergic neurons in health and disease. We investigated the internalization of WT and LRRK2 G2019S astrocyte-derived exosomes by dopaminergic neurons to determine if there exist genotype-dependent alterations in astrocyte-to-neuron exosome shuttling. To monitor production of EVs by astrocytes and assess uptake by neurons, we fluorescently labeled exosomes in astrocytes with one color (CD63-GFP) and labeled neurons with a second color (tdTomato). Subsequently, astrocytes and neurons were co-cultured, and the uptake of green-labeled exosomes by dopaminergic neurons was monitored by live-cell confocal microscopy (**Figure 7A and B**). We produced a 3D reconstitution of neurons and exosomes, and segmented intracellular and extracellular exosomal populations (**Figure 7B**). We observed the presence of astrocyte-derived exosomes in the somas and neurites of dopaminergic neurons (**Figure 7Bvi**), and endocytosis of extracellular exosomes (**Figure 7—figure supplement 1**). We then sought to determine if the efficiency of exosome uptake by dopaminergic neurons was influenced by the astrocyte genotype. We co-cultured dopaminergic neurons with either WT or G2019S astrocytes, but did not observe a difference in the proportion of exosome-containing neurons. On average, 40% of the dopaminergic neurons had internalized exosomes regardless of the astrocyte genotype (**Figure 7C**). Although there was no difference in the amount of exosomes internalized by neurons exposed to either WT or LRRK2 G2019S astrocytes (**Figure 7D**), exosomes were significantly over-represented in

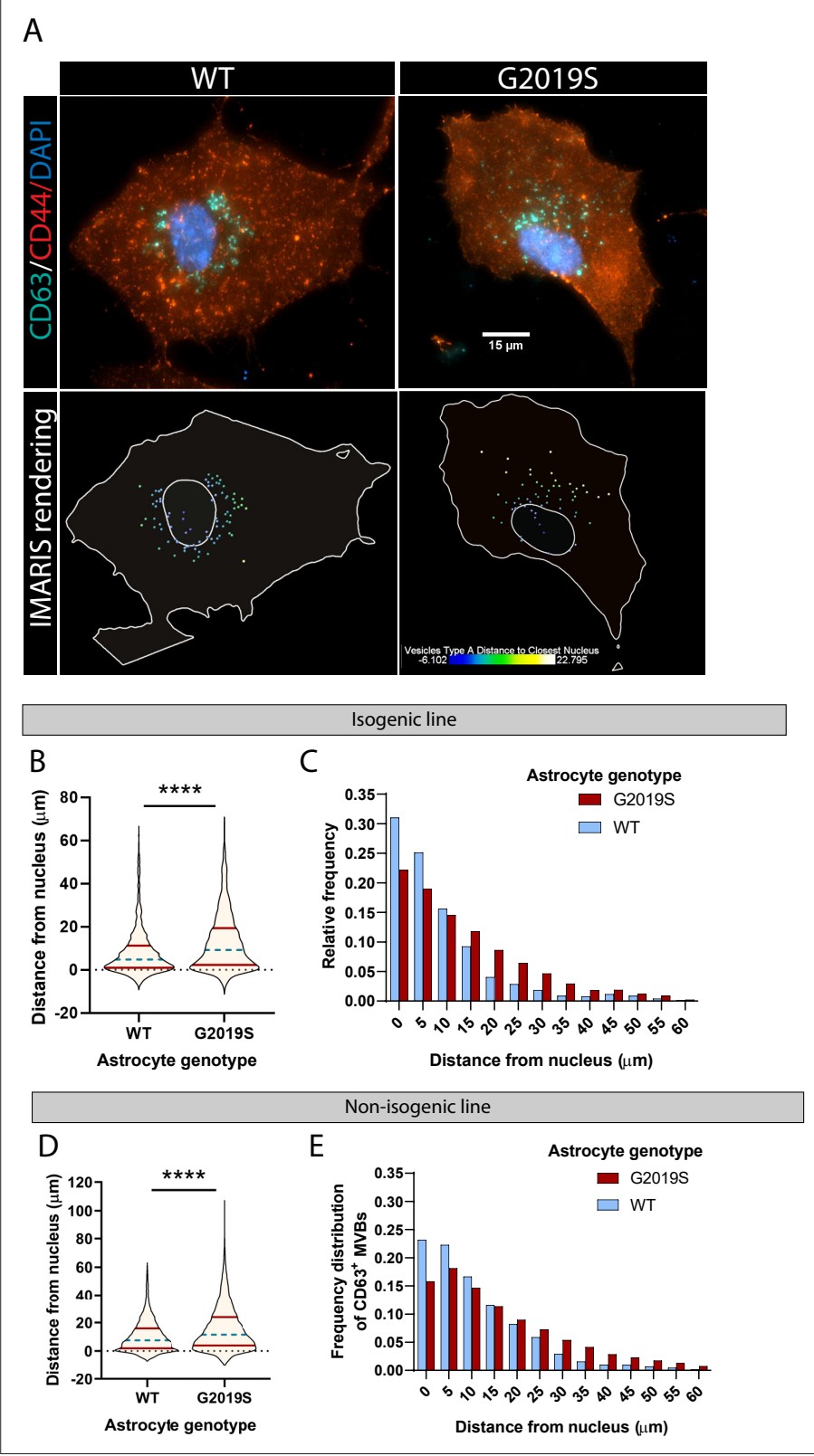

**Figure 5.** The spatial distribution of multivesicular bodies (MVBs) in LRRK2 G2019S astrocytes is altered. (**A**) Representative immunofluorescence images of WT and LRRK2 G2019S astrocytes labeled with the exosome marker CD63 (green), the astrocyte marker CD44 (red), and the nuclear marker DAPI (dark blue). The bottom images show the corresponding Imaris software rendering of CD63+ MVBs, color-coded by distance to the nucleus,

*Figure 5 continued on next page*

de Rus Jacquet *et al*. eLife 2021;10:e73062. DOI: https://doi.org/10.7554/eLife.73062

*Figure 5 continued*

from blue (closest) to white (farthest). The plain white lines indicate the cell boundary (outer line) and nucleus (inner circle). (**B–E**) Quantification of the distance of CD63⁺ MVBs from the nuclear membrane in WT and LRRK2 G2019S isogenic (**B, C**) or non-isogenic (**D, E**) astrocytes using Imaris software 'vesicles distance to closest nucleus' calculation. The violin plot shows the median (blue dashed line) and interquartile range (red solid line) (**B, D**). Data are from three independent biological replicates, 40–70 astrocytes (>3300 MVBs) were analyzed for each experimental condition. Statistical analysis was performed using a Mann–Whitney test (****p<0.0001).

the neurites of neurons exposed to LRRK2 G2019S astrocytes (***Figure 7E and F***). These results show that astrocyte-secreted CD63⁺ exosomes are internalized by neurons, but in the presence of PD astrocytes the pattern of uptake of exosomes is changed, suggesting that the astrocyte-to-neuron shuttling of EV-enclosed proteins may be affected.

## LRRK2 G2019S iPSC-derived astrocytes induce a non-cell-autonomous loss of iPSC-derived dopaminergic neurons

To further determine the functional consequences of the LRRK2 G2019S mutation in astrocyte-to-neuron communication, we investigated the impact of WT or LRRK2 G2019S astrocytes on neuronal health. We co-cultured astrocytes and dopaminergic neurons for 14 days and observed a loss of and

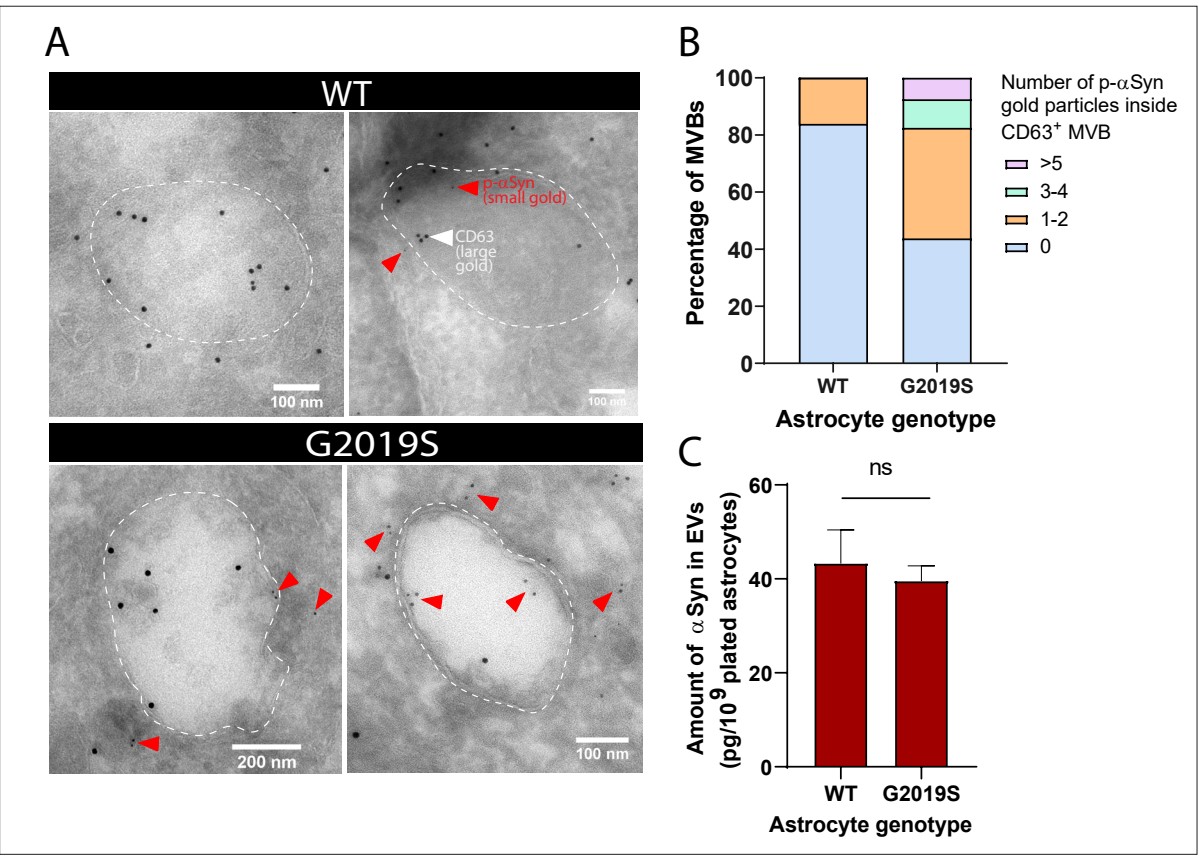

**Figure 6.** Alpha-synuclein and phospho-S129 alpha-synuclein are associated with multivesicular bodies (MVBs) in astrocytes and accumulate in the MVBs of LRRK2 G2019S mutant. (**A**) Immunogold labeling of phospho-S129 alpha-synuclein (p-αSyn, small gold, red arrowheads) and CD63 (large gold) shows localization of p-αSyn inside and in the vicinity of MVBs in astrocytes. The low abundance of small gold particles in the control sample is consistent with the observation that healthy brain tissues contain low levels of p-αSyn (***Fujiwara et al., 2002***). The dashed lines indicate the boundary of the MVBs. (**B**) Distribution of CD63⁺ MVBs according to their number of internal p-αSyn gold particles, in WT and LRRK2 G2019S astrocytes. Data are sampled from at least 20 astrocytes (30–79 MVBs) for each experimental condition. The distribution is significantly different in LRRK2 G2019S astrocytes compared to WT astrocytes (p-value=0.0014, chi-square test). (**C**) Quantification of the amount of αSyn in WT and LRRK2 G2019S extracellular vesicle (EV)-enriched fractions by ELISA. Data are from seven independent biological replicates; error bars represent mean + SEM; statistical analysis was performed using two-tailed unpaired Student's t-test with equal s.d. (ns: not significant).

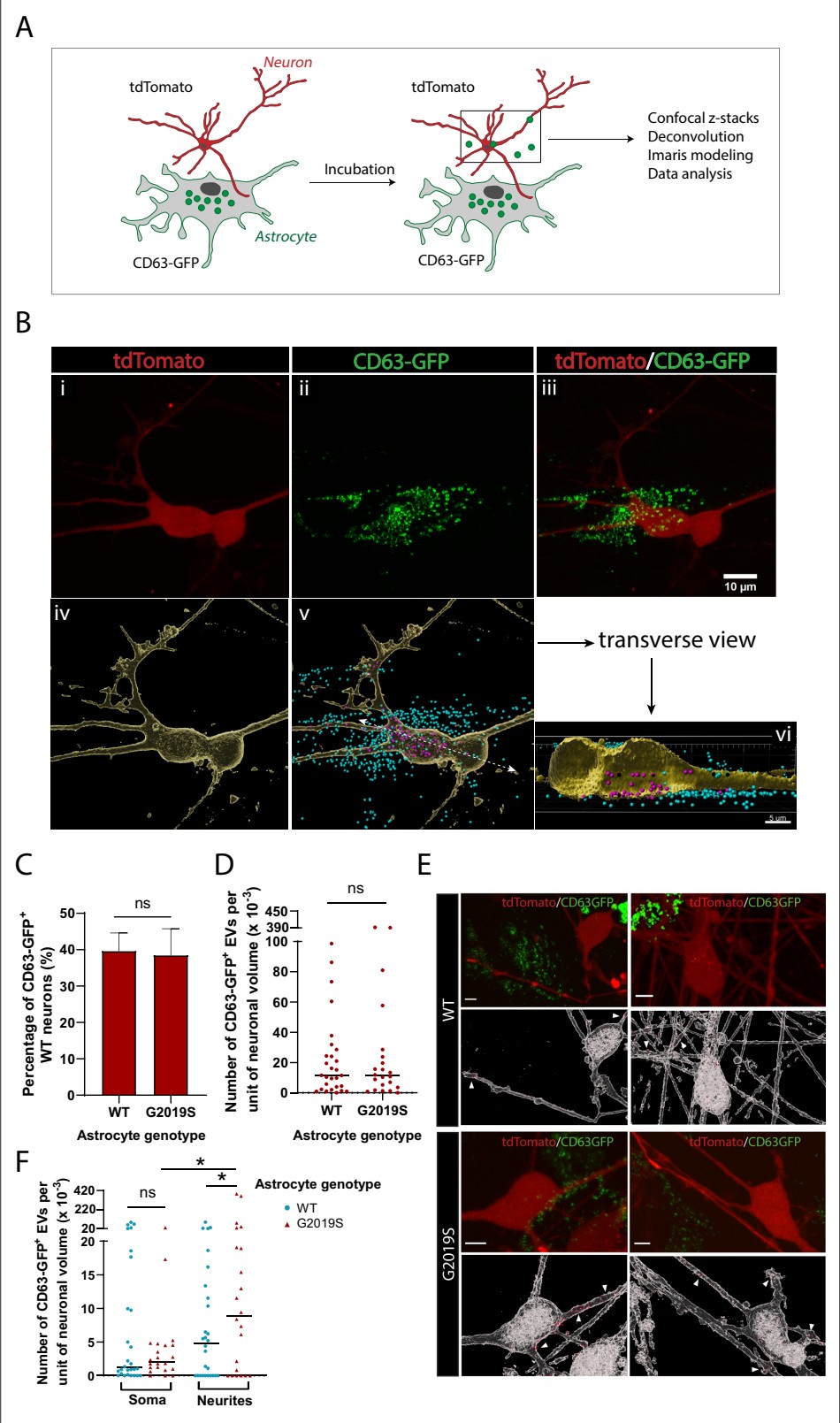

**Figure 7.** Dopaminergic neurons internalize exosomes secreted by WT and LRRK2 G2019S astrocytes. (**A**) WT dopaminergic neurons were transduced with rh10-CAG-tdTomato (red), and WT or LRRK2 G2019S astrocytes were transduced with lenti-CD63-GFP (green) to produce green-labeled exosomes. Neurons and astrocytes were co-cultured, and uptake of CD63-GFP exosomes by neurons was monitored by live-cell confocal microscopy,

*Figure 7 continued on next page*

*Figure 7 continued*

followed by deconvolution and Imaris modeling. (**B**) Confocal images of tdTomato neurons (**Bi**), CD63-GFP astrocytes (**Bii**), and the merged image (**Biii**). The corresponding Imaris software rendering represents tdTomato neurons in yellow (**Biv**), and the CD63-GFP exosomes in blue (outside the neurons) or purple (inside the neurons) (**Bv**). A transverse view of a neuron shows purple-labeled exosomes inside the somas and neurites (**Bvi**). (**C**) Quantification of the percentage of neurons with internalized WT or LRRK2 G2019S CD63-GFP exosomes at the time of live-cell imaging. (**D**) Quantification of the number of CD63-GFP exosomes per unit of neuronal volume. (**E**) Confocal images of tdTomato neurons co-cultured with WT or G2019S CD63-GFP isogenic astrocytes, and the corresponding Imaris software rendering representing neurons in white and CD63-GFP exosomes in red. White arrowheads show exosomes inside neurites. Scale bar: 5 μm. (**F**) Quantification of the number of CD63-GFP exosomes per unit of soma or neurite volume. For all datasets: data are from three independent biological replicates, ≥80 neurons were analyzed for each experimental condition. The scatter plot shows the median value. Statistical analysis was performed using two-tailed unpaired Student's t-test with equal s.d. (**C, D**), or one-way ANOVA with Newman–Keuls multiple comparisons (**E**) (ns: not significant, *p≤0.05).

The online version of this article includes the following figure supplement(s) for figure 7:

**Figure supplement 1.** Dopaminergic neurons internalize astrocyte-derived extracellular vesicles (EVs) by endocytosis.

morphological changes to dopaminergic neurons in presence of mutant astrocytes (*Figure 8A–D*). The neuronal dendrites shortened from an average of 60 μm in WT to 45 μm in LRRK2 G2019S isogenic cultures (*Figure 8C*). Similar morphological changes were observed when dopaminergic neurons were co-cultured with the second line of LRRK2 G2019S astrocytes compared to non-isogenic controls (*Figure 8E and F*), and these results are in agreement with a previous report that LRRK2 G2019S astrocytes induce loss of dopaminergic neurons compared to astrocytes derived from a healthy (non-isogenic) donor (*di Domenico et al., 2019*). To further confirm our results, we measured neuron survival using mouse primary midbrain cultures that consist of purified post-mitotic dopaminergic and GABAergic neurons co-cultured with purified astrocytes. A 14 days, co-culture of WT neurons with LRRK2 G2019S astrocytes resulted in loss of 37% more of the dopaminergic neurons than co-cultures containing WT astrocytes (*Figure 8G and H*). Moreover, this neuronal death was selective for dopaminergic neurons (*Figure 8I*), recapitulating a key feature of PD. LRRK2 G2019S astrocytes also reduced the viability of LRRK2 G2019S dopaminergic neurons (*Figure 8J*), but did not affect non-dopaminergic neuronal populations (*Figure 8K*).

To determine if these morphological changes are mediated by direct contact between astrocytes and neurons or if they rely solely on astrocyte secreted factors, WT dopaminergic neurons were cultured in ACM prepared from LRRK2 G2019S or isogenic control astrocytes. After 14 days in culture, the average dendrite length of dopaminergic neurons exposed to WT ACM was 64.4 μm compared to 50.3 μm if the dopaminergic neurons were exposed to basal NB/B27 medium (*Figure 9A*). This observation suggests that WT ACM, but not basal medium, contains neurotrophic factors necessary for dendrite outgrowth. LRRK2 G2019S ACM mimicked the phenotype observed by direct co-culture, with a 26% increase in the number of short neurites (0–40 μm), and the absence of long neurites (160–200 μm) compared to neurons exposed to WT ACM (*Figure 9B*). The average dendrite length of dopaminergic neurons cultured in basal medium or LRRK2 G2019S ACM is equivalent (50.3 μm vs. 49.1 μm, respectively) (*Figure 9A*). When mouse primary midbrain neurons are exposed to ACM prepared from LRRK2 G2019S mouse midbrain astrocytes, shortening of dendrite length is also observed in vulnerable dopaminergic neuron populations (*Figure 9C and D*). In addition, we found that LRRK2 G2019S mouse primary dopaminergic neuron survival is affected by LRRK2 G2019S but not WT ACM (*Figure 9—figure supplement 1*). We conclude that the morphological atrophy in dopaminergic neurons may result from a lack of neurotrophic support by the LRRK2 G2019S astrocyte secretome.

## Two distinct secretory pathways contribute to the trophic support of iPSC-derived dopaminergic neurons by WT but not LRRK2 G2019S iPSC-derived astrocytes

The astrocyte secretome comprises factors directly secreted into the medium as well as factors enclosed in EVs. To determine which of these two populations are responsible for the effects we

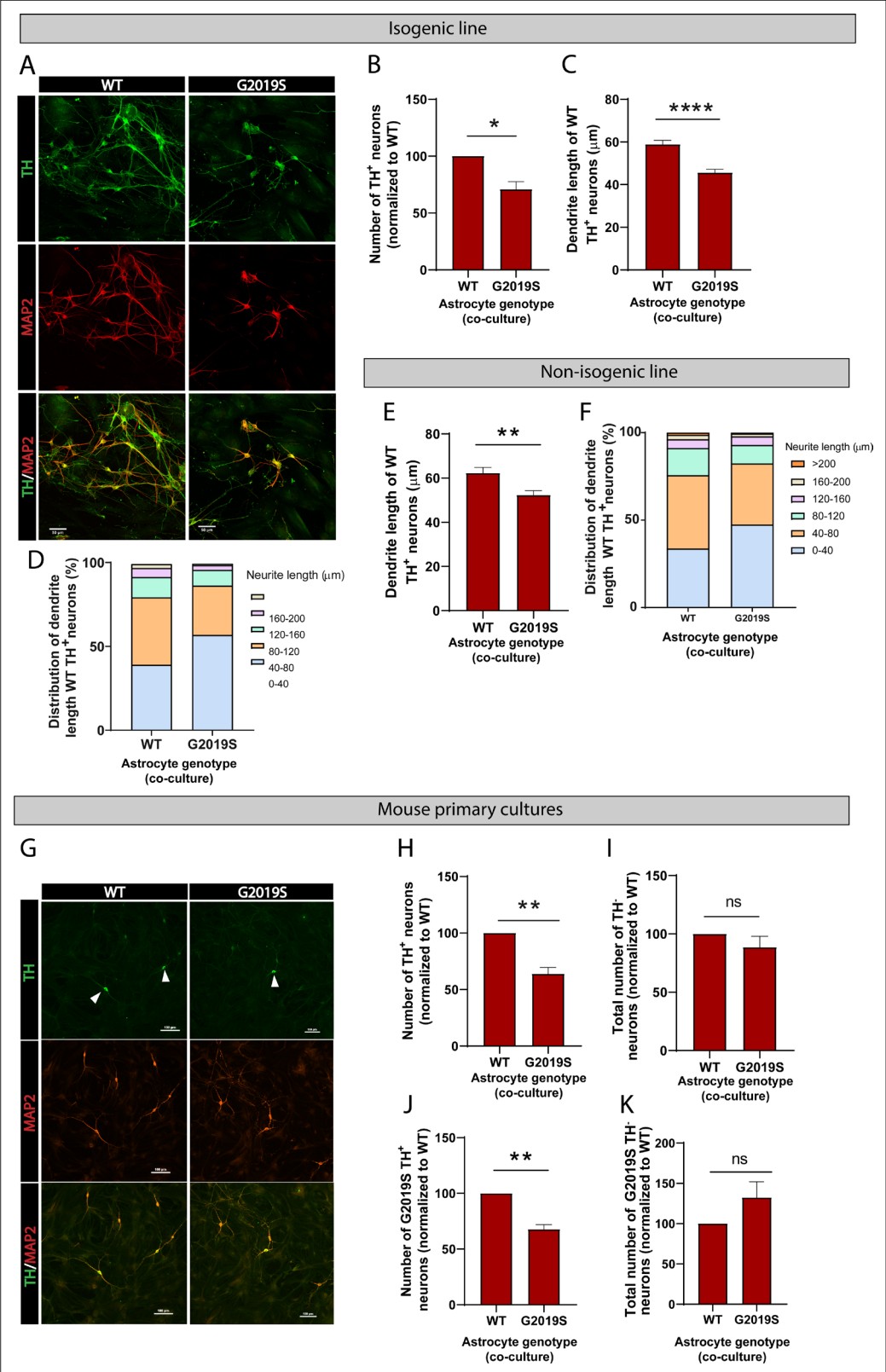

**Figure 8.** LRRK2 G2019S astrocytes affect the viability and morphology of dopaminergic neurons. (**A**) Confocal images showing neurons labeled by immunofluorescence with the pan-neuronal and dendrite marker MAP2 (red), the marker for dopaminergic neurons TH (green), and the merged images (MAP2, red, and TH, green). MAP2: microtubule-associated protein 2; TH: tyrosine hydroxylase. (**B–F**) Quantification of neuron viability (number of

*Figure 8 continued on next page*

*Figure 8 continued*

dopaminergic neurons remaining in culture) (**B**), average dendrite length (**C, E**), and dendrite length distribution (**D, F**) after 14 days in culture with WT or LRRK2 G2019S isogenic (**B–D**) or non-isogenic (**E, F**) astrocytes. Viability data are from five (**B, D**) independent biological replicates, and at least 500 neurons were counted per experimental condition and biological replicate. WT and 530 LRRK2 G2019S neurons were counted (**B**). Dendrite length data are from three independent biological replicates, and more than 300 (**E, F**) or 500 (**C, D**) neurites were measured for each experimental condition. (**G**) Confocal images showing WT mouse primary neurons co-cultured with WT or LRRK2 G2019S mouse primary astrocytes. Neurons are labeled by immunofluorescence and images show MAP2 (red), TH (green), and the merged images (MAP2, red, and TH, green). White arrowheads mark TH$^+$/MAP2$^+$ dopaminergic neurons. (**H, I**) The cells were scored for TH$^+$/MAP2$^+$ dopaminergic (**H**) or TH$^-$/MAP2$^+$ non-dopaminergic neuron survival (**I**) after 14 days co-culture with WT or LRRK2 G2019S astrocytes. (**J, K**) LRRK2 G2019S mouse primary neurons were co-cultured with mouse WT or LRRK2 G2019S astrocytes as described in (**H, I**) and scored for relative viability of TH$^+$/MAP2$^+$ dopaminergic (**J**) or TH$^-$/MAP2$^+$ non-dopaminergic neurons (**K**). Viability data for all mouse primary culture experiments are from three independent biological replicates, and at least 200 neurons were counted for each experimental condition and biological replicate. For all datasets, error bars represent mean + SEM; statistical analysis was performed using two-tailed unpaired Student's t-test with equal s.d. (*p≤0.05, **p<0.01, ****p<0.0001). Results shown in panel (**B**) support similar observations recently documented in a non-isogenic induced pluripotent stem cell (iPSC)-based model system (*di Domenico et al., 2019*).

observe on neurons, we assessed the neurotrophic potential of EVs vs. EV-free medium prepared from LRRK2 G2019S or isogenic control astrocytes. We collected ACM and prepared EV-free and EV-enriched fractions using sequential ultracentrifugation (*Figure 3B*). Dopaminergic neurons were cultured for 14 days in basal medium, or EV-free medium prepared from LRRK2 G2019S or isogenic control astrocytes. WT EV-free medium promoted neuronal health, as evidenced by an average dendrite outgrowth of 60.3 μm compared to 51 μm in basal medium (*Figure 9E*). In contrast, LRRK2 G2019S EV-free medium did not support dendrite outgrowth, as evidenced by an average length of 48.7 μm. The analysis of the distribution of dendrite lengths shows a greater proportion of short neurites (0–40 μm) in dopaminergic neurons exposed to basal medium or LRRK2 G2019S EV-free medium (*Figure 9F*). We conclude that soluble factors found in WT but not LRRK2 G2019S EV-free ACM mediate neuron growth.

Next, we tested the hypothesis that failure of mutant astrocytes to directly secrete trophic factors into the media was also accompanied by a failure to secrete EV-enclosed beneficial molecules., We therefore cultured dopaminergic neurons with basal medium, or with WT or LRRK2 G2019S EV-enriched fractions. WT EVs supported a 30% increase in dendrite length compared to basal medium (*Figure 9G*), which is higher than the 18% increase supported by WT soluble factors (*Figure 9E*). Interestingly, LRRK2 G2019S EVs were not as efficient as their WT counterparts, but they supported a 11% increase in dendrite length (*Figure 9G*) and reduced the proportion of short neurites (0–40 μm) (*Figure 9H*). These results suggest that WT astrocytes secrete neurotrophic factors not only via direct release into the extracellular compartment, but also via production and release of EVs. Supplementation of the neuronal basal medium with WT or LRRK2 G2019S EVs resulted in neuron growth, but LRRK2 G2019S EVs provided only a marginal trophic support compared to WT EVs. Together with the previous observation that LRRK2 G2019S affects the exosome/EV component in astrocytes and the distribution of internalized EVs in neurons, these data support a model of astrocyte-to-neuron trophic support via both direct secretion and EV-mediated shuttling of factors, and these systems appear to be dysregulated in the LRRK2 G2019S model of PD.

The nature of the secreted factors, either those secreted directly into the media or those enclosed in EVs, could be proteins, small signaling molecules, or essential lipids. A GO analysis of downregulated genes in LRRK2 G2019S astrocytes revealed a defect in cholesterol as well as isoprenoid biosynthetic processes (*Figure 10A and B*) in both isogenic and non-isogenic lines. Furthermore, genes associated with cell growth and survival were significantly downregulated in mutant astrocytes, including *BIRC5, FLOT1, HTRA1, PAK1, KCNMA1,* and *LZTS1* (*Figure 10C*). These genes regulate neuronal growth (*BIRC5, FLOT1, PAK1, LZTS1*) and/or function (*FLOT1, PAK1, LZTS1*) (*Contet et al., 2016*; *Du et al., 2020*; *Kawaue et al., 2019*; *Ko et al., 2011*; *Launay et al., 2008*; *Pan et al., 2015*; *Swanwick et al., 2010*), and can be found in EVs (*FLOT1, HTRA1, KCNMA1*) (*Skog et al., 2008*), suggesting that they could be shuttled between neurons and astrocytes via this communication pathway.

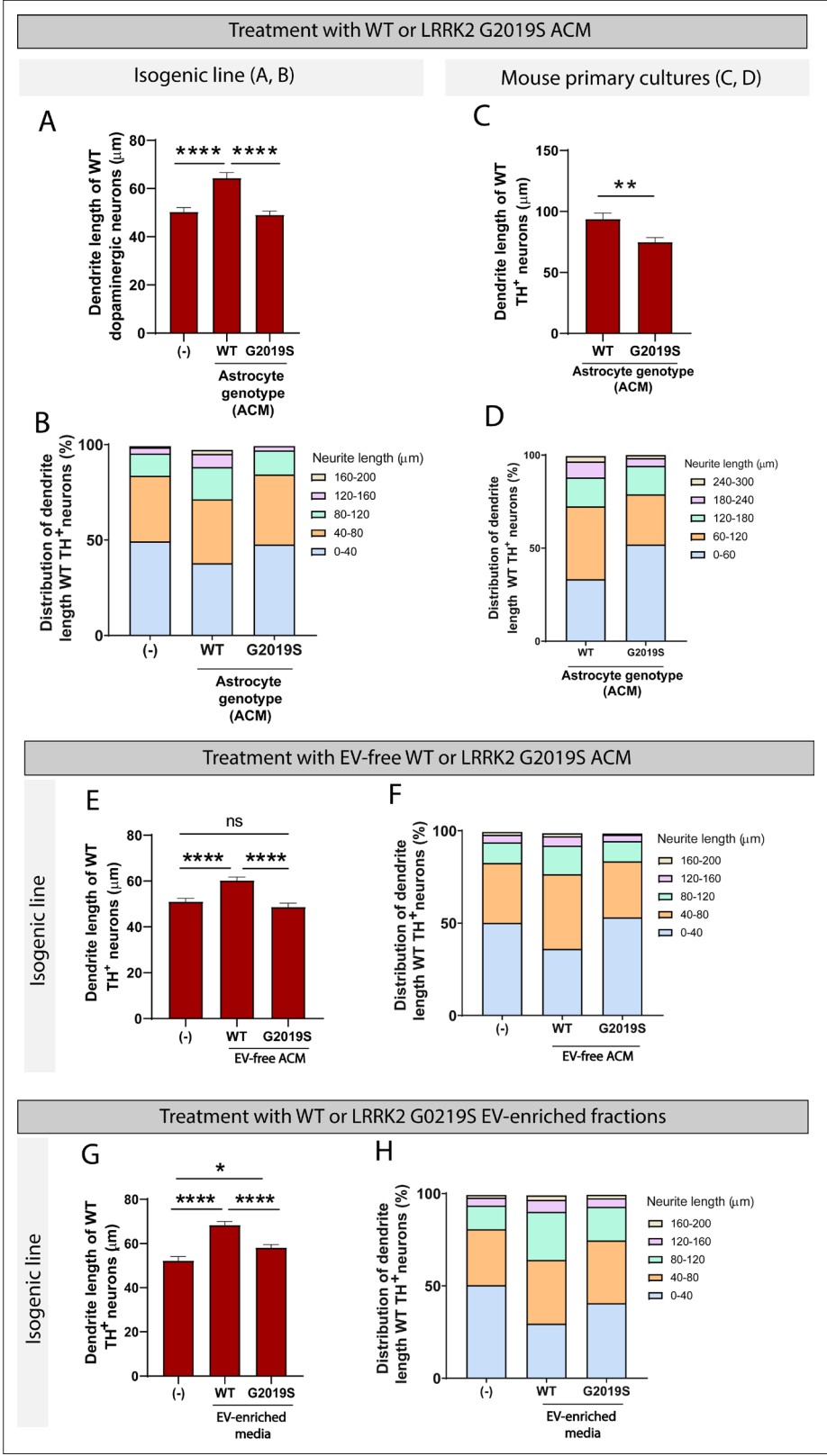

**Figure 9.** WT but not LRRK2 G2019S astrocytes release neurotrophic factors via two distinct secretory pathways to support the growth of dopaminergic neurons. (**A–D**) WT dopaminergic neurons were cultured for 14 days with basal medium (-), WT or LRRK2 G2019S astrocytes conditioned media (ACM), and the resulting dendrite length average (**A, C**) and distribution (**B, D**) were quantified. Data in (**A, B**) were collected using the isogenic induced

*Figure 9 continued on next page*

*Figure 9 continued*

pluripotent stem cell (iPSC)-based model, data in (**C, D**) used the primary mouse culture system. The data are from three independent biological replicates, and ≥450 (**A, B**) or 170 (**C, D**) neurites were measured per experimental condition. (**E, F**) Quantification of the average (**E**) and distribution (**F**) of dendrite lengths of WT dopaminergic neurons cultured for 14 days with basal medium (-), or WT or LRRK2 G2019S extracellular vesicle (EV)-free ACM. Data are from four independent biological replicates, and ≥650 neurites were measured for each experimental condition. (**G, H**) Quantification of the average (**G**) and distribution (**H**) of dendrite lengths of WT dopaminergic neurons cultured for 14 days with basal medium (-), or WT or LRRK2 G2019S EV-enriched fractions. Data are from three independent biological replicates, and ≥450 dendrite were measured for each experimental condition. For all datasets, error bars represent mean + SEM, and statistical analysis was performed using two-tailed unpaired Student's t-test with equal s.d. (**C**) or one-way ANOVA with Tukey's multiple comparisons correction (**A, E, G**) (ns: not significant, *$p \leq 0.05$, ****$p < 0.0001$).

The online version of this article includes the following figure supplement(s) for figure 9:

**Figure supplement 1.** LRRK2 G2019S dopaminergic neurons are sensitive to factors secreted by LRRK2 G2019S astrocytes in a mouse primary culture model.

## Discussion

### Dysregulation of iPSC-derived astrocyte-mediated EV biogenesis in PD

Here we document the dysregulation of the MVB-mediated secretory pathway in human iPSC-derived astrocytes containing the PD-related mutation LRRK2 G2019S. Transcriptomic profiling revealed the misregulated expression of genes encoding proteins found in exosomes in LRRK2 G2109S vs. WT astrocytes (*Figure 1*). Both LRRK2 G2019S MVBs and secreted EVs were smaller than their WT counterparts (Figures 2 and 3D and G), and the mutant CD63$^+$ MVBs lost their perinuclear distribution (*Figure 5*). The observation that LRRK2 G2019S MVBs are less frequently located in the perinuclear area suggests that they may spend less time loading cargo at the Trans-Golgi network, which could in turn produce smaller MVBs and EVs with a different size range compared to WT (*Edgar et al., 2014*; *Pegtel and Gould, 2019*). We did not observe a difference in the number of secreted EVs (total and CD63$^+$ subpopulation) between WT and LRRK2 G2019S astrocytes (*Figure 3C and H*), suggesting that the secretion of at least one population of EVs is independent of the astrocyte genotype. However, we cannot rule out that sub-populations of CD63$^-$ EVs may be differentially secreted in mutant astrocytes.

Accumulating evidence supports a role for LRRK2 in vesicular membrane-related pathways, which could be mediated by the kinase activity of LRRK2 and the resulting phosphorylation of a subset of Rab GTPases (*Alessi and Sammler, 2018*; *Steger et al., 2016*). Rab GTPases are master regulators of vesicle biogenesis and coordinate intracellular trafficking of vesicular organelles including autophagosomes, lysosomes, and MVBs (*Alessi and Sammler, 2018*). The LRRK2 mutation G2019S results in a gain-of-function phenotype with increased kinase activity and increased phosphorylation of Rab substrates (*Steger et al., 2016*). Thus, the observation that the distribution and phenotype of MVBs and EVs is altered by LRRK2 G2019S (*Figures 2 and 3*) may correlate with dysregulation of Rab GTPase signaling cascades. In fact, genetic manipulation of Rab27b in HeLa cells altered the size and distribution of CD63$^+$ MVBs, but did not change the size or protein content of the secreted EVs (*Ostrowski et al., 2010*). This suggests that while Rab27b may mediate LRRK2 G2019S-related alterations in MVB phenotype in astrocytes, it is likely that additional pathways are affected and alter EV morphology and cargo loading. For example, future studies could elucidate whether the increased kinase activity resulting from the LRRK2 G2019S mutation prevents EV maturation, as well as the rate of astrocyte-derived EV production and release.

We found that two important PD-related proteins, LRRK2 and αSyn, were localized inside MVBs and secreted in astrocyte-derived EVs (*Figures 4 and 6*). Furthermore, the LRRK2 G2019S mutation in astrocytes induced the accumulation of LRRK2 (*Figure 4C and D*) and p-αSyn (*Figure 6A and B*) within MVBs. We observed a trend towards greater levels of LRRK2 in mutant EVs compared to WT, but no genotype-dependent changes in αSyn levels (*Figure 6C*). However, the accumulation of p-αSyn in LRRK2 G2019S MVBs suggests that this aggregation-prone form of αSyn may also accumulate in mutant EVs and is consistent with studies showing that cerebral levels of p-αSyn are elevated in LRRK2 G2019S rodents (*Longo et al., 2017*). Furthermore, the LRRK2 mutation G2019S increases LRRK2 autophosphorylation at serine 1,292 (pS1292), and pS1292-LRRK2 was found in higher levels in EVs isolated from PD patients compared to healthy controls (*Fraser et al., 2016a*; *Wang et al., 2017*).

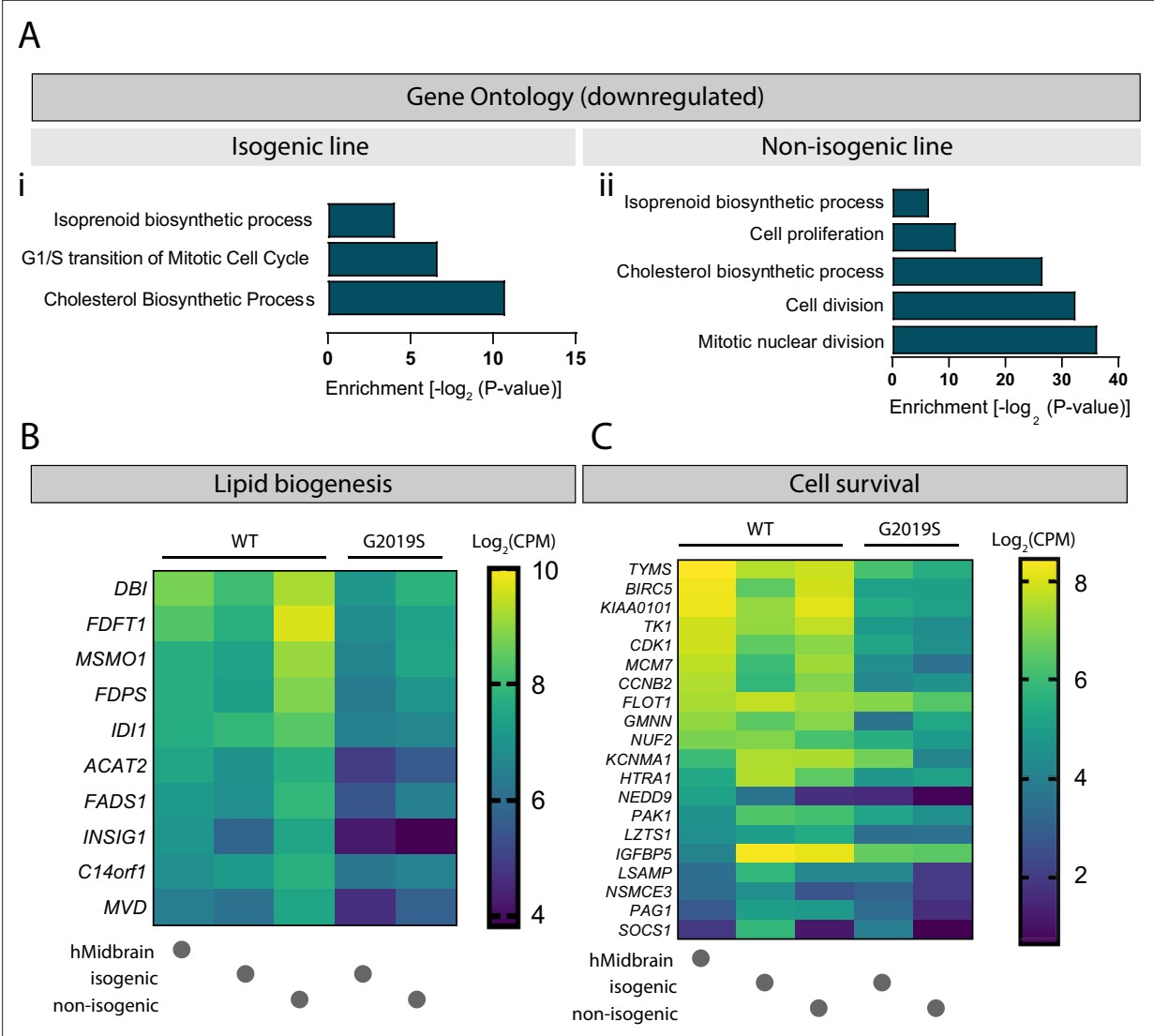

**Figure 10.** Cholesterol biogenesis is impaired in LRRK2 G2019S astrocytes. Gene ontology analysis of isogenic (**Ai**) and non-isogenic (**Aii**) LRRK2 G2019S vs. WT induced pluripotent stem cell (iPSC)-derived astrocytes showing downregulated components identified by RNA sequencing. Benjamini–Hochberg adjusted p-values were obtained from the Database for Annotation, Visualization and Integrated Discovery (DAVID) tool. (**B, C**) Heatmaps showing top significantly downregulated genes associated with lipid biogenesis (**B**) and cell survival (**C**) in LRRK2 G2019S vs. WT astrocytes using a 0.7-fold threshold and a false discovery rate of 0.05. Data shows $\log_2$(CPM) values calculated for WT and LRRK2 G2019S astrocytes prepared using the isogenic or non-isogenic lines, as well as human midbrain fetal astrocytes to use as a reference. Abbreviation: CPM, counts per million.

Further studies are needed to determine if pS1292-LRRK2 and p-αSyn differentially accumulate in EVs secreted from LRRK2 G2019S vs. WT astrocytes, as differences in EV profile between EV derived from healthy or PD astrocytes could open new avenues in the field of biomarker research. The investigation of biofluid-isolated EVs as noninvasive disease diagnostic tools in PD is an emerging field (*Thompson et al., 2016*), and the search for sensitive biomarkers is facing challenges. Many cell types, including cells of the central nervous system (*Fiandaca et al., 2015*, *Shi et al., 2014*) but also non-disease-relevant cell types, contribute to the pool of EVs isolated from biofluids. As a result, EV subpopulations that would best serve as markers of the disease are diluted in the total pool of EVs, which limits the sensitivity of a biomarker assay. Given the dysregulation of the EV secretory pathway in LRRK2 G2019S astrocytes, future studies could evaluate if astrocyte-derived EVs may provide a sensitive measure of disease progression. Astrocyte-derived exosomes could be isolated based on their expression of the Glial Fibrillary Acidic Protein (GFAP) or Glutamine Aspartate Transporter (GLAST)

astrocyte markers (*Venturini et al., 2019*; *Willis et al., 2017*; *Winston et al., 2019*), and their cargoes compared between healthy donors and PD patients.

## Significance of EV trophic support in PD

We also show that dopaminergic neurons internalize EVs secreted by WT or LRRK2 G2019S astrocytes (*Figure 7*). For this investigation, we focused our observations on a well-defined subpopulation of EVs that can be genetically labeled with GFP, thus enabling the documentation of their internalization in live neurons. The EV genotype did not affect the amount of CD63-GFP exosomes taken up by dopaminergic neurons (*Figure 7C*), but LRRK2 G2019S exosomes accumulated to greater levels in the neurites compared to the somas (*Figure 7F*). Our observations suggest that the localization of internalized EVs can vary based on the genotype of the donor astrocyte. However, the functional consequences of EV localization in the recipient neurons are unclear. Studies found that the composition of EVs and the health status of donor and recipient cells can affect EV secretion and uptake (*French et al., 2017*), and could therefore alter astrocyte-to-neuron EV-mediated communication. Our study identified the PD-related proteins LRRK2 and αSyn in EV-enriched fractions (*Figures 4F and 6*), but their effect in the recipient neurons is unknown. Further studies including proteomics-based characterization of WT and LRRK2 G2019S astrocyte-derived EVs are needed to more thoroughly determine their composition, and which relevant cargoes (or lack of) affect the physiology of the recipient neurons.

In support of our observations suggesting the dysregulation of EV-related pathways in LRRK2 G2019S astrocytes, we found that conditioned medium and EV-enriched fractions prepared from mutant astrocytes failed to provide neurotrophic support compared to WT astrocytes (*Figure 9*). We showed that in the presence of LRRK2 G2019S ACM or EVs dopaminergic neurons were not able to maintain their neurite length, as observed by a shortening of the dendrites equivalent to that of neurons cultured in the absence of trophic factors. However, we did not observe excessive neurite shortening (≤50% WT length) or fragmentation indicative of the presence of toxic factors (*di Domenico et al., 2019*; *Liddelow et al., 2017*). Neurons rely heavily on astrocytes via mechanisms involving metabolic coupling whereby astrocytes produce and supply a unique combination of cholesterol, cholesterol precursors, and fatty acids in the form of lipoproteins (*Pfrieger and Ungerer, 2011*). Our study revealed cholesterol and isoprenoid biosynthesis as two of the most significantly downregulated pathways in LRRK2 G2019S astrocytes (*Figure 10*). This finding suggests that mutant astrocytes may have limited capability to produce and supply essential lipids necessary for neuronal growth, function (*Zhang and Liu, 2015*) and lipid-mediated signaling (*Lane-Donovan et al., 2014*). A previous study showed that increased levels of cholesterol derivatives in the developing midbrain promote the growth of dopaminergic neurons (*Theofilopoulos et al., 2019*), and the lack of these astrocyte-secreted factors in our co-culture model could mediate neuron atrophy. Furthermore, our findings of an altered cholesterol profile in patient-derived astrocytes corroborate previous studies showing reduced cholesterol levels in the serum of PD patients and a correlation between serum cholesterol and disease progression (*Huang et al., 2007*; *Jin et al., 2019*). Altered membrane lipid composition and cholesterol metabolism appear to be a common pathological feature among several neurodegenerative disorders including AD, HD, and Niemann–Pick type C disease (*Vance, 2012*), and warrant further investigation as a mechanism of non-cell-autonomous neurodegeneration. In particular, cholesterol appears to be an essential factor in MVB formation and function (*Kobuna et al., 2010*), and impairment of the MVB/EV pathway in LRRK2 G2019S astrocytes could be mediated by changes in cholesterol biogenesis in these cells. Another outcome of our RNA-seq study is the identification of downregulated genes involved in cell growth and survival (*Figure 10C*). For example, decreased levels of the lipid raft protein flotillin-1 (*FLOT1* gene) have been associated with changes in membrane fluidity and reduced glutamate uptake by astrocytes with a PD-related mutation in *PARK7* (*Kim et al., 2016*), which could lead to neurodegeneration. In addition, flotillin-1 contributes to the function of dopaminergic neurons by forming microdomains responsible for the internalization and localization of the dopamine transporter at the plasma membrane (*Cremona et al., 2011*). Flotillin-1 has been detected in EVs (*Skog et al., 2008*), suggesting that it could be shuttled between cell types. Collectively, these observations suggest that LRRK2 G2019S astrocytes fail to deliver important trophic factors to dopaminergic neurons. However, without identifying the molecules responsible, we cannot rule out the possibility that release of neurotoxic molecules by astrocytes also contributes to the neuronal phenotype. Indeed, a previous study proposed a neurotoxic effect of LRRK2 G2019S

astrocytes on dopaminergic neurons, possibly mediated by the transfer and accumulation of astrocytic αSyn in neurons (*di Domenico et al., 2019*). Our work, together with that of di Domenico, suggests that astrocytes may impact dopaminergic neuron function by secreting neurotoxic molecules as well as by failing to secrete EV-enclosed trophic factors and essential lipids.

Evidence from this study and previous reports indicates that the LRRK2 G2019S mutation affects neurons through a variety of mechanisms. Here, we show a non-cell-autonomous effect on neuronal viability that appears to be mediated, at least in part, by impaired astrocyte-to-neuron trophic signaling, but the LRRK2 G2019S mutation can also mediate cell-autonomous dopaminergic neurode-generation (*Reinhardt et al., 2013*). These observations support the idea that the LRRK2 kinase may be involved in a large number of pathways essential to maintain cellular function, cell–cell communi-cation, and brain homeostasis, and disruption of LRRK2 in one cell type has cascading effects on other neighboring cell types. In conclusion, our study suggests a novel effect of the PD-related mutation LRRK2 G2019S in astrocytes, and in their ability to support dopaminergic neurons. This study supports a model of astrocyte-to-neuron signaling and trophic support mediated by EVs, and dysregulation of this pathway contributes to LRRK2 G2019S astrocyte-mediated dopaminergic neuron atrophy.

# Materials and methods

## Key resources table

| Reagent type (species) or resource | Designation | Source or reference | Identifiers | Additional information |
|---|---|---|---|---|
| Antibody | Alexa Fluor-conjugated secondary antibodies | Thermo Fisher Scientific | Multiple | IF (1:1000) |
| Antibody | Anti-alpha-synuclein (rabbit monoclonal) | Abcam | ab138501 | IF (1:200) |
| Antibody | Anti-APC magnetic beads | Miltenyi | 130-090-855 | MACS (1:5) |
| Antibody | Anti-CD133, APC-conjugated (mouse monoclonal) | BD Biosciences | 566596 | MACS (1:20) |
| Antibody | Anti-CD44 (rat monoclonal) | BD Biosciences | 550538 | IF (1:500) |
| Antibody | Anti-CD63 (mouse monoclonal) | Thermo Fisher Scientific | MA1-19281 | IF (1:500). IEM: (1:40) |
| Antibody | Anti-GFAP (mouse monoclonal) | BD Biosciences | 556328 | IF (1:500) |
| Antibody | Anti-LRRK2 (rabbit monoclonal) | Abcam | ab133474 | IF (1:200). IEM (1:20) |
| Antibody | Anti-MAP2 (chicken polyclonal) | Thermo Fisher Scientific | PA1-10005 | IF (1:3000) |
| Antibody | Anti-mouse IgG conjugated to 12 nm gold particles (goat polyclonal) | Jackson ImmunoResearch | 115-205-146 | IEM (1:20) |
| Antibody | Anti-mouse IgG conjugated to 6 nm gold particles (goat polyclonal) | Jackson ImmunoResearch | 115-195-146 | IEM (1:40) |
| Antibody | Anti-phospho alpha-synuclein (S129) (rabbit monoclonal) | Abcam | ab51253 | IEM (1:20) |
| Antibody | Anti-rabbit IgG conjugated to 6 nm gold particles (goat polyclonal) | Jackson ImmunoResearch | 111-195-144 | IEM (1:40) |
| Antibody | Anti-TH (rabbit polyclonal) | PhosphoSolutions | 2025-THRAB | IF (1:500) |

*Continued on next page*

*Continued*

| Reagent type (species) or resource | Designation | Source or reference | Identifiers | Additional information |
|---|---|---|---|---|
| Antibody | Anti-vimentin (rat monoclonal) | R&D | MAB2105 | IF (1:100) |
| Cell line (*Homo sapiens*) | Human midbrain astrocytes | ScienCell | 1850 | |
| Cell line (*Homo sapiens*) | Isogenic control iPSCs | *Reinhardt et al., 2013* | IM2-GC | Prof. Dr. Thomas Gasser (Universitätsklinikum Tübingen) and Prof. Dr. Hans R. Schöler (Max-Planck Institute) |
| Cell line (*Homo sapiens*) | LRRK2 G2019S iPSCs | *Reinhardt et al., 2013* | IM2 | Prof. Dr. Thomas Gasser (Universitätsklinikum Tübingen) and Prof. Dr. Hans R. Schöler (Max-Planck Institute) |
| Cell line (*Homo sapiens*) | Healthy control iPSCs | RUCDR Infinite Biologics | RUID: NN0004085, line ID ND36091 | Dr. Randall T. Moon (Howard Hughes Medical Institute/University of Washington) |
| Cell line (*Homo sapiens*) | LRRK2 G2019S iPSCs | RUCDR Infinite Biologics | RUID: NN0004061, line ID ND33879 | Dr. Randall T. Moon (Howard Hughes Medical Institute/University of Washington) |
| Strain, strain background (*Mus musculus*) | WT mouse | Taconic | C57BL/6NTac | |
| Strain, strain background (*Mus musculus*) | LRRK2 G2019S constitutive knock-in mouse | Taconic | C57BL/6-Lrrk2$^{tm4.1Arte}$ | |
| Chemical compound, drug | Ascorbic acid | Sigma | A92902 | |
| Chemical compound, drug | CHIR99021 | STEMCELL Technologies | 72054 | |
| Chemical compound, drug | Cytosine β-D-arabinofuranoside (AraC) | Sigma | C1768 | |
| Chemical compound, drug | DAPT | Tocris Bioscience | 2634 | |
| Chemical compound, drug | Dibutyryl cAMP | Millipore Sigma | 28745 | |
| Chemical compound, drug | LDN193189 hydrochloride | Sigma | SML0559 | |
| Chemical compound, drug | Purmorphamine | Millipore Sigma | 540223 | |
| Chemical compound, drug | ROCK inhibitor (Y-27632) | STEMCELL Technologies | 72304 | |
| Chemical compound, drug | SB431542 | STEMCELL Technologies | 72234 | |
| Commercial assay or kit | Control library phiX | Illumina | FC-110-3001 | |
| Commercial assay or kit | ExoELISA-ULTRA Complete Kit (CD63 Detection) | System Biosciences | EXEL-ULTRA-CD63-1 | |
| Commercial assay or kit | ExoGlow -Protein EV Labeling Kit (Red) | System Biosciences | EXOGP100A-1 | |
| Commercial assay or kit | KAPA library quantification kit | Roche | 7960298001 | |
| Commercial assay or kit | KAPA SYBR FAST qPCR kit | Roche | 7959397001 | |

*Continued*

| Reagent type (species) or resource | Designation | Source or reference | Identifiers | Additional information |
|---|---|---|---|---|
| Commercial assay or kit | Nextera XT DNA library preparation kit | Illumina | FC-131-1024 | |
| Commercial assay or kit | RevertAid First Strand cDNA Synthesis Kit | Thermo Fisher Scientific | K1622 | |
| Commercial assay or kit | RNA purification kit | Qiagen | 74104 | |
| Commercial assay or kit | TaqMan LRRK2 SNP genotyping assay | Thermo Fisher Scientific | 4351379 | |
| Commercial assay or kit | TaqMan hPSC Scorecard Panel | Thermo Fisher Scientific | A15871 | |
| Other | 96-well plates with optically clear bottom | Ibidi | 89626 | |
| Other | LD columns | Miltenyi Biotech | 130-042-901 | |
| Other | Polycarbonate aluminum bottles with cap assembly | Beckman Coulter | 355618 | |
| Other | Prolong Diamond with DAPI | Thermo Fisher Scientific | P36962 | |
| Peptide, recombinant protein | BDNF (recombinant human) | PeproTech | 450-02 | |
| Peptide, recombinant protein | EGF (recombinant human) | Thermo Fisher Scientific | PHG0311 | |
| Peptide, recombinant protein | FGF basic (146 aa) (recombinant human) | R&D Systems | 233-FB-025 | |
| Peptide, recombinant protein | GDNF (recombinant human) | PeproTech | 450-10 | |
| Peptide, recombinant protein | SHH(C25II) (recombinant mouse) | R&D Systems | 464-SH-200 | |
| Peptide, recombinant protein | TGF-β3 (recombinant human) | R&D Systems | 8420-B3-025 | |
| Recombinant DNA reagent | pCT-CD63-GFP | System Biosciences | CYTO120-PA-1 | |
| Sequence-based reagent | Human CD82_F | This paper | qPCR primer | 5′GGTTTCGTGGAAGGAAGC3′ |
| Sequence-based reagent | Human CD82_R | This paper | qPCR primer | 5′AAGATCAAGTTGAAGAGGAAGAG 3′ |
| Sequence-based reagent | Human Rab27b_F | This paper | qPCR primer | 5′AACTGGATGAGCCAACTG3′ |
| Sequence-based reagent | Human Rab27b_R | This paper | qPCR primer | 5′CTTGCCGTTCATTGACTTC3′ |
| Software, algorithm | Beacon Designer Lite 8.16 | Premier Biosoft (San Francisco, CA) | N/A | |

*Continued on next page*

*Continued*

| Reagent type (species) or resource | Designation | Source or reference | Identifiers | Additional information |
|---|---|---|---|---|
| Software, algorithm | Database for Annotation, Visualization and Integrated Discovery (DAVID) v6.8 | N/A | https://david.ncifcrf.gov | |
| Software, algorithm | EBseq v3.8 | *Leng et al., 2013* | http://www.bioconductor.org/packages/devel/bioc/html/EBSeq.html | |
| Software, algorithm | ComplexHeatmap v3.11 | *Gu et al., 2016* | http://bioconductor.org/packages/release/bioc/html/ComplexHeatmap.html | |
| Software, algorithm | Imaris 9.3 | Bitplane (Belfast, UK) | N/A | |
| Software, algorithm | STAR v.2.7.3a | *Dobin et al., 2013* | https://github.com/alexdobin/STAR/releases, *Alexander, 2021* | |
| Software, algorithm | Starcode v1.1 | *Zorita et al., 2015* | https://github.com/gui11aume/starcode, *Guillaume, 2021* | |

## Culture of iPSCs

Patient-derived iPSCs with the LRRK2 mutation G2019S and the corresponding isogenic control were provided by Prof. Dr. Thomas Gasser (Universitätsklinikum Tübingen) and Prof. Dr. Hans R. Schöler (Max-Planck Institute). The second line of patient-derived iPSCs with the LRRK2 mutation G2019S and the sex- and age-matched healthy control were provided by Dr. Randall T. Moon (Howard Hughes Medical Institute/University of Washington). These iPSCs were produced in collaboration with the Tom and Sue Ellison Stem Cell Core (Institute for Stem Cell & Regenerative Medicine, University of Washington) via episomal reprogramming of human primary fibroblasts as described previously (*Okita et al., 2011*; *Yu et al., 2009*). On day 0, $1 \times 10^6$ cells were electroporated with 1.5 µg each of pCXLE-hsk (Addgene plasmid #27078), pCXLE-huI (Addgene plasmid #27080), and pCXLS-hOCT3/4 (Addgene plasmid #27076) episomal vectors (*Okita et al., 2011*), resuspended in fibroblast media (DMEM high glucose, 10% FBS, 100 U/ml penicillin/streptomycin) and plated into a gelatin-coated 10 cm dish. On day 1, the cultures were replenished with fresh fibroblast media and on day 3 or 4 the transfected cells were passaged onto a gelatin-coated 10 cm dish and maintained in fibroblast media. On day 7, the cells were passaged at three different densities ($1 \times 10^5$, $5 \times 10^4$, $2 \times 10^4$, and $1 \times 10^4$) onto 10 cm dishes pre-plated with mouse embryonic fibroblasts (MEFs), and subsequently cultured in DMEM/F12 media supplemented with 0.1 mM sodium butyrate, 50 nM SAHA and 15% KnockOut serum replacement. Media was changed every 2–3 days and observed daily to identify iPSC colony formation. Colonies were picked and plated onto 24-well plates pre-plated with MEFs and cultured in DMEM/F12 media supplemented with 20% KnockOut serum replacement, 1% sodium pyruvate, 1% non-essential amino acids, 100 µM 2-mercaptoethanol, 100 U/ml penicillin/streptomycin, and 20 ng/ml FGF2. IPSC clones were subsequently expanded on MEF feeders and subcultured using Dispase (Thermo Fisher Scientific) until the lines demonstrated robust growth (usually 3–5 passages). The culture medium was then replaced with mTesR1 or mTesR Plus medium (STEMCELL Technologies, Vancouver, Canada) and MEFs were replaced by a Geltrex basement membrane matrix (Thermo Fisher Scientific). Successful fibroblast reprogramming into iPSCs was validated by confirming expression of pluripotency markers and demonstrating trilineage differentiation potential, as described in the *Immunofluorescence* and *RNA isolation and RT-qPCR* sections of Materials and methods.

The isogenic and non-isogenic iPSCs were genotyped to confirm the presence of the LRRK2 mutation G2019S using a TaqMan SNP genotyping assay (Thermo Fisher Scientific, Waltham, MA), and karyotyped every 2 months to confirm genomic stability (Cell Line Genetics, Madison, WI). The isogenic and non-isogenic iPSCs were cultured in mTesR1 or mTesR Plus medium (STEMCELL Technologies),

regularly passaged as aggregates using ReLeSR (STEMCELL Technologies), and cultured on Geltrex-coated plates (Thermo Fisher Scientific).

## Preparation and culture of iPSC-derived NPCs, dopaminergic neurons, and astrocytes

The step-by-step protocols, media recipes, and quality control experiments to prepare iPSC-derived NPCs, dopaminergic neurons, and astrocytes have been described in extensive detail (*de Rus Jacquet, 2019*). Briefly, iPSCs were plated at a density of $1.6 \times 10^6$ cells/well of a Geltrex-coated 6-well plate and neuralized by dual SMAD inhibition (*Kriks et al., 2011*): day 0 (100 nM LDN193189, 10 μM SB431542), days 1 and 2 (100 nM LDN193189, 10 μM SB431542, SHH 100 ng/ml, purmorphamine 2 μM), days 3 and 5 (100 nM LDN193189, 10 μM SB431542, 100 ng/ml SHH, 2 μM purmorphamine, 3 μM CHIR99021), and days 7 and 9 (100 nM LDN193189, 3 μM CHIR99021). At day 11, the cells were harvested and replated at $7 \times 10^6$ cells/well on a Geltrex-coated 6-well plate. At days 11 and 12, the medium was supplemented with 3 μM CHIR99021, 20 ng/ml FGF2, and 20 ng/ml EGF. At day 13, the differentiation of the iPSCs into midbrain floor-plate NPCs was complete, and the cells were maintained in NPC medium (Neurobasal/B27 without vitamin A, 20 ng/ml FGF2, and 20 ng/mL EGF) supplemented with ROCK inhibitor (Y-27632, STEMCELL Technologies) at each single-cell passage.

To prepare dopaminergic neurons, NPCs were harvested in Accumax cell dissociation solution (Innovative Cell Technologies, San Diego, CA) and plated at $7 \times 10^6$ cells/well on a Geltrex-coated 6-well plate. The next day, NPCs were washed with DPBS and incubated in neuron differentiation medium (NB/B27, 0.5 mM dibutyryl cAMP, 10 μM DAPT, 0.2 mM ascorbic acid, 20 ng/ml BDNF, 20 ng/ml GDNF, 1 ng/ml TGF-β3) (*Kriks et al., 2011*). Neuron differentiation medium was replaced every 2 days for a total of 7–10 days. The resulting culture consisted of neurons and undifferentiated NPCs. To isolate a pure neuronal population, the cells were harvested in Accumax medium, diluted to a density of $1 \times 10^6$ cells in 100 μl MACS buffer (HBSS, 1% v/v sodium pyruvate, 1% GlutaMAX, 100 U/ml penicillin/streptomycin, 1% HEPES, 0.5% bovine serum albumin) supplemented with CD133 antibody (5% v/v, BD Biosciences, San Jose, CA, cat. # 566596), and the CD133+ NPCs were depleted by magnetic-activated cell sorting (MACS) using an LD depletion column (Miltenyi Biotech, San Diego, CA), as described previously (*de Rus Jacquet, 2019*). The final cultures are depleted of non-neuronal cells and contain approximately 70% dopaminergic neurons, the remaining neurons consisting of uncharacterized non-dopaminergic populations.

To prepare astrocytes, NPCs were plated on Geltrex-coated plates at 15,000 cells/cm² in Scien-Cell astrocyte medium (ScienCell Research Laboratories, Carlsbad, CA), passaged every 7 days at 15,000 cells/cm², and cryopreserved after 28 days of differentiation or used for experiments (*de Rus Jacquet, 2019*; *Tcw et al., 2017*).

## Culture of primary human astrocytes

Fetal human midbrain astrocytes were obtained from ScienCell Research Laboratories. To match the gender of the iPSCs used in this study, the primary astrocytes were selected from a female donor. The cells were cultured in Astrocyte Medium (ScienCell Research Laboratories), plated at a density of 40,000 cells/cm², cultured for 48 hr, and harvested in TRIzol for RNA-seq analysis.

## Preparation of mouse primary cultures

Primary cultures were prepared via dissection of P1 to P3 mouse pups obtained from C57BL/6NTac WT mice or constitutive LRRK2 G2019S knock-in mice generated and maintained on the C57BL/6NTac background (*Matikainen-Ankney et al., 2016*) (Taconic Biosciences, NY) using methods approved by the Janelia Institutional Animal Care and Use Committee. The mesencephalic region containing the *substantia nigra* and ventral tegmental area was isolated stereoscopically, and the tissue was digested with papain (Worthington Biochemical, NJ) for 30 min at 37 °C, gently triturated to dissociate the cells, filtered with a cell strainer, and centrifuged to remove papain and cell debris.

## Astrocyte isolation procedure

Astrocytes were prepared as described before with some modifications (*McCarthy and de Vellis, 1980*). The cells were resuspended in Astrocyte Media (ScienCell Research Laboratories) and plated at a density of 70,000 cells/cm² in rat collagen-coated flasks. The next day, the old media was removed

and replaced with fresh astrocyte media every 3 days until the culture became confluent (5–7 days), at which point the cells were frozen, used for experiments, or expanded by passaging once and no more than twice. For neuron-astrocyte co-cultures, the astrocytes were plated at a density of 9600 cells/well of a 96-well plate.

## Neuron isolation procedure and culture

The cells were resuspended in buffer (phenol-red free HBSS with calcium and magnesium, 1% sodium pyruvate, 1% GlutaMAX, 1% HEPES, 0.5% w/v BSA, 5% FBS, 100 U/ml penicillin and streptomycin) and neurons were isolated by MACS using a mouse neuron isolation kit as recommended by the manufacturer (Miltenyi Biotech). Monocultures of neurons were used for ACM experiments and plated at a density of 45,000 cells/well in Geltrex-coated black 96-well plates with clear bottom. The cultures were maintained in Neurobasal A media supplemented with 2% B27 without vitamin A, BDNF (20 ng/ml), GDNF (20 ng/ml), TGFb (4 ng/ml), cAMP (500 μM), 0.5% FBS, 1% GlutaMAX, and 100 U/ml penicillin and streptomycin. At day 0 of the treatment, the maintenance medium was replaced with 100% ACM and replenished every 3–5 days until analysis. For co-culture experiments, neurons were plated directly onto astrocytes at a density of 15,000–20,000 neurons/well in a 96-well plate, and cultured in Neurobasal A medium supplemented with 0.5% FBS, 1% GlutaMAX, and 100 U/ml penicillin and streptomycin. A 50% medium change was performed every 3 days to replenish nutrients while ensuring the presence of secreted factors in the culture. After 24 hr in culture, isolated or co-cultured neurons were treated with 15 μM 5-fluoro-2'-deoxyuridine for 10 hr to prevent growth of non-neuronal cells.

## Preparation of ACM

Astrocytes were first grown to 80–90% confluence, and astrocyte media was replaced by phenol red-free Neurobasal media supplemented with 2% B27 without vitamin A, 1% GlutaMAX, and 100 U/ml penicillin and streptomycin. The volumes used were 10 ml per T75 flasks and 3.3 ml per T25 flasks. When preparing ACM from iPSC-derived astrocytes, 1% FBS was added to the media to improve survival. ACM was collected after 3 days, stored at 4 °C, and the flasks were replenished with fresh media for a second ACM collection. The two batches were then pooled and centrifuged to remove cell debris, sterile filtered, and frozen as single-use aliquots at –20 °C.

## Isolation of EVs and preparation of EV-free ACM

EVs were isolated from FBS-free ACM by ultracentrifugation following an established protocol (*Purushothaman, 2019*). Astrocytes were plated in ScienCell astrocyte medium at a density of 40,000 cells/cm$^2$ in 15 cm dishes (5–8 dishes depending on the number of astrocytes available) and grown to confluence for 24 hr. The cells were subsequently triple washed with PBS to remove cell debris and traces of FBS, and 25 ml of NB/B27 medium was added to each 15 cm dish. The ACM was collected twice, at 48 hr intervals, and processed for EV isolation at 4 °C. First, large debris was removed by serial centrifugation steps in a Sorvall ST8R centrifuge (Thermo Fisher Scientific) at 300 × g for 10 min and at 2000 × g for 20 min, followed by 12,000 rpm for 30 min using a fixed angle type 70 Ti rotor (Beckman Coulter, Brea, CA) and polycarbonate aluminum bottles with cap assembly (Beckman Coulter). The cleared supernatant was subsequently centrifuged at 40,000 rpm for 70 min to pellet the EVs using the type 70 Ti rotor in an Optima XPN-80 ultracentrifuge (Beckman Coulter). The supernatant resulting from this high-speed centrifugation is the EV-free ACM, and it was filtered through a 0.22 μm membrane and stored at –80 °C. The EV pellet was resuspended in PBS and centrifuged at 40,000 rpm for 70 min using the type 70 Ti rotor in an Optima XPN-80 ultracentrifuge (Beckman Coulter). The pellet obtained after this last centrifugation step was the purified EV fraction, which was resuspended in PBS and processed immediately for cryo-EM or aliquoted and stored at –80 °C. The number of CD63$^+$ EVs collected in the EV-enriched fractions was estimated by ELISA (System Biosciences). The ELISA standards provided in the kit are calibrated by NTA to measure the number of exosomes and establish a standard curve based on exosome abundance, and therefore enable an estimation of the number of exosomes in the experimental sample (*Figure 3H*). Furthermore, the total number of EVs and their size distribution was quantified by NTA (System Biosciences).

## Cryo-electron microscopy

Cryo grids were prepared using two types of TEM grids: 400 mesh Quantifoil R1.2/1.3 grids with empty holes and 400 mesh Quantifoil R1.2/1.3 grids coated with a 2 nm thin carbon layer (Quantifoil Micro Tools GmbH, Germany). The grids were rendered hydrophilic by glow discharge using a Pelco easiGlow glow discharger (Ted Pella Inc, Redding, CA) for 60 s, with a current of 15 mA and a vacuum of 0.39 mbar. The cryo grids were then prepared using a Mark IV Vitrobot (FEI Company, OR) with the operation chamber cooled to 4 °C at a set humidity level of 100% . A total of 3 µl of fresh EVs was applied to a glow-discharged grid. The grid was subsequently blotted for 3 s from both sides with blot force 3 and plunged into a liquid ethane bath for vitrification.

The cryo grids were loaded into a Tecnai F20 transmission electron microscope (FEI Company, OR) for imaging through a side entry Gatan 626 cryo holder (Gatan Inc, CA). The microscope was operated at 200 kV and was equipped with a standard Field Emission Gun (s-FEG) and a Gatan K2 Summit direct electron detector (Gatan Inc). Data collection was carried out using the automation software package SerialEM (University of Colorado). To identify the localization of the EVs on the grid, survey maps were first taken for various grid squares at a nominal magnification of 1700× , with a calibrated pixel size of 2.15 nm (corresponding to a calibrated magnification of 2325.6× ). Once the EVs were identified on the survey maps, high-magnification images were taken at a nominal magnification of 29,000× with a pixel size of 0.15 nm (corresponding to a calibrated magnification of 33,333× ). The illumination was set to produce a dose rate of 10.5 electrons per pixel per second, and the K2 camera was operated in counted mode with dose fractionation of 8 s total exposure time for 40 frames (0.2 s per frame). Frames were aligned for motion correction using the default algorithm in SerialEM.

## Transmission electron microscopy

WT and LRRK2 G2019S astrocytes were cultured in 60 mm dishes to 80% confluence and immersion fixed in freshly prepared 2% (w/v) paraformaldehyde (Sigma-Aldrich) and 2% glutaraldehyde (Electron Microscopy Sciences, Hatfield, PA) in Hanks' balanced salt solution (20 mM HEPES, pH 7.4) for 2 hr at room temperature. To facilitate handling, cells were pelleted and resuspended in 2% (w/v) agar as described before (*Paez-Segala et al., 2015*). Cell pellets were then post-fixed in 1% osmium tetroxide (Electron Microscopy Sciences, Hatfield) for 1 hr at room temperature, dehydrated in graded ethanol solutions (30, 50, 70, 80, 95, and 100%), and embedded in Eponate-12 (Ted Pella Inc). Ultrathin sections (60 nm) were cut using a Leica UC6 ultramicrotomes, placed on Formvar/carbon-coated copper grids, and stained with 3% aqueous uranyl acetate (15 min) and Sato's lead solution (5 min). Sections were imaged using a FEI Spirit BioTWIN transmission electron microscope operating at 80 kV, equipped with a 4k × 4k Gatan OneView Camera.

## Immunogold electron microscopy

Cultured astrocytes were fixed in 4% paraformaldehyde and 0.2% glutaraldehyde in Hanks' balanced salt solution (20 mM HEPES, pH 7.4) for 2 hr at room temperature. After pelleting, samples were transferred to carriers filled with 20% BSA and frozen in a HPF Compact 01 high-pressure-freezing machine (M. Wohlwend GmbH, Sennwald, Switzerland) for cryo-immobilization. Samples were freeze substituted in acetone containing 0.1% uranyl acetate plus 3% distilled water in a freeze substitution unit (Leica EM AFS2, Leica Microsystems, Buffalo Grove, IL) at −90 °C for 20 hr. Afterwards, the temperature was brought up slowly to −45 °C; samples were rinsed with pure acetone, and infiltrated with 30, 50, and 70% Lowicryl HM20 resin (Polysciences Inc, Warrington, PA) in acetone for 2 hr each followed by 100% HM20 resin overnight. Samples were embedded in Lowicryl HM20 and UV polymerized for 2 days at −45 °C and 2 days at 20 °C. Postembedding immunogold labeling was performed on ultrathin HM20 sections (70 nm) of astrocytes to detect CD63. The ultrathin sections were conditioned in blocking buffer containing 1% BSA (w/v, Sigma-Aldrich), 0.01% (v/v) Triton X-100, and 0.01% (v/v) Tween 20 in PBS for 20 min at room temperature, and incubated overnight at 4 °C with an anti-CD63 primary antibody (1:20 dilution, Thermo Fisher Scientific, cat# MA1-19281) in blocking buffer. After extensive rinses in PBS, sections were incubated for 2 hr at room temperature with a secondary antibody conjugated to 6 nm gold particles (Jackson ImmunoResearch, cat# 115-195-146) diluted to 1:40 in blocking buffer. Samples were rinsed in PBS, and then fixed with 1% (v/v) glutaraldehyde in PBS for 10 min. Gold particles were amplified with silver enhancement (HQ silver kit from Nanoprobes, Yaphank, NY) for 4 min at room temperature and contrast stained with 3% UA and Sato's triple lead.

For double-labeling experiments, cultured cells were fixed in 4% paraformaldehyde and 0.2% glutaraldehyde in Hanks' balanced salt solution (20 mM HEPES, pH 7.4) for 2 hr at room temperature, pelleted in 2% (w/v) agar, cut into small blocks, and infused in 2 M sucrose containing 15% (w/v) polyvinylpyrrolidone (10 kDa) for freezing. Samples were frozen and ultrathin sections were prepared using a Leica UC6 ultramicrotome equipped with a Leica EM FC6 cryochamber. The cryo-sections were lifted onto nickel grids and stored overnight on gelatin at 4 °C. Before immunolabeling, the gelatin was liquefied at 37 °C, and the sections were washed by floating the sections on droplets of PBS. For immunogold localization, the grids with the attached thin sections were conditioned on droplets containing 1% (w/v) BSA, 0.01% (v/v) Triton X-100, and 0.01% (v/v) Tween 20 in PBS (blocking buffer) for 10 min at room temperature. Grids were incubated for 2 hr in the presence of anti-LRRK2 (1:20 dilution, Abcam, cat# ab133474) and anti-CD63 antibodies (1:40 dilution), or anti-p-αSyn antibody (1:20 dilution, Abcam, cat# ab51253) with mouse monoclonal anti-CD63 (1:40 dilution) in blocking buffer. After extensive rinses in PBS, the sections were incubated for 1 hr at 35 °C with a secondary antibody conjugated to 6 nm gold particles (1:40 dilution, Jackson ImmunoResearch, cat# 111-195-144) and a secondary antibody conjugated to 12 nm gold particles (1:20 dilution, Jackson ImmunoResearch, cat# 115-205-146) in blocking buffer. Lastly, grids with the attached thin sections were rinsed in PBS, fixed with 2% (v/v) glutaraldehyde in PBS for 10 min, and negative or positive contrast stained. Electron micrographs were taken with a FEI Spirit BioTWIN transmission electron microscope operating at 80 kV, equipped with a 4k × 4k Gatan OneView camera.

## Immunofluorescence

Cells were grown on German glass coverslips (Electron Microscopy Sciences) or 96-well plates with an optically clear bottom (Ibidi, Gräfelfing, Germany) coated with Geltrex. The cells were washed once with PBS and fixed by adding 4% paraformaldehyde (Thermo Fisher Scientific) for 20 min at room temperature. The cells were subsequently washed once with PBS and blocked/permeabilized in blocking buffer (0.3% TX-100, 1% BSA, 10% FBS in PBS) for 1 hr at room temperature. The cells were then washed once with PBS and incubated in primary antibody overnight at 4 °C with the following antibodies: anti-MAP2 (1:3000, Thermo Fisher Scientific, cat# PA1-10005), TH (1:500, PhosphoSolutions, Aurora, CO, cat# 2025-THRAB), anti-CD63 (1:500, Thermo Fisher Scientific, cat# MA1-19281), anti-LRRK2 (1:200, Abcam, cat# ab133474), anti-GFAP (1:500, BD Biosciences, cat# 556328), anti-vimentin (1:100, R&D, Minneapolis, MN, cat# MAB2105), and anti-CD44 (1:500, BD Biosciences, cat# 550538) in BSA buffer (1% BSA in PBS). Alternatively, iPSCs were incubated with primary antibodies provided in the StemLight Pluripotency antibody kit (Cell Signaling Technologies, cat#9656) as per the manufacturer's instructions. The next day, the cells were washed twice with DPBS, then incubated in Alexa-conjugated secondary antibodies (Thermo Fisher Scientific) diluted 1:1000 in BSA buffer for 1 hr at room temperature. The cells were washed three times with DPBS, then coverslips were mounted on slides with ProLong Diamond with DAPI (Thermo Fisher Scientific) and cured for 24 hr in the dark at room temperature.

## RNA isolation and RT-qPCR

Confluent cell monolayers grown into individual wells of a Geltrex-coated 6-well plate were first washed with DPBS to remove the cell culture medium and cell debris, then homogenized in TRIzol (Thermo Fisher Scientific), and processed immediately or stored at –80 °C. Chloroform (20% v/v) was added to the TRIzol samples, which were shaken vigorously for 20 s and allowed to sit at room temperature for 3 min. The samples were subsequently centrifuged at 12,000 × g for 15 min, at 4 °C, to separate the aqueous solution from the organic phase. The top, aqueous phase was collected, and mixed 1:1 with ethanol. The RNA-containing samples were processed using an RNeasy kit (Qiagen, Hilden, Germany, cat#74104) following the manufacturer's instructions, and the isolated RNA was processed immediately or stored at –80 °C. cDNA synthesis was carried out with a RevertAid First Strand cDNA Synthesis Kit (Thermo Fisher Scientific) as described in the manufacturer's instructions. Forward and reverse primers for RT-qPCR were designed at exon-exon junctions, using Beacon Designer Lite 8.16 (Premier Biosoft, San Francisco, CA). The primer sequences were as follows:

Human Rab27b: $^{5'}$AACTGGATGAGCCAACTG$^{3'}$ (forward) and $^{5'}$CTTGCCGTTCATTGACTTC$^{3'}$ (reverse).

Human CD82: 5′GGTTTCGTGGAAGGAAGC3′ (forward) and 5′AAGATCAAGTTGAAGA GGAAGAG 3′ (reverse).

Each RT-qPCR measurement was performed in technical duplicates, with a total of four independent biological replicates. Relative mRNA levels were calculated for each gene using the formula

$$2^{-\Delta\Delta Ct} = 2^{-\{(Ct,TG-Ct,\,\beta actin)LRRK2\,G20195-(Ct,TG-Ct,\beta actin)WT}$$

where 'Ct, TG' represents the cycle threshold (Ct) for the target gene (TG), and 'Ct, βactin' represents the cycle threshold for the loading reference *Actb* (β-actin). The KAPA SYBR FAST qPCR (Roche, Basel, Switzerland) master mix was used to prepare the reactions, and RT-qPCR was performed using a Roche LightCycler 480.

Tri-lineage differentiation potential of the newly reprogrammed, non-isogenic iPSCs was assessed using a validated protocol based on embryoid body (EB) formation and spontaneous differentiation (Thermo Fisher Scientific, Pub. No. MAN0013784). Briefly, iPSCs were dissociated as large aggregates using ReLeSR (STEMCELL Technologies), resuspended in EB culture medium (DMEM/F12, 20% KnockOut Serum Replacement, 1% MEM non-essential amino acids, 55 μm 2-mercaptoethanol, 100 U/ml penicillin/streptomycin) supplemented with 4 ng/ml FGF2, and plated on non-adherent tissue culture dishes. After 24 hr, the growth medium was changed to fresh EB medium without FGF2 and replaced every other day. On day 10, EBs were dissociated using Accutase, homogenized in TRIzol, and mRNA and cDNA were obtained as described above. Expression of endoderm, ectoderm, mesoderm, and self-renewal marker genes was assessed using a TaqMan hPSC Scorecard Panel (Thermo Fisher Scientific) as per the manufacturer's instructions.

## RNA-sequencing and analysis

Samples were homogenized in TRIzol, then total RNA was isolated according to the manufacturer's instructions, quantified by Nanodrop (Thermo Fisher Scientific), and diluted to 1 ng/μl in nuclease-free water. A total of 1 ng of RNA was added to 2.5 μl of cell lysis buffer (nuclease-free water with 0.2% v/v Triton X-100 and 0.1 U/μl RNase inhibitor), and subjected to cDNA synthesis and amplification as described before (*Cembrowski et al., 2018*). Libraries were prepared using a modified Nextera XT DNA protocol (Illumina, San Diego, CA) where 5 μM P5NEXTPT5 was substituted for the i5 primers in the kit. Libraries were quantified by qPCR (Roche), normalized to 2 nM, then pooled and sequenced on a NextSeq550 flowcell with 25 bases in read 1, 8 bases in the i7 index read, and 50 bases in read 2. The control library phiX (Illumina) was spiked in at a final concentration of 15% to improve color balance in read 1.

## STARsolo analysis pipeline

Sequencing adapters were trimmed from the reads with Cutadapt v2.10 (*Martin, 2011*) prior to alignment with STAR v2.7.5c (*Dobin et al., 2013*) to the *Homo sapiens* GRCh38.95 genome assembly (Ensembl). Gene counts were generated with the STARsolo algorithm using the following additional parameters: "`--soloType CB_UMI_Simple --soloCBwhitelist smartscrb_whitelist. txt --soloCBstart 2 --soloCBlen 8 --soloUMIstart 10 --soloUMIlen 10 --soloBarcodeReadLength 0 --soloCBmatchWLtype 1 MM_multi_pseudocounts --soloStrand Forward --soloFeatures Gene --soloUMIdedup 1 MM_All --soloUMIfiltering MultiGeneUMI --soloCellFilter None`". The full set of 384 barcodes designed for this assay was used as the whitelist. Gene counts for the subset of barcodes used in this experiment were extracted using custom R scripts.

Differential gene expression analysis was performed using EBseq v3.8 in R (*Leng et al., 2013*), with condition 1 being LRRK2 G2019S and condition 2 being WT. EBseq calculated a median normalization of the sequencing counts using the median of ratios methods (*Anders and Huber, 2010*). First, the geometric mean of the sequencing counts was calculated for each gene and across all samples to create a pseudo-reference sample, and the ratio of the sequencing counts to the pseudo-reference counts was calculated for every gene. Then, for each sample, the median value of all the ratios was taken as the normalization factor. Finally, the median normalized sequencing counts were calculated by dividing the sample's sequencing counts by the sample's normalization factor. A false discovery rate of 0.05 and a fold change threshold of 1.4 or 0.7 were used to identify dysregulated genes in LRRK2 G2019S vs. WT astrocytes. A k-means clustering algorithm (k = 3) was used to group genes

into low, moderate, and highly expressed transcripts based on their $\log_{10}(1 + \text{median normalized counts})$ value, using ComplexHeatmap v3.11 in R (*Gu et al., 2016*). GO enrichment analysis was done using the Database for Annotation, Visualization, and Integrated Discovery (DAVID v6.8, https://david.ncifcrf.gov/) (*Huang et al., 2009a*; *Huang et al., 2009b*). A list of all genes detected in all the samples was exported for use as the background gene set in DAVID.

## Accession numbers

Raw RNA-seq datasets have been deposited in the GEO data repository (NCBI), accession number GSE152768.

## Internalization of CD63[+] exosomes by dopaminergic neurons

The study of the internalization of astrocyte-derived EVs by dopaminergic neurons followed a specific multistep protocol that is detailed below: (1) prepare separate cultures of dopaminergic neurons and astrocytes, (2) transduce each cell type to express a protein of interest (tdTomato or CD63-GFP), (3) harvest the astrocytes and plate them directly on the neurons, and (4) monitor the internalization of EVs by live cell microscopy. As a first step to this protocol, astrocytes were plated at 40,000 cells/cm$^2$ in ScienCell astrocyte medium and transduced the next day with a lentivirus expressing CD63-GFP (produced from pCT-CD63-GFP, System Biosciences, Palo Alto, CA). The astrocytes expressed GFP between 48 and 72 hr after transduction. In parallel, iPSC-derived dopaminergic neurons were plated at 70,000 cells/well in a Geltrex-coated 96-well optically clear plate (Ibidi) in neuron differentiation medium. After 2–3 days in culture, the dopaminergic neurons were transduced with an adeno-associated virus rh10 vector expressing CAG-tdTomato. After 72 hr, the rh10-CAG-tdTomato-containing medium was removed, and 20,000 CD63-GFP astrocytes were plated in each neuron-containing well of the 96-well plate, on top of the tdTomato dopaminergic neurons. To ensure a healthy co-culture, the cells were maintained in NB/B27 with 0.5% FBS for 3 days before imaging. At the time of imaging, the cell culture medium was changed to prewarmed phenol red-free medium to reduce nonspecific background fluorescence. The uptake of astrocyte-secreted CD63-GFP EVs by dopaminergic neurons was monitored for 2 hr by live-cell confocal microscopy using a temperature- and CO$_2$-controlled incubation chamber mounted on a Zeiss LSM 880 inverted laser scanning confocal microscope.

## Confocal and widefield microscopy

The following microscopes were used: (1) Zeiss LSM 880 inverted laser scanning confocal microscope equipped with a plan-apochromatic 63× oil objective lens (Zeiss, NA = 1.4), 40× oil objective lens (Zeiss, NA = 1.3), 405 nm, 488 nm, and 561 nm laser lines, and ZEN software (Zeiss, Oberkochen, Germany). (2) Nikon Eclipse TiE inverted microscope equipped with a CFI Plan Apo Lambda 10× (Nikon Instruments, NA = 0.45) and a CFI Plan Apo VC 60× oil objective lens (Nikon Instruments, NA = 1.4), sCMOS camera (Zyla 4.2, Andor), SPECTRA X light engine illumination source (Lumencor, Beaverton, OR), and NIS-Elements software (Nikon Instruments, Tokyo, Japan). The experimental details for each imaging experiment are provided in the appropriate subsection in Materials and methods.

## Image analysis

### CD63[+] marker distribution in astrocytes

Images in Zeiss CZI file format were properly converted to Imaris native IMS file format and analyzed in Imaris 9.3 (Bitplane, Belfast, UK). Each image was composed of three channels representing the CD44 membrane marker, CD63 exosome/MVB marker, and DAPI nuclear marker. The components in each fluorescence channel were detected using the cell creation wizard without the need for additional image preprocessing. Intensity thresholds were calculated for each channel such that the values were applied to all datasets since they were acquired using identical acquisition settings to minimize user bias; however, for cases in which the cytoplasmic staining was heterogeneous, the intensity thresholds were manually adjusted. For nucleus detection, a size filter was used to remove small and irregular shapes. For cell detection, a surface was generated by expanding it on a single nucleus accompanied by a size filter. For vesicle detection, CD63[+] structures were modeled as spots with an

estimated diameter of 0.8 µm with region growing disabled. Statistics were exported for each cell object, and 'Vesicles Distance to Closest Nucleus' statistics were used to analyze CD63⁺ MVB distribution in astrocytes.

## Colocalization of LRRK2 and CD63

Images in Zeiss CZI file format were properly converted to Imaris native IMS file format and analyzed in Imaris 9.3. All fluorescence channels were first deconvolved using the 'robust' algorithm (15 iterations) in Imaris ClearView, which estimated the PSF based on a widefield fluorescence template and the acquisition parameters used. A surface object was created for the CD63 (exosome/MVB) channel with background subtraction enabled (Gaussian filter width of 1 µm). LRRK2 was detected as spots with an estimated diameter of 0.4 µm and with background subtraction enabled; 'region growing on spots' was used to more accurately determine their intersections with surfaces. Once LRRK2 spots were created, they were segregated into two populations depending on if the distances from the spot centers to the exterior of CD63 surface objects were greater than 0.2 µm (half the estimated diameter); this was accomplished using the 'Find Spots Close To Surface' MATLAB (MathWorks, Natick, MA) algorithm from Imaris XTensions. A cell object was then created by importing CD63 surface objects as cells and LRRK2-associated spots as vesicles. The relevant statistics for each LRRK2 spot, including the cell IDs corresponding to the associated CD63 surface objects, were exported. The data were aggregated in MATLAB to generate histograms.

## Internalization of CD63-GFP EVs by dopaminergic neurons

Confocal z-stack images were properly converted from Zeiss CZI file format to Imaris native IMS file format and analyzed in Imaris 9.3. The green channel with the CD63-GFP signal was first deconvolved (10 iterations) in Imaris ClearView, which estimated the PSF based on a confocal fluorescence template and the acquisition parameters used. The red channel with the tdTomato signal was processed with a Gaussian filter (0.18 µm filter width) to reduce image noise. A total of three layers were created to represent the total neuronal surfaces, soma-only surfaces, and CD63-GFP EVs. Neuronal surfaces encompassing the soma and neurites were modeled using the 'surface' tool with background subtraction and a sphericity filter to remove background signal. The soma layer was modeled using the 'surface' tool with background subtraction enabled, 'split touching objects' enabled, and with seed point diameters of 10 µm to eliminate non-soma surfaces. The resulting surfaces created by Imaris were manually curated if necessary to trim and delete non-soma surfaces and ensure an accurate representation of the somas. CD63-GFP exosomes were detected as spots with an estimated diameter of 0.4 µm and background subtraction enabled. Once CD63 spots were created, they were segregated into two populations (intracellular and extracellular) depending on their distances from the spot centers to the exterior of neuron or soma surface objects; this was accomplished using the 'Find Spots Close To Surface' MATLAB (MathWorks) algorithm from Imaris XTensions.

## Astrocyte effects on dopaminergic neuron viability and morphological changes

Dopaminergic neurons were identified as MAP2⁺/TH⁺ populations, and the number of MAP2⁺/TH⁺ neurons after experimental treatments was counted as a measure of neuronal viability. The length of the dendrites was measured by tracing the MAP2 signal of dopaminergic neurons using the NIS Elements (Nikon Instruments) tracing tool. Sample identity was masked using a numerical code to blind the experimenter during image analysis.

### Statistical analysis

Data was analyzed using GraphPad Prism version 8.0 (La Jolla, CA). p-Values were calculated using two-tailed unpaired Student's t-test assuming equal standard deviation (*Figure 1E,G,H*, *Figure 2B*, *Figure 3C,H*, *Figure 4B,E*, *Figure 6C* , *Figure 7C,D*, *Figure 8B,C,E,H-K*, *Figure 9C*), Mann–Whitney test (*Figure 5B and D*), chi-square test (*Figures 4D and 6B*), or using one-way ANOVA with Newman–Keuls multiple comparisons (*Figure 7F*) or Tukey's multiple comparisons correction (*Figure 9A, E and G*).

## Acknowledgements

We thank the members of the O'Shea lab and our colleagues Dr. Deepika Walpita, Dr. Jennifer Lippincott-Schwartz, Dr. Ulrike Heberlein, and Dr. Erik Snapp for helpful discussions and feedback, and Michelle Quiambao for administrative support. We also thank Dr. Damien Alcor and the Light Microscopy Center at Janelia for assistance setting up the imaging platform, Dr. Zhiheng Yu and the staff at Janelia's Cryo-Electron Microscopy and Electron Microscopy Center, and the staff at Janelia's Virus Services. We thank Prof. Dr. Thomas Gasser (Universitätsklinikum Tübingen) and Prof. Dr. Hans R Schöler (Max-Planck Institute) for providing the iPSCs with the LRRK2 G2019S mutation and isogenic control, and Dr. Randall T Moon (Howard Hughes Medical Institute/University of Washington) for providing the non-isogenic iPSC lines. This work was supported by the Howard Hughes Medical Institute.

## Additional information

### Competing interests

Erin K O'Shea: is Chief Scientific Officer and a Vice President at the Howard Hughes Medical Institute, one of the three founding funders of eLife. The other authors declare that no competing interests exist.

### Funding

No external funding was received for this work.

### Author contributions

Aurelie de Rus Jacquet, Conceptualization, Data curation, Formal analysis, Investigation, Methodology, Supervision, Validation, Visualization, Writing – original draft, Writing – review and editing; Jenna L Tancredi, Formal analysis, Investigation, Writing – review and editing; Andrew L Lemire, Data curation, Formal analysis, Investigation, Methodology, Software, Visualization, Writing – original draft; Michael C DeSantis, Data curation, Formal analysis, Methodology, Software, Validation, Writing – review and editing; Wei-Ping Li, Investigation, Methodology, Writing – original draft; Erin K O'Shea, Conceptualization, Formal analysis, Funding acquisition, Project administration, Supervision, Validation, Writing – review and editing

### Author ORCIDs

Aurelie de Rus Jacquet (iD) http://orcid.org/0000-0002-5548-8045
Andrew L Lemire (iD) http://orcid.org/0000-0002-0624-3789
Michael C DeSantis (iD) http://orcid.org/0000-0002-7214-2740
Erin K O'Shea (iD) http://orcid.org/0000-0002-2649-1018

### Ethics

Patient-derived induced pluripotent stem cells were obtained from RUCDR Infinite Biologics, a University-based biorepository. A second line was provided by Prof. Dr. Thomas Gasser (Universitätsklinikum Tübingen) and Prof. Dr. Hans R. Schöler (Max-Planck Institute), for which informed consent was obtained from all patients prior to cell donation and the Ethics Committee of the Medical Faculty and the University Hospital Tübingen previously approved this consent form (see publication by Reinhardt et al., 2013. Cell Stem Cell, doi 10.1016/j.stem.2013.01.008).
Primary cultures were prepared via dissection of P1 to P3 mouse pups obtained from C57BL/6NTac WT mice or constitutive LRRK2 G2019S knock-in mice generated and maintained on the C57BL/6NTac background (Matikainen-Ankney et al., 2016) (Taconic Biosciences, NY) using methods approved by the Janelia Institutional Animal Care and Use Committee (IACUC protocol #18-168).

### Decision letter and Author response

Decision letter https://doi.org/10.7554/eLife.73062.sa1
Author response https://doi.org/10.7554/eLife.73062.sa2

## Additional files

### Supplementary files
• Supplementary file 1. List of exosome-related genes upregulated in the LRRK2 G2019S astrocytes compared to isogenic controls and identified by RNA-sequencing.

• Supplementary file 2. List of exosome-related genes upregulated in the LRRK2 G2019S astrocytes compared to non-isogenic controls and identified by RNA-sequencing.

• Transparent reporting form

### Data availability
Sequencing data have been deposited in GEO under accession codes GSE152768.

The following dataset was generated:

| Author(s) | Year | Dataset title | Dataset URL | Database and Identifier |
|---|---|---|---|---|
| Jacquet AdR, Tancredi JL, Lemire AL, DeSantis MC, W-P Li, O'Shea EK | 2021 | The LRRK2 G2019S mutation alters astrocyte-to-neuron communication via extracellular vesicles and induces neuron atrophy in a human iPSC-derived model of Parkinson's disease | https://www.ncbi.nlm.nih.gov/geo/query/acc.cgi?acc=GSE152768 | NCBI Gene Expression Omnibus, GSE152768 |

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
