## [Decision Letter]

**Acceptance summary:**

The contribution of glia to the long-term health of neurons is of high interest. Using genetic models of Parkinson's disease (PD), this study shows that PD-related LRRK2 astrocytes derived from patient induced pluripotent stem cells leads to alterations in astrocyte-released extracellular vesicle size, distribution, and content. Moreover, it reveals that these vesicles are taken up by dopaminergic neurons but that they contain diminished neurotrophic support when derived from PD-related LRRK2 astrocytes. Together, this suggests a non-cell autonomous effect on the health of dopaminergic neurons, leading to the selective vulnerable cells that are typically lost in PD.

**Decision letter after peer review:**

[Editors’ note: the authors submitted for reconsideration following the decision after peer review. What follows is the decision letter after the first round of review.]

Thank you for submitting your work entitled "The LRRK2 G2019S mutation alters astrocyte-to-neuron communication via extracellular vesicles and induces neuron atrophy in a human iPSC-derived model of Parkinson's disease" for consideration by *eLife*. Your article has been reviewed by two peer reviewers and a reviewing editor, and the evaluation has been overseen by a Senior Editor. The reviewers have opted to remain anonymous.

Our decision has been reached after consultation between the reviewers. Based on these discussions and the individual reviews below, we regret to inform you that your work will not be considered further for publication in *eLife*.

The discussion between reviewers and editors centered on a few key points. First, all reviewer's felt that it is of utmost important that a justified and appropriate number of hiPSCs and their appropriate controls are utilized throughout. In particular, there is concern that G2019S-related phenotypes may be more variable than other presumed monogenetic causes of disease, for example a low penetrance of disease causation associated with G2019S in people (e.g., 20% lifetime penetrance for PD) that may necessitate more lines analyzed than usual, and possible lines from carriers of the mutation that appear resilient to disease. Studies in the past decade that use only one or a few lines of G2019S hIPSCs have generally failed to replicate in more than one laboratory, possibly due to low power. The reviewer's were not sure how rigorous the study was in this regard. Second, reviewer's felt there was over-interpretation and speculation regarding the possible roles of differential trophic factors released by the astrocytes in EVs and conditioned media without many measures of specific trophic factors, or rescue experiments, to *help* define mechanism. Third, the EV data are not broadly supported by NTA (like Zeta or nanosight) or quantitative measures fairly standard in the EV field. For example, the authors did not clearly quantify the total number of EVs secreted in WT vs. G2019S conditions, which would be a basic experiment needed to create interest in the study in the EV community.

For these reasons, the study, while of interest, is felt too preliminary to consider further revisions.

*Reviewer #1:*

Jacquet et al., The LRRK2 G2019S mutation alters astrocyte-to-neuron communication via extracellular vesicles and induces neuron atrophy in a human iPSC-derived model of Parkinson's disease.

In this manuscript titled "The LRRK2 G2019S mutation alters astrocyte-to-neuron communication via extracellular vesicles and induces neuron atrophy in a human iPSC-derived model of Parkinson's disease", Jacquet and colleagues investigated the role of Parkinsonism gene mutation LRRK2 G2019S in hiPSC-differentiated astrocytes. By isolating extracellular vesicles from ACM and examining astrocytes with various electron microscopy techniques, the authors found that LRRK2 G2019S affects the morphology and distribution of MVBs and the morphology of secreted EVs in hiPSC-differentiated astrocytes. Furthermore, the authors observed that astrocyte-derived EVs can be internalized by dopaminergic neurons and such EVs support neuronal survival. However, LRRK2 G2019S EVs lost the ability of promoting neuronal survival. This is an interesting study showing a non-cell autonomous contribution to dopaminergic neuron loss in PD.

The proposed idea of how LRRK2 G2019S dysregulates EV-mediated astrocyte-to-neuron communication is novel and exciting. However, the authors present some conflicting data that is not addressed during the discussion: they first conclude upregulated exosome biogenesis by RNAseq in G2019S vs WT astrocytes, but later show a decrease in the number of <120nm particles in G2019S mutants suggesting a decrease in the classical exosome-sized vesicle secreted compared to WT. Lastly, their MVB images show less CD63 gold particles in G2019S compared to WT control (though this was not quantified). Do the authors suggest and increase or decrease in exosome biogenesis in G2019S mutants? How do they reconcile these seemingly contradicting data? Several experiments, controls and additional analyses are needed to fully demonstrate the validity of proposed mechanism.

1. In figure 1 A authors demonstrate iPSC-derived astrocytes characterization. Since there is no one unified and validated method for astrocytes differentiation, there is a need for more accurate characterization of iPSC-derived astrocytes. Authors should demonstrate the percentage of cells positive to astrocytic markers and to prove that obtained astrocytes are functional (able to promote synaptogenesis and uptake glutamate). I would also recommend analyzing the iPSC-derived astrocyte cultures for expression of more specific astrocytic markers as GLT1, *SOX9* in addition to those which have been analyzed. Moreover, it is highly important to know what is the proportion of astrocytes derived from LRRK2 G2019S line and its isogenic control in order to be able to compare their effect on neurons.

2. In Figure 1, the authors found a significant upregulation of exosome components in astrocytes, demonstrating an important role of LRRK2 G2019S in EV signaling pathway. In the discussion, the authors briefly mentioned 'sub-populations of CD63- EVs may be differentially secreted in mutant astrocytes'. Since the authors have obtained the RNA-seq data, it would be nice to dig deep into the data and comment on potential EV sub-populations which can be differentially secreted. This information can be very beneficial for follow-up studies in the PD and LRRK2 field. Furthermore, the authors should assess the expression of Rab27a and CD82 in WT and LRRK2 G2019S astrocytes by western blots to verify RT-qPCR data. Furthermore, the authors should present specifically exosome biogenesis or secretion genes are altered to provide further insight into the stage of exosome biogenesis that is affected (ESCRT0-3, VPS4, ALIX, etc).

3. In Figure 2A and B, data shows that both WT and LRRK2 G2019S astrocytes produce MVBs and MVBs in LRRK2 G2019S astrocytes is smaller than in WT astrocytes. In Figure 2E, the authors showed the abundance of CD63 localized within MVBs in WT astrocytes but did not show the CD63 localization in MVBs in G2019S astrocytes. However, it is important to show CD63 localization in MVBs in G2019S astrocytes to fully support the conclusion that CE63+ MVBs are present in LRRK2 G2019S astrocytes. In addition, CD44 is a marker for astrocyte-restricted precursor cells. Although CD44+ positive cells are committed to give rise to astrocytes, it is crucial to include another astrocyte marker to ensure these cells are indeed mature astrocytes.

Related, authors should consider citing some of the MVB maturation literature to guide the readers.

4. In Figure 3, it is impressive that the authors are able to image EVs using cyro-EM approach and analyze their sizes. The authors also observed different shapes of EVs. Is there any shape difference between WT EVs and G2019S EVs? Is there a way that the authors could categorize these shapes and do a detailed analysis in EV shapes? Also, In Figure 3D, both WT EV and G2019S EV images should present side by side for comparison.

. Related, the size frequencies of EVs presented suggest a difference in the types of EV's released. Interestingly, exosomes are classically known to range from ~50-120nm and this population is significantly decreased in G2019S compared to WT. What does this suggest?

5. In figure 3c, SBI ELISA claims to quantify CD63+ vesicles, the authors should present more standardized particle quantification data (either by CD63 FACs for isolated EVs in WT vs G2019S or ZetaView/QNano particle tracking). The authors should also directly quantify the total number of EVs secreted in WT vs G2019S conditions (not only CD63+).

6. In Figure 4, the authors quantify LRRK2+/CD63+ particles by imaging. Importantly, it appears that there are less CD63 "large gold" particles in MVB of G2019S compared to control. This CD63 baseline quantification in MVB of WT vs. G2019S should be presented in this figure. These data are not convincing and should be quantified by FACS in secreted EV. Supplementary figure 3 should be brought into this figure.

7. In Figure 5, using CD63 as a MVB marker is not the most accurate approach. ESCRT markers should be co-stained with these experiments to truly show MVB localization (CD63 can localize to MVBs but is known to have a wider distribution throughout the cell compared to TSG1010 or other ESCRT complex proteins). Additionally, the authors must show their Supplemental Figure 3 ELISA quantification of p-aSyn in this main figure, and comment on why they conclude higher p-aSyn content in MVBs based on their IEM but then find no differences in aSyn in secreted EVs in WT vs. G2019S by ELISA.

8. In figure 6, it is even more clear that there is a stark difference between the CD63 presence in/near MVBs between WT and G2019S conditions. Since the authors normalize several pieces of data to CD63 (MVB localization, LRRK2 co-localization, etc), it is critical to quantify the number of baseline CD63 gold particles in MVBs in WT vs G2019S.

9. In Figure 7, the authors used the co-culture of astrocytes and neurons to assess astrocyte-derived EV uptake by dopaminergic neurons. Although 3D reconstitution of neurons and exosomes can be precise, the data may not be 100% clean. It would be better if the authors collect ACM containing EV fraction from WT astrocyte and G2019S astrocytes and then incubate dopaminergic neurons with ACM containing EV fraction. In this way, only dopaminergic neurons are in the culture and there will be no CD63-GFP expressed astrocytes to contaminate the CD63-GFP signal in neurons.

10. In Figure 9, the authors must show their ACM control. They show untreated, EV-free, and EV-rich ACM, but do not show unmanipulated ACM control.

*Reviewer #2:*

In this manuscript by de Rus Jacquet et al., authors present an interesting study to detect changes in extracellular vesicles in human PD patient derived iPSC-derived astrocytes carrying the LRRK2 G2019S mutation. Isogenic gene corrected iPSCs were used as controls in all experiments. Authors first performed RNA-Seq for global gene expression changes between G2019S and "WT" gene corrected astrocytes. GO analysis showed an upregulation of extracellular compartments (including exosome compartments) in LRRK2 astrocytes. Subsequent experiments focusing on extracellular vesicles (EVs) and multivesicular bodies (MVBs), showed specific differences of MVB area and the size of secreted EVs. Secreted EVs from G2019S astrocytes also contained more LRRK2 particles and G2019S EVs contained more phosphorylated aSyn particles. Co-culture of LRRK2 astrocytes with human dopamine neurons showed accumulation of CD63+ exosomes in neurites, compared to co-culture with WT astrocytes. Co-culture with LRRK2 astrocytes decreased viability of TH^+^ neurons and LRRK2 dendrites/neurites were also shorter. These co-culture findings were replicated using EV-enriched conditioned media. Finally, authors showed that the trophic effect of astrocytes on neurons was due both to soluble factors released into the media, and production and release of EVs. Overall, this is a well-written and systematically performed study. This reviewer has several comments as detailed below.

1) Based on their data, authors conclude that astrocyte-to-neuron signaling and trophic support mediated by EVs is disrupted in LRRK2 G2019S astrocytes. Have authors measured the differences in trophic factors released by LRRK2 astrocytes in EVs and in conditioned media?

2) Authors differentiate cells (astrocytes and neurons) from midbrain lineage NPCs. The data show convincing effects of the LRRK2 derived astrocytes on neurons, but one question is whether this is specific to dopaminergic cells. Would this genotype specific effect also be expected in other lineages, e.g. cortical neurons? Authors should discuss this point.

3) Prior work has demonstrated reductions in neurite length in neurons derived from LRRK2 G2019S iPSCs (not specific to dopaminergic neurons in LRRK2 cells) (for example Reinhard et al. 2013). It is curious that the LRRK2 G2019S mutation itself can cause such a phenotype in neurons mono-cultures, and as shown in the current study, that LRRK2 G2019S astrocytes also induce a similar effect on WT neurons in co-culture. Can authors expand on this point in the Discussion?

4) Authors should provide data on % dopaminergic neurons generated in the cultures.

5) p7. Authors refer to phosphorylated a-synuclein as accelerating PD pathogenesis, but the references cited do not show this. In fact, Gorbatyuk et al. 2008, showed that overexpression of S129 with constitutive phosphorylation eliminated a-synuclein induced nigrostriatal degeneration. The Fujiwara et al. 2002 reference showed the presence of phospho a-syunclein in Lewy bodies and neurites. Authors should revise their statement that phospho a-synuclein is associated with *accelerated* pathology.

6) Please provide details on the number of iPSC lines used for these experiments.

7) Clarify whether the WT neurons used for co-culture were derived from the isogenic human neurons?

---

## [Author Response]

[Editors’ note: the authors resubmitted a revised version of the paper for consideration. What follows is the authors’ response to the first round of review.]

First, all reviewer's felt that it is of utmost important that a justified and appropriate number of hiPSCs and their appropriate controls are utilized throughout. In particular, there is concern that G2019S-related phenotypes may be more variable than other presumed monogenetic causes of disease, for example a low penetrance of disease causation associated with G2019S in people (e.g., 20% lifetime penetrance for PD) that may necessitate more lines analyzed than usual, and possible lines from carriers of the mutation that appear resilient to disease. Studies in the past decade that use only one or a few lines of G2019S hIPSCs have generally failed to replicate in more than one laboratory, possibly due to low power. The reviewer's were not sure how rigorous the study was in this regard.

We appreciate the reviewers’ concerns, and have now collected additional data using (i) a second pair of iPSCs with the LRRK2 G2019S mutation and corresponding age-and sex- matched non-isogenic control, and (ii) primary midbrain cultures isolated from a LRRK2 G2019S mouse model of PD. We demonstrate that experiments performed with the non-isogenic pair and the primary mouse cultures reproduce findings obtained using the isogenic iPSC line. In addition, we note that some of the data on neuron atrophy have been independently described in a recent publication utilizing distinct iPSC lines (di Domenico et al., 2019), supporting our observations. The updated manuscript now contains (i) a second RNA-sequencing (RNA-seq) analysis of astrocytes derived from the non-isogenic pair (Figure 1D,E), (ii) quantification of the distance of CD63^+^ MVBs from the nuclear membrane in WT and LRRK2 G2019S isogenic and non-isogenic astrocytes (Figure 5D,E), (iii) analysis of dendrite length and viability of dopaminergic and nondopaminergic neurons exposed to WT or mutant astrocytes using the second non-isogenic iPSC pair and mouse primary cultures (Figure 8E-K, Supplementary Figure 5), (iv) quantification of dendrite length in mouse primary dopaminergic neurons exposed to conditioned medium prepared from mouse primary WT or mutant astrocytes (Figure 9C,D), and (v) an analysis of down-regulated genes by RNA-seq in both isogenic and non-isogenic astrocyte pairs. We have updated the results and material and methods section to reflect the new data:

Results – Differentiation of LRRK2 G2019S iPSCs and gene-corrected isogenic controls into neural progenitor cells and astrocytes

“We obtained iPSCs reprogrammed from the dermal fibroblasts of two patients with the LRRK2 G2019S mutation, and one line was gene-corrected to produce an isogenic control (Reinhardt et al., 2013) while the second line was paired with iPSCs derived from a sex- and age-matched healthy individual (Supplementary Figure 1A-C). This enables the direct comparison of two independent pairs of iPSCs to take into consideration the inherent variability that exists between different patient-derived cell lines. Of particular importance, one of our disease and control iPSC pair differs only in a single point mutation in LRRK2, which is essential to ensure that the experimental observations are a consequence of the mutation of interest, and do not result from differences in genetic background, or from differences that could arise during fibroblast reprogramming of each non-isogenic pair (Gore et al., 2011; Soldner et al., 2009).”

Results – Expression of exosome components in iPSC-derived astrocytes is altered by the LRRK2 G2019S mutation

“Gene ontology (GO) analysis revealed that components of the extracellular compartment are highly up-regulated, and include the extracellular region, extracellular matrix and extracellular exosomes (Figure 1D,F). The exosome component is one of the most significantly up-regulated GO terms in both isogenic and non-isogenic astrocytes, and is comprised of a total of 67 (isogenic pair) or 95 (non-isogenic pair) genes (Supplementary Files 1 and 2).”

Results – The LRRK2 mutation G2019S alters MVB distribution in iPSC-derived astrocytes The median distance of CD63^+^ MVBs to the nuclear membrane was 9.2 µm in “LRRK2 G2019S astrocytes compared to 4.8 µm in isogenic controls, and 11.5 µm in LRRK2 G2019S astrocytes compared to 7.5 µm in the non-isogenic control. Furthermore, 75 % of the isogenic control CD63^+^ MVBs were localized at a distance to the nucleus smaller than 11.23 µm, compared to 19.34 µm in LRRK2 G2019S astrocytes (Figure 5B-C). In non-isogenic lines, 75 % of the control CD63^+^ MVBs were localized at a distance to the nucleus smaller than 15.82 µm, compared to 23.79 µm in LRRK2 G2019S astrocytes (Figure D-E).”

Results – LRRK2 G2019S iPSC-derived astrocytes induce a non-cell autonomous loss of iPSC-derived dopaminergic neurons

“The neuronal dendrites shortened from an average of 60 µm in WT to 45 µm in LRRK2 G2019S isogenic cultures (Figure 8C). Similar morphological changes were observed when dopaminergic neurons were co-cultured with the second line of LRRK2 G2019S astrocytes compared to nonisogenic controls (Figure 8E-F), and these results are in agreement with a previous report that LRRK2 G2019S astrocytes induce loss of dopaminergic neurons compared to astrocytes derived from a healthy (non-isogenic) donor (di Domenico et al., 2019). To further confirm our results, we measured neuron survival using mouse primary midbrain cultures that consist of purified postmitotic dopaminergic and GABAergic neurons co-cultured with purified astrocytes. A 14-day coculture of WT neurons with LRRK2 G2019S astrocytes resulted in loss of 37 % more of the dopaminergic neurons than co-cultures containing WT astrocytes (Figure 8G, H). Moreover, this neuronal death was selective for dopaminergic neurons (Figure 8I), recapitulating a key feature of PD. LRRK2 G2019S astrocytes also reduced the viability of LRRK2 G2019S dopaminergic neurons (Figure 8J), but did not affect non-dopaminergic neuronal populations (Figure 8K).”

“To determine if these morphological changes are mediated by direct contact between astrocytes and neurons or if they rely solely on astrocyte secreted factors, WT dopaminergic neurons were cultured in ACM prepared from LRRK2 G2019S or isogenic control astrocytes. After 14 days in culture, the average dendrite length of dopaminergic neurons exposed to WT ACM was 64.4 µm, compared to 50.3 µm if the dopaminergic neurons were exposed to basal NB/B27 medium (Figure 9A). This observation suggests that WT ACM, but not basal medium, contains neurotrophic factors necessary for dendrite outgrowth. LRRK2 G2019S ACM mimicked the phenotype observed by direct co-culture, with a 26 % increase in the number of short neurites (0-40 µm), and the absence of long neurites (160-200 µm) compared to neurons exposed to WT ACM (Figure 9B). The average dendrite length of dopaminergic neurons cultured in basal medium or LRRK2 G2019S ACM is equivalent (50.3 µm vs 49.1 µm respectively) (Figure 9A). When mouse primary midbrain neurons are exposed to ACM prepared from LRRK2 G2019S mouse midbrain astrocytes, shortening of dendrite length is also observed in vulnerable dopaminergic neuron populations (Figure 9C,D). In addition, we found that LRRK2 G2019S mouse primary dopaminergic neuron survival is affected by LRRK2 G2019S but not WT ACM (Supplementary Figure 5).”

Material and Methods – Culture of iPSCs

“Patient-derived iPSCs with the LRRK2 mutation G2019S and the corresponding isogenic control were provided by Prof. Dr. Thomas Gasser (Universitätsklinikum Tübingen) and Prof. Dr. Hans R. Schöler (Max-Planck Institute). The second line of patient-derived iPSCs with the LRRK2 mutation G2019S and the sex- and age-matched healthy control were provided by Dr. Randall T. Moon (Howard Hughes Medical Institute/University of Washington). These iPSCs were produced in collaboration with the Tom and Sue Ellison Stem Cell Core (Institute for Stem Cell and Regenerative Medicine, University of Washington) via episomal reprogramming of human primary fibroblasts as described previously (Okita et al., 2011; Yu et al., 2009). On Day 0, 1x106 cells were electroporated with 1.5 µg each of pCXLE-hsk (Addgene plasmid #27078), pCXLE-hul (Addgene plasmid #27080) and pCXLS-hOCT3/4 (Addgene plasmid #27076) episomal vectors (Okita et al., 2011), resuspended in fibroblast media (DMEM high glucose, 10 % FBS, 100 U/ml penicillin/streptomycin) and plated into a gelatin-coated 10 cm dish. On Day 1, the cultures were replenished with fresh fibroblast media and on Day 3 or 4 the transfected cells were passaged onto a gelatin-coated 10 cm dish and maintained in fibroblast media. On Day 7, the cells were passaged at three different densities (1x105, 5x104, 2x104 and 1x104) onto 10 cm dishes preplated with mouse embryonic fibroblasts (MEFs), and subsequently cultured in DMEM/F12 media supplemented with 0.1 mM sodium butyrate, 50 nM SAHA and 15 % KnockOut serum replacement. Media was changed every 2 to 3 days and observed daily to identify iPSC colony formation. Colonies were picked and plated onto 24-well plates pre-plated with MEFs and cultured in DMEM/F12 media supplemented with 20 % KnockOut serum replacement, 1 % sodium pyruvate, 1 % non-essential amino acids, 100 µM 2-Mercaptoethanol, 100 U/ml penicillin/streptomycin and 20 ng/mL FGF2. IPSC clones were subsequently expanded on MEF feeders and subcultured using Dispase (Thermo Fisher Scientific) until the lines demonstrated robust growth (usually three to five passages). The culture medium was then replaced with mTesR1 or mTesR Plus medium (StemCell Technologies, Vancouver, Canada) and MEFs were replaced by a Geltrex basement membrane matrix (Thermo Fisher Scientific). Successful fibroblast reprogramming into iPSCs was validated by confirming expression of pluripotency markers and demonstrating tri-lineage differentiation potential, as described in the Immunofluorescence and RNA isolation and RT-qPCR sections of the Material and Methods.”

Material and Methods – Preparation of mouse primary cultures

“Primary cultures were prepared via dissection of P1 to P3 mouse pups obtained from C57BL/6NTac WT mice or constitutive LRRK2 G2019S knock-in mice generated and maintained on the C57BL/6NTac background (Matikainen-Ankney et al., 2016) (Taconic Biosciences, NY) using methods approved by the Janelia Institutional Animal Care and Use Committee. The mesencephalic region containing the *substantia nigra* and ventral tegmental area was isolated stereoscopically, and the tissue was digested with papain (Worthington Biochemical, NJ) for 30 min at 37 °C, gently triturated to dissociate the cells, filtered with a cell strainer and centrifuged to remove papain and cell debris.

*Astrocyte isolation procedure*. Astrocytes were prepared as described before with some modifications (McCarthy and de Vellis, 1980). The cells were resuspended in Astrocyte Media (ScienCell Research Laboratories) and plated at a density of 70,000 cells/cm^2^ in rat collagencoated flasks. The next day, the old media was removed and replaced with fresh astrocyte media every 3 days until the culture became confluent (5 to 7 days), at which point the cells were frozen, used for experiments, or expanded by passaging once and no more than twice. For neuronastrocyte co-cultures the astrocytes were plated at a density of 9,600 cells/well of a 96-well plate.”

*“Neuron isolation procedure and culture*. The cells were resuspended in buffer (phenol-red free HBSS with calcium and magnesium, 1 % sodium pyruvate, 1 % Glutamax, 1 % HEPES, 0.5 % w/v BSA, 5 % FBS, 100 U/ml penicillin and streptomycin) and neurons were isolated by magneticactivated cell sorting (MACS) using a mouse neuron isolation kit as recommended by the manufacturer (Miltenyi Biotech). Monocultures of neurons were used for ACM experiments, and plated at a density of 45,000 cells/well in Geltrex-coated black 96-well plates with clear bottom. The cultures were maintained in Neurobasal A media supplemented with 2 % B27 without vitamin A, BDNF (20 ng/mL), GDNF (20 ng/mL), TGFb (4 ng/mL), cAMP (500 uM), 0.5 % FBS, 1 % Glutamax, 100 U/ml penicillin and streptomycin. At day 0 of the treatment, the maintenance medium was replaced with 100 % ACM and replenished every 3 to 5 days until analysis. For coculture experiments, neurons were plated directly onto astrocytes at a density of 15,000 to 20,000 neurons/well in a 96-well plate, and cultured in Neurobasal A medium supplemented with 0.5 % FBS, 1 % Glutamax, 100 U/ml penicillin and streptomycin. A 50 % medium change was performed every 3 days to replenish nutrients while ensuring the presence of secreted factors in the culture. After 24 h in culture, isolated or co-cultured neurons were treated with 15 µM 5-Fluoro-2'deoxyuridine for 10 h to prevent growth of non-neuronal cells.”

Material and Methods – Immunofluorescence

“Alternatively, iPSCs were incubated with primary antibodies provided in the StemLight Pluripotency antibody kit (Cell Signaling Technologies, cat. #9656) as per the manufacturer’s instructions.”

Material and Methods – RNA isolation and RT-qPCR

“Tri-lineage differentiation potential of the newly reprogrammed, non-isogenic iPSCs was assessed using a validated protocol based on embryoid body (EB) formation and spontaneous differentiation (ThermoFisher Scientific, Pub. No. MAN0013784). Briefly, iPSCs were dissociated as large aggregates using ReLeSR (StemCell Technologies), resuspended in EB culture medium (DMEM/F12, 20 % KnockOut Serum Replacement, 1 % MEM non-essential amino acids, 55 µm 2-Mercaptoethanol, 100 U/ml penicillin/streptomycin) supplemented with 4 ng/mL FGF2, and plated on non-adherent tissue culture dishes. After 24 h, the growth medium was changed to fresh EB medium without FGF2, and replaced every other day. On Day 10, EBs were dissociated using Accutase, homogenized in Trizol, and mRNA and cDNA were obtained as described above. Expression of endoderm, ectoderm, mesoderm and self-renewal marker genes was assessed using a TaqMan hPSC Scorecard Panel (Thermo Fisher Scientific) as per the manufacturer’s instructions.”

Second, reviewer's felt there was over-interpretation and speculation regarding the possible roles of differential trophic factors released by the astrocytes in EVs and conditioned media without many measures of specific trophic factors, or rescue experiments, to hep define mechanism.

The isolation of specific neurotrophic factors contained in the astrocyte conditioned medium (ACM)/extracellular vesicles (EVs) and rescue experiments are essential to understand the molecular mechanisms involved in the loss of dopaminergic neurons. While this is beyond the scope of our study, the new version of the manuscript now provides a summary of down-regulated genes of interest identified by RNA-seq analysis of LRRK2 G2019S and control astrocytes (new Figure 10). Gene ontology analysis reveals impaired cholesterol and isoprenoid biosynthesis in both our isogenic and non-isogenic lines of iPSC-derived mutant astrocytes, and we include two heatmaps showing the most significantly down-regulated genes associated with lipid biogenesis (Figure 10B) and cell survival (Figure 10C) in LRRK2 G2019S vs WT astrocytes. Identification of an impaired cholesterol pathway in LRRK2 G2019S astrocytes is particularly interesting because astrocytes are the main producers and suppliers of cholesterol in the brain, and cholesterol is not only a signaling factor but also essential for cellular growth and metabolism (Czuba, Steliga, Lietzau, and Kowianski, 2017; Pfrieger and Ungerer, 2011). Therefore, it is reasonable to hypothesize that dopaminergic neurons in our models lack astrocyte-derived cholesterol supplied via EVs or lipoproteins, which in turn limits their growth and survival. The manuscript has been updated to reflect these findings, and the new text reads as follows:

Results section:

“The nature of the secreted factors, either those secreted directly into the media or those enclosed in EVs, could be proteins, small signaling molecules, or essential lipids. A gene ontology analysis of down-regulated genes in LRRK2 G2019S astrocytes revealed a defect in cholesterol as well as isoprenoid biosynthetic processes (Figure 10 A,B) in both isogenic and non-isogenic lines. Furthermore, genes associated with cell growth and survival were significantly downregulated in mutant astrocytes, including *BIRC5*, *FLOT1*, *HTRA1*, PAK1, *KCNMA1* and *LZTS1* (Figure 10C). These genes regulate neuronal growth (*BIRC5*, *FLOT1*, *PAK1*, *LZTS1*) and/or function (*FLOT1*, *PAK1*, *LZTS1*) (Contet, Goulding, Kuljis, and Barth, 2016; Du et al., 2020; Kawaue et al., 2019; Ko et al., 2011; Launay et al., 2008; Pan et al., 2015; Swanwick, Shapiro, Vicini, and Wenthold, 2010), and can be found in EVs (*FLOT1*, *HTRA1*, *KCNMA1*) (Skog et al., 2008), suggesting that they could be shuttled between neurons and astrocytes via this communication pathway.”

Discussion section:

“Neurons rely heavily on astrocytes via mechanisms involving metabolic coupling whereby astrocytes produce and supply a unique combination of cholesterol, cholesterol precursors and fatty acids in the form of lipoproteins (Pfrieger and Ungerer, 2011). […] Collectively, these observations suggest that LRRK2 G2019S astrocytes may fail to deliver important factors to dopaminergic neurons. Further experimentation is needed to test this hypothesis.”

Third, the EV data are not broadly supported by NTA (like Zeta or nanosight) or quantitative measures fairly standard in the EV field. For example, the authors did not clearly quantify the total number of EVs secreted in WT vs. G2019S conditions, which would be a basic experiment needed to create interest in the study in the EV community.

As suggested by the reviewer, we have now included a nanotracking particle analysis (NTA) to complement our cryo-EM approach. We quantified the total number of EVs secreted in WT vs. G2019S astrocytes by NTA and find that there are no significant differences in the number of EVs secreted by WT and LRRK2 G2019S (Figure 3C). The updated text reads as follows:

Results section:

“The astrocyte-derived EV pellet is enriched in exosomes, as demonstrated by the expression of 8 exosomal markers and the absence of cellular contamination (Supplementary Figure 3D). [...] We conclude from these results that the total number and morphology of EVs produced by WT and LRRK2 G2019S astrocytes are similar, but mutant EVs may have a different size distribution compared to WT.”

Material and Methods:

“The ELISA standards provided in the kit are calibrated by NTA to measure the number of exosomes and establish a standard curve based on exosome abundance, and therefore enable an estimation of the number of exosomes in the experimental sample (Figure 3H). Furthermore, the total number of EVs and their size-distribution was quantified by NTA (System Biosciences).”

For these reasons, the study, while of interest, is felt too preliminary to consider further revisions.Reviewer #1:Jacquet et al., The LRRK2 G2019S mutation alters astrocyte-to-neuron communication via extracellular vesicles and induces neuron atrophy in a human iPSC-derived model of Parkinson's disease.In this manuscript titled "The LRRK2 G2019S mutation alters astrocyte-to-neuron communication via extracellular vesicles and induces neuron atrophy in a human iPSC-derived model of Parkinson's disease", Jacquet and colleagues investigated the role of Parkinsonism gene mutation LRRK2 G2019S in hiPSC-differentiated astrocytes. By isolating extracellular vesicles from ACM and examining astrocytes with various electron microscopy techniques, the authors found that LRRK2 G2019S affects the morphology and distribution of MVBs and the morphology of secreted EVs in hiPSC-differentiated astrocytes. Furthermore, the authors observed that astrocyte-derived EVs can be internalized by dopaminergic neurons and such EVs support neuronal survival. However, LRRK2 G2019S EVs lost the ability of promoting neuronal survival. This is an interesting study showing a non-cell autonomous contribution to dopaminergic neuron loss in PD.The proposed idea of how LRRK2 G2019S dysregulates EV-mediated astrocyte-to-neuron communication is novel and exciting. However, the authors present some conflicting data that is not addressed during the discussion: they first conclude upregulated exosome biogenesis by RNAseq in G2019S vs WT astrocytes, but later show a decrease in the number of <120nm particles in G2019S mutants suggesting a decrease in the classical exosome-sized vesicle secreted compared to WT. Lastly, their MVB images show less CD63 gold particles in G2019S compared to WT control (though this was not quantified). Do the authors suggest and increase or decrease in exosome biogenesis in G2019S mutants? How do they reconcile these seemingly contradicting data? Several experiments, controls and additional analyses are needed to fully demonstrate the validity of proposed mechanism.

The RNA-sequencing data of LRRK2 G2019S astrocytes showed an enrichment in genes associated with the “extracellular exosome” gene ontology term but not with the MVB/EV trafficking or secretion pathways. While we found CD82 and Rab27b to be upregulated, the classical biogenesis markers of MVB/EV trafficking and secretion (e.g. VTA1, VPS4, ALIX) were not dysregulated. Instead, the gene list shows an overwhelming dysregulation of genes coding for EV-enclosed proteins which do not have known roles in MVB/EV biogenesis or function (we now discuss this point in the main text). As a result, we do not believe that exosome biogenesis is upregulated but instead propose the working hypothesis that the EV pathway may contribute to LRRK2 G2019S astrocyte dysfunction. To complement the sequencing data, our study provides a characterization of this pathway by (i) describing the cellular distribution of CD63^+^ structures in astrocytes, (ii) measuring the size of secreted EVs, and (iii) analyzing the neurotrophic potential of control and LRRK2 G2019S astrocyte-secreted EVs. We have not characterized the cellular biology of exosome/EV biogenesis in depth, and we do not propose a mechanism by which the LRRK2 G2019S mutation dysregulates these pathways. These questions are beyond the scope of our study, which is focused on the role of astrocytes in neurodegeneration.

In his/her concerns, the reviewer also referred to the CD63 immunogold staining used in Figures 4C and 6A to localize MVBs. After careful quantification of the number of CD63 gold particles in WT and LRRK2 G2019S MVBs, we conclude that there are no differences between the two genotypes and we apologize for selecting non-representative images. We have now replaced these with representative images. Regarding the shift in the size of WT vs. LRRK2 G2019S vesicles, we complemented our cryo-EM analysis with new data generated using Nanoparticle Tracking Analysis (NTA) (Figure 3C,D). The NTA analysis enabled the quantification of a greater number of particles, and we found that both WT and LRRK2 G2019S astrocytes secrete a significant number of particles in the 0-120 nm range. The cryo-EM data suggested that mutant astrocytes secreted fewer particles in this size range, but this is not observed in the NTA analysis. This discrepancy could be explained by the following: (i) in contrast to cryo-EM, NTA does not distinguish EVs from cell debris, which could bias the quantification and increase the number of small particles quantified (Noble et al., 2020), and (ii) studies showed that the size distributions between NTA and cryo-EM differ, the latter enabling the identification of larger particles (Noble et al., 2020). These two techniques are therefore complementary in the study of secreted EV and our manuscript now presents data generated using these two approaches (Figure 3C-G).

Results – Expression of exosome components in iPSC-derived astrocytes is altered by the LRRK2 G2019S mutation

“Gene ontology (GO) analysis revealed that components of the extracellular compartment are upregulated in LRRK2 G2019S astrocytes – these include GO terms corresponding to the extracellular region, extracellular matrix and extracellular exosomes (Figure 1D,F). The exosome component is one of the most significantly up-regulated GO terms in both isogenic and nonisogenic astrocytes, and is comprised of a total of 67 (isogenic pair) or 95 (non-isogenic pair) genes (Supplementary Files 1 and 2). The large majority (~ 98 %) of these gene products are described to be enclosed in exosomes (e.g. CBR1) but do not perform specific functions related to EV formation or secretion. Only a few genes are associated with exosome biogenesis (e.g. CD82) and trafficking (e.g. Rab27b) (Andreu and Yanez-Mo, 2014; Chiasserini et al., 2014; Ostrowski et al., 2010) and we did not detect differences in the expression of canonical factors that regulate MVB formation (e.g. VTA1, VPS4 or ALIX).”

Profiling WT and LRRK2 G2019S EVs secreted by iPSC-derived astrocytes

“The astrocyte-derived EV pellet is enriched in exosomes, as demonstrated by the expression of 8 exosomal markers and the absence of cellular contamination (Supplementary Figure 3D). […] We conclude from these results that the total number and morphology of EVs produced by WT and LRRK2 G2019S astrocytes are similar, but mutant EVs may have a different size distribution compared to WT vesicles.”

1. In figure 1 A authors demonstrate iPSC-derived astrocytes characterization. Since there is no one unified and validated method for astrocytes differentiation, there is a need for more accurate characterization of iPSC-derived astrocytes. Authors should demonstrate the percentage of cells positive to astrocytic markers and to prove that obtained astrocytes are functional (able to promote synaptogenesis and uptake glutamate). I would also recommend analyzing the iPSC-derived astrocyte cultures for expression of more specific astrocytic markers as GLT1, SOX9 in addition to those which have been analyzed. Moreover, it is highly important to know what is the proportion of astrocytes derived from LRRK2 G2019S line and its isogenic control in order to be able to compare their effect on neurons.

We thank the reviewer for these suggestions. It is true that there exist many different astrocyte differentiation protocols, and this study uses a protocol developed by TCW et al. that has been further optimized by our lab to derive astrocytes from a midbrain-patterned population of neural progenitor cells (NPCs) (de Rus Jacquet, 2019; Tcw et al., 2017). The protocol is published, and shows that these astrocytes are functional – they respond to inflammatory factors and alter secretion of the IL-6 cytokine. Furthermore, Supplementary Figure 2D shows a whole transcriptome analysis (by RNA-seq) of the cell populations produced for this study and demonstrates that iPSC-derived astrocytes cluster with human primary midbrain astrocytes and away from iPSCs or NPCs in an unsupervised cluster analysis. However, we agree that in-depth characterization of iPSC-derived astrocytes is essential, and the updated manuscript now shows that (i) the astrocyte differentiation protocol yields 100 % GFAP^+^ cells with both WT and mutant lines (Supplementary Figure 2B), (ii) expression of six astrocyte markers (GLT1, *SOX9*, APOE, BHLHE41, CD44, GLUD1) (Supplementary Figure 2Aii, B), as well as (iii) transient intracellular calcium signaling (Supplementary Figure 2E), and (iv) synaptosome uptake (Supplementary Figure 2F) in both WT and LRRK2 G2019S astrocytes. We also updated the text as follows:

Results section

“Midbrain-patterned NPCs carrying the LRRK2 G2019S mutation or its isogenic control were differentiated into astrocytes as described previously (de Rus Jacquet, 2019; Tcw et al., 2017). As expected, astrocytes expressed the markers GFAP, vimentin, and CD44 as demonstrated by immunofluorescence (Figure 1A) and flow cytometry analyses (Supplementary Figure 2A). Differentiation was equally effective in WT and LRRK2 G2019S cells, with 100 % of the differentiated astrocytes expressing GFAP (Supplementary Figure 2Bi). To further demonstrate the successful differentiation of iPSCs into astrocytes, we analyzed gene expression using RNAsequencing analysis (RNA-seq), including primary human midbrain astrocyte samples in the RNA-seq study to serve as a positive control for human astrocyte identity. iPSC-derived and human midbrain astrocytes expressed similar levels of genes markers of astrocyte identity, including *SOX9* and GLUT1 (Supplementary Figure 2B). In addition, principal component and unsupervised cluster analyses separated undifferentiated iPSCs, iPSC-derived NPCs and iPSCderived astrocytes into independent clusters, demonstrating that our differentiation strategy produces distinct cell types (Supplementary Figure 2C-D). Importantly, the transcriptome of iPSCderived astrocytes showed more similarities to fetal human midbrain astrocytes than to NPCs or iPSCs, further validating their astrocyte identity (Supplementary Figure 2D). Lastly, control and LRRK2 G2019S astrocytes showed classic astrocytic functional phenotypes such as spontaneous and transient calcium signaling and synaptosome uptake (Supplementary Figure 2E-F).”

2. In Figure 1, the authors found a significant upregulation of exosome components in astrocytes, demonstrating an important role of LRRK2 G2019S in EV signaling pathway. In the discussion, the authors briefly mentioned 'sub-populations of CD63- EVs may be differentially secreted in mutant astrocytes'. Since the authors have obtained the RNA-seq data, it would be nice to dig deep into the data and comment on potential EV sub-populations which can be differentially secreted. This information can be very beneficial for follow-up studies in the PD and LRRK2 field. Furthermore, the authors should assess the expression of Rab27a and CD82 in WT and LRRK2 G2019S astrocytes by western blots to verify RT-qPCR data. Furthermore, the authors should present specifically exosome biogenesis or secretion genes are altered to provide further insight into the stage of exosome biogenesis that is affected (ESCRT0-3, VPS4, ALIX, etc).

In the first comment, the reviewer refers to the observation that the number of total and CD63positive EVs secreted by astrocytes is unchanged between the WT and LRRK2 G2019S genotypes. The classification of different EV sub-populations based on marker proteins is an evolving field of research, and an important study by Kowal et al. defined generic and subpopulation-specific EV markers (Kowal et al., 2016). Our RNA-seq dataset revealed five upregulated genes identified in the Kowal study, namely actin, GAPDH, actinin, complement and fibronectin, but unfortunately there is no clear pattern correlated with specific EV sub-populations. For example, actin and GAPDH are two upregulated proteins that can be found in multiple types of EVs, actinin is enriched in large and medium-sized EVs, and complement and fibronectin are enriched in high density but small EVs (Kowal et al., 2016). The majority of dysregulated genes identified in our sequencing experiment are not proteins classically used to categorize EVs, so unfortunately our data does not allow us to address the reviewer’s question. To make sure that the data is readily accessible to the scientific community, we have prepared a supplementary table with a list of extracellular exosome-related genes identified in the RNA-sequencing study. To respond to the reviewer’s comment on a specific stage of EV biogenesis/secretion altered in LRRK2 G2019S, the sequencing data presented in this manuscript does not allow to conclude that there is such a dysregulation. Our gene list corresponding to the “extracellular exosome” gene ontology term contains a large majority of genes coding for proteins enclosed within EVs that do not play a role in biogenesis/secretion. For example, the gene list does not contain *ESCRT0-3, VPS4, ALIX* or other classical markers involved in EV biogenesis and we cannot conclude anything about the alteration of MVB/EV biogenesis or defects in specific stages of MVB trafficking or EV secretion. In addition, we thank the reviewer for suggesting the validation of RTqPCR data by western blot. The purpose of the RT-qPCR experiment was to validate the gene expression data collected by RNA-seq. Given that our objective was to confirm gene expression levels, and that we do not further study CD82 and Rab27b, we think that collecting protein expression levels is not necessary in the context of this study.

3. In Figure 2A and B, data shows that both WT and LRRK2 G2019S astrocytes produce MVBs and MVBs in LRRK2 G2019S astrocytes is smaller than in WT astrocytes. In Figure 2E, the authors showed the abundance of CD63 localized within MVBs in WT astrocytes but did not show the CD63 localization in MVBs in G2019S astrocytes. However, it is important to show CD63 localization in MVBs in G2019S astrocytes to fully support the conclusion that CD63+ MVBs are present in LRRK2 G2019S astrocytes. In addition, CD44 is a marker for astrocyte-restricted precursor cells. Although CD44+ positive cells are committed to give rise to astrocytes, it is crucial to include another astrocyte marker to ensure these cells are indeed mature astrocytes.Related, authors should consider citing some of the MVB maturation literature to guide the readers.

We agree with the reviewer’s suggestion, and we added images showing the subcellular localization of CD63 in both WT and LRRK2 G2019S MVBs by immunogold staining (Figure 2E). Validation experiments available in Figure 1A and Supplementary Figure 2A and 2B confirm that our astrocytes express CD44 as well as markers of mature astrocytes (BHLHE41, *SOX9*, GLUT1, APOE, GLUD1). The reason for showing CD44 instead of a more mature marker such as GFAP in Figure 2 of the manuscript is because CD44 is a membrane marker, and it therefore enables a clear visualization of the astrocyte surface area. We also note that, as shown in Supplementary Figure 2Aii, iPSC-derived and human fetal astrocytes express CD44, but iPSCs and NPCs do not significantly express this marker gene. In addition, as suggested by the reviewer, we added information related to MVB maturation in the introduction and the new text reads as follows:

Introduction section

“The sorting and loading of exosome cargo is an active and regulated process (Temoche-Diaz et al., 2019), and the regulatory factors involved in EV/exosome biogenesis are just beginning to be identified. Among the well-known factors, Rab proteins are essential mediators of MVB trafficking and they regulate endosomal MVB formation/maturation as well as microvesicle budding directly from the plasma membrane (Pegtel and Gould, 2019; T. Wang et al., 2014). In addition, membrane remodeling is an essential aspect of MVB/EV formation that appears to be regulated, at least in part, by the endosomal sorting complex required for transport (ESCRT) machinery (Pegtel and Gould, 2019; Schoneberg, Lee, Iwasa, and Hurley, 2017).”

4. In Figure 3, it is impressive that the authors are able to image EVs using cyro-EM approach and analyze their sizes. The authors also observed different shapes of EVs. Is there any shape difference between WT EVs and G2019S EVs? Is there a way that the authors could categorize these shapes and do a detailed analysis in EV shapes? Also, In Figure 3D, both WT EV and G2019S EV images should present side by side for comparison.Related, the size frequencies of EVs presented suggest a difference in the types of EV's released. Interestingly, exosomes are classically known to range from ~50-120nm and this population is significantly decreased in G2019S compared to WT. What does this suggest?

As suggested by the reviewer, we classified the two main EV shapes as “simple” and “multiple” EVs, and found no quantitative differences between WT and LRRK2 G2019S. This new data and side-by-side images of WT and LRRK2 G2019S EV images are available in Figure 3EG, and the text has been updated accordingly. One of the observations of Figure 3 is that there exist genotype-specific differences in the size distribution of EVs, which suggests that different classes of vesicles may be preferably produced by WT vs. LRRK2 G2019S astrocytes. This could be the result of differences in dynamics related to cargo loading, or a shift from MVB-released exosomes to membrane budding and microvesicle production. These observations are of great interest and we added a short discussion but they are beyond the scope of this study focused on EV neurotrophic properties, and we do not currently have evidence to support these hypotheses.

Results – LRRK2 G2019S affects the size of EVs secreted by iPSC-derived astrocytes

“EVs mostly displayed a circular morphology (as opposed to the cup-shaped morphology observed by TEM) (Figure 3E), but a variety of other shapes were also observed (Supplementary Figure 3C). (…) Quantification of the number of simple vs. multiple EV structures did not differ between the two lines, and represent up to 16 % of the EV population (Figure 3G).”

Discussion – Dysregulation of iPSC-derived astrocyte-mediated EV biogenesis in Parkinson’s disease

“The observation that LRRK2 G2019S MVBs are less frequently located in the perinuclear area suggests that they may spend less time loading cargo at the Trans-Golgi network, which could in turn produce smaller MVBs and EVs with a different size range compared to WT (Edgar, Eden, and Futter, 2014; Pegtel and Gould, 2019). We did not observe a difference in the number of secreted EVs (total and CD63^+^ subpopulation) between WT and LRRK2 G2019S astrocytes (Figure 3C,H), suggesting that the secretion of at least one population of EVs is independent of the astrocyte genotype.”

5. In figure 3c, SBI ELISA claims to quantify CD63+ vesicles, the authors should present more standardized particle quantification data (either by CD63 FACs for isolated EVs in WT vs G2019S or ZetaView/QNano particle tracking). The authors should also directly quantify the total number of EVs secreted in WT vs G2019S conditions (not only CD63+).

The updated manuscript now contains the NTA analysis of WT and LRRK2 G2019S EVs (Figure 3C,D) which provides the total number of EVs secreted by WT and LRRK2 G2019S astrocytes. We refer the reviewer to General comments, Comment no 3 for a more detailed response.

6. In Figure 4, the authors quantify LRRK2+/CD63+ particles by imaging. Importantly, it appears that there are less CD63 "large gold" particles in MVB of G2019S compared to control. This CD63 baseline quantification in MVB of WT vs. G2019S should be presented in this figure. These data are not convincing and should be quantified by FACS in secreted EV. Supplementary figure 3 should be brought into this figure.

As suggested by the reviewer, we quantified the number of CD63 large gold particles per MVB in WT and LRRK2 G2019S lines (Supplementary Figure 3A,B), and we re-introduced Supplementary Figure 3 into the main text (Figure 4E). We also updated the text. Additionally, we present extensive quantification of LRRK2 levels in MVBs and secreted EVs via imaging and biochemical analysis (ELISA), two different but complementary analytical methods.

Results – LRRK2 G2019S affects the size of MVBs in iPSC-derived astrocytes

“Tetraspanins are transmembrane proteins, and the tetraspanin CD63 is enriched in exosomes and widely used as an exosomal marker (Escola et al., 1998; Men et al., 2019). However, celltype specificities in the expression of exosomal markers such as CD63 have been documented (Jorgensen et al., 2013; Yoshioka et al., 2013). We therefore confirmed the presence of CD63positive MVBs in iPSC-derived isogenic astrocytes by immunofluorescence (Figure 2D) and immunogold electron microscopy (IEM) (Figure 2E). Analysis of IEM images showed an abundance and similar levels of CD63 localized within MVBs in WT and LRRK2 G2019S astrocytes (Figure 2E, Supplementary Figure 3A,B), confirming that CD63 can be used as a marker of MVBs and exosomes in iPSC-derived astrocytes.”

7. In Figure 5, using CD63 as a MVB marker is not the most accurate approach. ESCRT markers should be co-stained with these experiments to truly show MVB localization (CD63 can localize to MVBs but is known to have a wider distribution throughout the cell compared to TSG1010 or other ESCRT complex proteins). Additionally, the authors must show their Supplemental Figure 3 ELISA quantification of p-aSyn in this main figure, and comment on why they conclude higher p-aSyn content in MVBs based on their IEM but then find no differences in aSyn in secreted EVs in WT vs. G2019S by ELISA.

We thank the reviewer for the suggestion to use ESCRT proteins as MVB markers. We decided to use CD63 because it is recognized in the literature as an MVB and EV marker (Beatty, 2008; Edgar et al., 2014), and we now refer to these two studies in the manuscript to support this choice. Using ESCRT complex proteins as MVB markers is an interesting alternative, but we note that proteins associated with this complex are also found to regulate other biological processes such as autophagy (Takahashi et al., 2018) and plasma membrane repair (Jimenez et al., 2014), and so they can co-localize to non-MVB structures (e.g. autophagosomes or plasma membrane). Similarly, TSG101 can also localize to non-MVB structures such as the nucleus and Golgi complex (Xie, Li, and Cohen, 1998), and also lipid droplet (LD) membranes where it promotes LD-mitochondria contact (J. Wang et al., 2021). As suggested by the reviewer, Supplemental Figure 3 has been re-introduced into the main text (Figure 6C). Regarding αSyn, the immunogold staining specifically detects the phosphorylated form of αSyn (p-αSyn), while the ELISA detects all forms of αSyn (total αSyn). We observed increased p-αSyn in LRRK2 G2019S MVBs, but similar levels of total αSyn in WT vs LRRK2 G2019S EVs. This observation suggests that the phosphorylated form of αSyn, but not the total amount of αSyn, is affected by the experimental conditions. The text has been updated and reads as follows.

Results – LRRK2 is associated with MVBs and EVs in iPSC-derived astrocytes

“In light of our observations that mutations in LRRK2 result in altered astrocytic MVB and EV phenotypes, we asked if LRRK2 is directly associated with MVBs in astrocytes and if this association is altered by the LRRK2 G2019S mutation. We analyzed and quantified the colocalization of LRRK2 with CD63 (Figure 4A), a marker for MVBs (Beatty, 2008; Edgar et al., 2014), and found that the proportion of LRRK2^+^/CD63^+^ structures remains unchanged between WT and LRRK2 G2019S isogenic astrocytes (Figure 4B).”

Results – The LRRK2 G2019S mutation increases the amount of phosphorylated α synuclein (Ser129) in MVBs

“Since the MVB/EV secretion pathway is altered in our LRRK2 G2019S model of PD, we reasoned that mutant astrocytes might produce αSyn-enriched EVs by accumulating the protein in its native or phosphorylated form in MVBs or EVs. IEM analysis revealed an abundance of p-αSyn (small gold) inside and in the vicinity of MVBs of LRRK2 G2019S iPSC-derived astrocytes, but not isogenic control astrocytes (Figure 6A). We observed that 55 % of the CD63^+^ (large gold) MVBs in LRRK2 G2019S astrocytes are also p-αSyn^+^ (small gold), compared to only 16 % in WT MVBs. LRRK2 G2019S astrocytes contained on average 1.3 p-αSyn small gold particles per MVB compared to only 0.16 small gold particles in isogenic control astrocytes, and MVB populations containing more than 3 p-αSyn small gold particles were only observed in LRRK2 G2019S astrocytes (Figure 6B). When we analyzed the content of EVs by ELISA, we found that total αSyn levels (phosphorylated and non-phosphorylated) in EV-enriched fractions are similar between isogenic controls and LRRK2 G2019S (Figure 6C). These results suggest that astrocytes secrete αSyn-containing EVs, and the LRRK2 G2019S mutation appears to alter the ratio of p-αSyn/total αSyn in MVB-related astrocyte secretory pathways.”

8. In figure 6, it is even more clear that there is a stark difference between the CD63 presence in/near MVBs between WT and G2019S conditions. Since the authors normalize several pieces of data to CD63 (MVB localization, LRRK2 co-localization, etc), it is critical to quantify the number of baseline CD63 gold particles in MVBs in WT vs G2019S.

After careful quantification of the number of CD63 gold particles in WT and LRRK2 G2019S MVBs (available in Supplementary Figure 3A,B), we conclude that there are no significant differences between the two genotypes, and the MVB images initially selected in Figure 6 are not representative. We therefore replaced Figure 6A with new images.

9. In Figure 7, the authors used the co-culture of astrocytes and neurons to assess astrocyte-derived EV uptake by dopaminergic neurons. Although 3D reconstitution of neurons and exosomes can be precise, the data may not be 100% clean. It would be better if the authors collect ACM containing EV fraction from WT astrocyte and G2019S astrocytes and then incubate dopaminergic neurons with ACM containing EV fraction. In this way, only dopaminergic neurons are in the culture and there will be no CD63-GFP expressed astrocytes to contaminate the CD63-GFP signal in neurons.

We understand the concerns raised by the reviewer, and we can ensure that state-of-the-art imaging technologies and image post-processing techniques have been used to prevent astrocytic CD63 signal from contaminating the neuronal signal. We performed confocal microscopy with a 63X oil objective lens (numerical aperture = 1.4), and the images were processed with a Gaussian Filter (0.18 μm filter width) to reduce background noise in the MAP2 channel, and deconvolved (10 iteration) to enhance confocal image resolution in the CD63 channel. Furthermore, CD63-positive structures were detected with background subtraction enabled.

10. In Figure 9, the authors must show their ACM control. They show untreated, EV-free, and EV-rich ACM, but do not show unmanipulated ACM control.

The results of dendrite length analysis for unmanipulated ACM was initially available in Figures 8E and 8F. For clarity, we prepared a new Figure 9 that shows treatment with unmanipulated ACM, EV-free ACM, and EV-enriched fractions.

Reviewer #2:In this manuscript by de Rus Jacquet et al., authors present an interesting study to detect changes in extracellular vesicles in human PD patient derived iPSC-derived astrocytes carrying the LRRK2 G2019S mutation. Isogenic gene corrected iPSCs were used as controls in all experiments. Authors first performed RNA-Seq for global gene expression changes between G2019S and "WT" gene corrected astrocytes. GO analysis showed an upregulation of extracellular compartments (including exosome compartments) in LRRK2 astrocytes. Subsequent experiments focusing on extracellular vesicles (EVs) and multivesicular bodies (MVBs), showed specific differences of MVB area and the size of secreted EVs. Secreted EVs from G2019S astrocytes also contained more LRRK2 particles and G2019S EVs contained more phosphorylated aSyn particles. Co-culture of LRRK2 astrocytes with human dopamine neurons showed accumulation of CD63+ exosomes in neurites, compared to co-culture with WT astrocytes. Co-culture with LRRK2 astrocytes decreased viability of TH^+^ neurons and LRRK2 dendrites/neurites were also shorter. These co-culture findings were replicated using EV-enriched conditioned media. Finally, authors showed that the trophic effect of astrocytes on neurons was due both to soluble factors released into the media, and production and release of EVs. Overall, this is a well-written and systematically performed study. This reviewer has several comments as detailed below.1) Based on their data, authors conclude that astrocyte-to-neuron signaling and trophic support mediated by EVs is disrupted in LRRK2 G2019S astrocytes. Have authors measured the differences in trophic factors released by LRRK2 astrocytes in EVs and in conditioned media?

This is an important question, and we have not measured the levels of various neurotrophic factors in the medium. We concluded that LRRK2 G2019S astrocytes failed to secrete neurotrophic factors based on the neuron viability data. Healthy neurons cultured with disease astrocytes displayed dendrite shortening equivalent to that of neurons cultured in basal medium lacking neurotrophic factors. Furthermore, the morphological alterations occurred over a long period of time (2 weeks) and did not recapitulate the rapid and high level of neuron death and neurite fragmentation typically observed as a result of exposure to neurotoxins (Liddelow et al., 2017). However, we performed a new analysis of our RNA-seq data and identified dysregulated trophic processes of interest in LRRK2 G2019S astrocytes. We would like to refer the reviewer to General comments, comment no 2 for more detailed explanation of the new data and accompanying text.

2) Authors differentiate cells (astrocytes and neurons) from midbrain lineage NPCs. The data show convincing effects of the LRRK2 derived astrocytes on neurons, but one question is whether this is specific to dopaminergic cells. Would this genotype specific effect also be expected in other lineages, e.g. cortical neurons? Authors should discuss this point.

The reviewer is making an excellent point. We prepared mouse primary midbrain cultures, and co-cultured WT midbrain neurons with WT or LRRK2 G2019S astrocytes. We found that the survival of WT midbrain dopaminergic neurons was significantly affected by LRRK2 G2019S astrocytes, but the viability of non-dopaminergic midbrain neurons was not changed when cocultured with WT or disease astrocytes. A previous study by di Domenico et al. also showed that dopaminergic neurons are more sensitive to the effect of LRRK2 G2019S astrocytes compared to non-dopaminergic cell types (di Domenico et al., 2019). We refer the reviewer to our response to reviewer 1, comment no 13 more detailed explanations of the new data and accompanying text.

3) Prior work has demonstrated reductions in neurite length in neurons derived from LRRK2 G2019S iPSCs (not specific to dopaminergic neurons in LRRK2 cells) (for example Reinhard et al. 2013). It is curious that the LRRK2 G2019S mutation itself can cause such a phenotype in neurons mono-cultures, and as shown in the current study, that LRRK2 G2019S astrocytes also induce a similar effect on WT neurons in co-culture. Can authors expand on this point in the Discussion?

We thank the reviewer for this question, and we added a new point of discussion in our manuscript, which reads as follows:

“Evidence from this study and previous reports indicates that the LRRK2 G2019S mutation affects neurons through a variety of mechanisms. Here, we show a non-cell autonomous effect on neuronal viability via impairment of essential astrocyte-to-neuron trophic signaling, but the LRRK2 G2019S mutation can also mediate cell-autonomous dopaminergic neurodegeneration (Reinhardt et al., 2013). These observations support the idea that the LRRK2 kinase may be involved in a large number of pathways essential to maintain cellular function, cell-cell communication and brain homeostasis, and disruption of LRRK2 in one cell type has cascading effects on other neighboring cell types. In conclusion, our study suggests a novel effect of the PDrelated mutation LRRK2 G2019S in astrocytes, and in their ability to support dopaminergic neurons. This study supports a model of astrocyte-to-neuron signaling and trophic support mediated by EVs, and dysregulation of this pathway contributes to LRRK2 G2019S astrocytemediated dopaminergic neuron atrophy.”

4) Authors should provide data on % dopaminergic neurons generated in the cultures.

We agree that this is important information, and we updated the latest version of the manuscript with this information. We estimate that the neuron cultures consist of 50 to 70 % dopaminergic neurons, and they are depleted of non-neuronal cells as explained in Material and Methods.

Material and Methods – Preparation and culture of iPSC-derived NPCs, dopaminergic neurons and astrocytes

“To isolate a pure neuronal population, the cells were harvested in Accumax medium, diluted to a density of 1 × 10^6^ cells in 100 µl MACS buffer (HBSS, 1 % v/v sodium pyruvate, 1 % GlutaMAX, 100 U/ml penicillin/streptomycin, 1 % HEPES, 0.5 % bovine serum albumin) supplemented with CD133 antibody (5 % v/v, BD Biosciences, San Jose, CA, cat. # 566596), and the CD133^+^ NPCs were depleted by magnetic-activated cell sorting (MACS) using an LD depletion column (Miltenyi Biotech, San Diego, CA), as described previously (de Rus Jacquet, 2019). The final cultures are depleted of non-neuronal cells and contain approximately 70 % dopaminergic neurons, the remaining neurons consisting of uncharacterized non-dopaminergic populations.”

5) p7. Authors refer to phosphorylated a-synuclein as accelerating PD pathogenesis, but the references cited do not show this. In fact, Gorbatyuk et al. 2008, showed that overexpression of S129 with constitutive phosphorylation eliminated a-synuclein induced nigrostriatal degeneration. The Fujiwara et al. 2002 reference showed the presence of phospho a-syunclein in Lewy bodies and neurites. Authors should revise their statement that phospho a-synuclein is associated with *accelerated* pathology.

The reviewer is correct. We meant to highlight that there is a correlation between phosphorylated αSyn levels and PD pathogenesis, not that phosphorylated αSyn causes an acceleration of PD pathogenesis. We rephrased the sentence as follows, and replaced the study by Gorbatyuk et al. with a study by Anderson et al. that shows presence of phosphorylated αSyn in Lewy bodies:

“EVs isolated from the biofluids of PD patients exhibit accumulation of αSyn (Lamontagne-Proulx et al., 2019; Shi et al., 2014; Zhao et al., 2018), a hallmark protein whose phosphorylation at the serine residue 129 (p-αSyn) is correlated with PD pathogenesis (Anderson et al., 2006; Fujiwara et al., 2002).”

6) Please provide details on the number of iPSC lines used for these experiments.

Experiments in the first version of this manuscript were performed using a single LRRK2 G2019S iPSC line and its gene-corrected control. The manuscript now presents the results collected using a second, independent non-isogenic iPSC line, as well as mouse primary cultures. We would like to refer the reviewer to General comments, comment no 1. for detail explanations of the new models and data collected.

7) Clarify whether the WT neurons used for co-culture were derived from the isogenic human neurons?

We confirm that the WT neurons used for co-culture experiments were derived from isogenic controls. We added subtitles to our figures to clarify when data show results from isogenic or nonisogenic iPSC-derived cells.

References

Anderson, J. P., Walker, D. E., Goldstein, J. M., de Laat, R., Banducci, K., Caccavello, R. J.,... Chilcote, T. J. (2006). Phosphorylation of Ser-129 is the dominant pathological modification of α-synuclein in familial and sporadic Lewy body disease. *J Biol Chem, 281*(40), 29739-29752. doi:10.1074/jbc.M600933200

Andreu, Z., and Yanez-Mo, M. (2014). Tetraspanins in extracellular vesicle formation and function.

*Front Immunol, 5*, 442. doi:10.3389/fimmu.2014.00442

Beatty, W. L. (2008). Late endocytic multivesicular bodies intersect the chlamydial inclusion in the absence of CD63. *Infect Immun, 76*(7), 2872-2881. doi:10.1128/IAI.00129-08

Chiasserini, D., van Weering, J. R., Piersma, S. R., Pham, T. V., Malekzadeh, A., Teunissen, C. E.,... Jimenez, C. R. (2014). Proteomic analysis of cerebrospinal fluid extracellular vesicles: a comprehensive dataset. *J Proteomics, 106*, 191-204. doi:10.1016/j.jprot.2014.04.028

Contet, C., Goulding, S. P., Kuljis, D. A., and Barth, A. L. (2016). BK Channels in the Central Nervous System. *Int Rev Neurobiol, 128*, 281-342. doi:10.1016/bs.irn.2016.04.001

Cremona, M. L., Matthies, H. J., Pau, K., Bowton, E., Speed, N., Lute, B. J.,... Yamamoto, A. (2011). Flotillin-1 is essential for PKC-triggered endocytosis and membrane microdomain localization of DAT. *Nature Neuroscience, 14*(4), 469-477. doi:10.1038/nn.2781

Czuba, E., Steliga, A., Lietzau, G., and Kowianski, P. (2017). Cholesterol as a modifying agent of the neurovascular unit structure and function under physiological and pathological conditions. *Metab Brain Dis, 32*(4), 935-948. doi:10.1007/s11011-017-0015-3

de Rus Jacquet, A. (2019). Preparation and Co-Culture of iPSC-Derived Dopaminergic Neurons and Astrocytes. *Curr Protoc Cell Biol, 85*(1), e98. doi:10.1002/cpcb.98

di Domenico, A., Carola, G., Calatayud, C., Pons-Espinal, M., Munoz, J. P., Richaud-Patin, Y.,.

.. Consiglio, A. (2019). Patient-Specific iPSC-Derived Astrocytes Contribute to Non-CellAutonomous Neurodegeneration in Parkinson's Disease. *Stem Cell Reports, 12*(2), 213229. doi:10.1016/j.stemcr.2018.12.011

Du, X., Carvalho-de-Souza, J. L., Wei, C., Carrasquel-Ursulaez, W., Lorenzo, Y., Gonzalez, N.,... Gomez, C. M. (2020). Loss-of-function BK channel mutation causes impaired mitochondria and progressive cerebellar ataxia. *Proc Natl Acad Sci U S A, 117*(11), 60236034. doi:10.1073/pnas.1920008117

Edgar, J. R., Eden, E. R., and Futter, C. E. (2014). Hrs- and CD63-dependent competing mechanisms make different sized endosomal intraluminal vesicles. *Traffic, 15*(2), 197211. doi:10.1111/tra.12139

Escola, J. M., Kleijmeer, M. J., Stoorvogel, W., Griffith, J. M., Yoshie, O., and Geuze, H. J. (1998). Selective enrichment of tetraspan proteins on the internal vesicles of multivesicular endosomes and on exosomes secreted by human B-lymphocytes. *J Biol Chem, 273*(32), 20121-20127. doi:10.1074/jbc.273.32.20121

Fujiwara, H., Hasegawa, M., Dohmae, N., Kawashima, A., Masliah, E., Goldberg, M. S.,... Iwatsubo, T. (2002). α-Synuclein is phosphorylated in synucleinopathy lesions. *Nat Cell Biol, 4*(2), 160-164. doi:10.1038/ncb748

Gore, A., Li, Z., Fung, H. L., Young, J. E., Agarwal, S., Antosiewicz-Bourget, J.,... Zhang, K.

(2011). Somatic coding mutations in human induced pluripotent stem cells. *Nature, 471*(7336), 63-67. doi:10.1038/nature09805

Huang, X., Chen, H., Miller, W. C., Mailman, R. B., Woodard, J. L., Chen, P. C.,... Poole, C. (2007). Lower low-density lipoprotein cholesterol levels are associated with Parkinson's disease. *Mov Disord, 22*(3), 377-381. doi:10.1002/mds.21290

Jimenez, A. J., Maiuri, P., Lafaurie-Janvore, J., Divoux, S., Piel, M., and Perez, F. (2014). ESCRT machinery is required for plasma membrane repair. *Science, 343*(6174), 1247136.

doi:10.1126/science.1247136

Jin, U., Park, S. J., and Park, S. M. (2019). Cholesterol Metabolism in the Brain and Its Association with Parkinson's Disease. *Exp Neurobiol, 28*(5), 554-567. doi:10.5607/en.2019.28.5.554

Jorgensen, M., Baek, R., Pedersen, S., Sondergaard, E. K., Kristensen, S. R., and Varming, K. (2013). Extracellular Vesicle (EV) Array: microarray capturing of exosomes and other extracellular vesicles for multiplexed phenotyping. *J Extracell Vesicles, 2*. doi:10.3402/jev.v2i0.20920

Kawaue, T., Shitamukai, A., Nagasaka, A., Tsunekawa, Y., Shinoda, T., Saito, K.,... Kawaguchi, A. (2019). Lzts1 controls both neuronal delamination and outer radial glial-like cell generation during mammalian cerebral development. *Nat Commun, 10*(1), 2780. doi:10.1038/s41467-019-10730-y

Kim, J. M., Cha, S. H., Choi, Y. R., Jou, I., Joe, E. H., and Park, S. M. (2016). DJ-1 deficiency impairs glutamate uptake into astrocytes via the regulation of flotillin-1 and caveolin-1 expression. *Sci Rep, 6*, 28823. doi:10.1038/srep28823

Ko, C. Y., Tsai, M. Y., Tseng, W. F., Cheng, C. H., Huang, C. R., Wu, J. S.,... Hu, C. H. (2011). Integration of CNS survival and differentiation by HIF2alpha. *Cell Death Differ, 18*(11), 1757-1770. doi:10.1038/cdd.2011.44

Kobuna, H., Inoue, T., Shibata, M., Gengyo-Ando, K., Yamamoto, A., Mitani, S., and Arai, H. (2010). Multivesicular body formation requires OSBP-related proteins and cholesterol. *PLoS Genet, 6*(8). doi:10.1371/journal.pgen.1001055

Kowal, J., Arras, G., Colombo, M., Jouve, M., Morath, J. P., Primdal-Bengtson, B.,... Thery, C. (2016). Proteomic comparison defines novel markers to characterize heterogeneous populations of extracellular vesicle subtypes. *Proc Natl Acad Sci U S A, 113*(8), E968-977. doi:10.1073/pnas.1521230113

Lamontagne-Proulx, J., St-Amour, I., Labib, R., Pilon, J., Denis, H. L., Cloutier, N.,... Cicchetti, F. (2019). Portrait of blood-derived extracellular vesicles in patients with Parkinson's disease. *Neurobiol Dis, 124*, 163-175. doi:10.1016/j.nbd.2018.11.002

Lane-Donovan, C., Philips, G. T., and Herz, J. (2014). More than cholesterol transporters:

lipoprotein receptors in CNS function and neurodegeneration. *Neuron, 83*(4), 771-787. doi:10.1016/j.neuron.2014.08.005

Launay, S., Maubert, E., Lebeurrier, N., Tennstaedt, A., Campioni, M., Docagne, F.,... Vivien, D. (2008). HtrA1-dependent proteolysis of TGF-β controls both neuronal maturation and developmental survival. *Cell Death Differ, 15*(9), 1408-1416. doi:10.1038/cdd.2008.82

Liddelow, S. A., Guttenplan, K. A., Clarke, L. E., Bennett, F. C., Bohlen, C. J., Schirmer, L.,... Barres, B. A. (2017). Neurotoxic reactive astrocytes are induced by activated microglia. *Nature, 541*(7638), 481-487. doi:10.1038/nature21029

Matikainen-Ankney, B. A., Kezunovic, N., Mesias, R. E., Tian, Y., Williams, F. M., Huntley, G. W., and Benson, D. L. (2016). Altered Development of Synapse Structure and Function in Striatum Caused by Parkinson's Disease-Linked LRRK2-G2019S Mutation. *J Neurosci, 36*(27), 7128-7141. doi:10.1523/JNEUROSCI.3314-15.2016

McCarthy, K. D., and de Vellis, J. (1980). Preparation of separate astroglial and oligodendroglial cell cultures from rat cerebral tissue. *J Cell Biol, 85*(3), 890-902. doi:10.1083/jcb.85.3.890

Men, Y., Yelick, J., Jin, S., Tian, Y., Chiang, M. S. R., Higashimori, H.,... Yang, Y. (2019). Exosome reporter mice reveal the involvement of exosomes in mediating neuron to astroglia communication in the CNS. *Nat Commun, 10*(1), 4136. doi:10.1038/s41467-01911534-w

Noble, J. M., Roberts, L. M., Vidavsky, N., Chiou, A. E., Fischbach, C., Paszek, M. J.,... Kourkoutis, L. F. (2020). Direct comparison of optical and electron microscopy methods for structural characterization of extracellular vesicles. *J Struct Biol, 210*(1), 107474. doi:10.1016/j.jsb.2020.107474

Ostrowski, M., Carmo, N. B., Krumeich, S., Fanget, I., Raposo, G., Savina, A.,... Thery, C. (2010). Rab27a and Rab27b control different steps of the exosome secretion pathway. *Nat Cell Biol, 12*(1), 19-30; sup pp 11-13. doi:10.1038/ncb2000

Pan, X., Chang, X., Leung, C., Zhou, Z., Cao, F., Xie, W., and Jia, Z. (2015). PAK1 regulates cortical development via promoting neuronal migration and progenitor cell proliferation. *Mol Brain, 8*, 36. doi:10.1186/s13041-015-0124-z

Pegtel, D. M., and Gould, S. J. (2019). Exosomes. *Annu Rev Biochem, 88*, 487-514. doi:10.1146/annurev-biochem-013118-111902

Pfrieger, F. W., and Ungerer, N. (2011). Cholesterol metabolism in neurons and astrocytes. *Prog Lipid Res, 50*(4), 357-371. doi:10.1016/j.plipres.2011.06.002

Reinhardt, P., Schmid, B., Burbulla, L. F., Schondorf, D. C., Wagner, L., Glatza, M.,... Sterneckert, J. (2013). Genetic correction of a LRRK2 mutation in human iPSCs links parkinsonian neurodegeneration to ERK-dependent changes in gene expression. *Cell Stem Cell, 12*(3), 354-367. doi:10.1016/j.stem.2013.01.008

Schoneberg, J., Lee, I. H., Iwasa, J. H., and Hurley, J. H. (2017). Reverse-topology membrane scission by the ESCRT proteins. *Nat Rev Mol Cell Biol, 18*(1), 5-17. doi:10.1038/nrm.2016.121

Shi, M., Liu, C., Cook, T. J., Bullock, K. M., Zhao, Y., Ginghina, C.,... Zhang, J. (2014). Plasma exosomal α-synuclein is likely CNS-derived and increased in Parkinson's disease. *Acta Neuropathol, 128*(5), 639-650. doi:10.1007/s00401-014-1314-y

Skog, J., Wurdinger, T., van Rijn, S., Meijer, D. H., Gainche, L., Sena-Esteves, M.,... Breakefield, X. O. (2008). Glioblastoma microvesicles transport RNA and proteins that promote tumour growth and provide diagnostic biomarkers. *Nat Cell Biol, 10*(12), 1470-1476.

doi:10.1038/ncb1800

Soldner, F., Hockemeyer, D., Beard, C., Gao, Q., Bell, G. W., Cook, E. G.,... Jaenisch, R. (2009). Parkinson's disease patient-derived induced pluripotent stem cells free of viral reprogramming factors. *Cell, 136*(5), 964-977. doi:10.1016/j.cell.2009.02.013

Swanwick, C. C., Shapiro, M. E., Vicini, S., and Wenthold, R. J. (2010). Flotillin-1 mediates neurite branching induced by synaptic adhesion-like molecule 4 in hippocampal neurons. *Mol Cell Neurosci, 45*(3), 213-225. doi:10.1016/j.mcn.2010.06.012

Takahashi, Y., He, H., Tang, Z., Hattori, T., Liu, Y., Young, M. M.,... Wang, H. G. (2018). An autophagy assay reveals the ESCRT-III component CHMP2A as a regulator of phagophore closure. *Nat Commun, 9*(1), 2855. doi:10.1038/s41467-018-05254-w

Tcw, J., Wang, M., Pimenova, A. A., Bowles, K. R., Hartley, B. J., Lacin, E.,... Brennand, K. J.

(2017). An Efficient Platform for Astrocyte Differentiation from Human Induced Pluripotent

Stem Cells. *Stem Cell Reports, 9*(2), 600-614. doi:10.1016/j.stemcr.2017.06.018

Temoche-Diaz, M. M., Shurtleff, M. J., Nottingham, R. M., Yao, J., Fadadu, R. P., Lambowitz, A. M., and Schekman, R. (2019). Distinct mechanisms of microRNA sorting into cancer cellderived extracellular vesicle subtypes. *ELife, 8*. doi:10.7554/*eLife*.47544

Theofilopoulos, S., Abreu de Oliveira, W. A., Yang, S., Yutuc, E., Saeed, A., Abdel-Khalik, J.,... Arenas, E. (2019). 24(S),25-Epoxycholesterol and cholesterol 24S-hydroxylase (CYP46A1) overexpression promote midbrain dopaminergic neurogenesis in vivo. *J Biol Chem, 294*(11), 4169-4176. doi:10.1074/jbc.RA118.005639

Vance, J. E. (2012). Dysregulation of cholesterol balance in the brain: contribution to neurodegenerative diseases. *Dis Model Mech, 5*(6), 746-755. doi:10.1242/dmm.010124

Wang, J., Fang, N., Xiong, J., Du, Y., Cao, Y., and Ji, W. K. (2021). An ESCRT-dependent step in fatty acid transfer from lipid droplets to mitochondria through VPS13D-TSG101 interactions. *Nat Commun, 12*(1), 1252. doi:10.1038/s41467-021-21525-5

Wang, T., Gilkes, D. M., Takano, N., Xiang, L., Luo, W., Bishop, C. J.,... Semenza, G. L. (2014). Hypoxia-inducible factors and RAB22A mediate formation of microvesicles that stimulate breast cancer invasion and metastasis. *Proc Natl Acad Sci U S A, 111*(31), E3234-3242. doi:10.1073/pnas.1410041111

Xie, W., Li, L., and Cohen, S. N. (1998). Cell cycle-dependent subcellular localization of the TSG101 protein and mitotic and nuclear abnormalities associated with TSG101 deficiency. *Proc Natl Acad Sci U S A, 95*(4), 1595-1600. doi:10.1073/pnas.95.4.1595

Yoshioka, Y., Konishi, Y., Kosaka, N., Katsuda, T., Kato, T., and Ochiya, T. (2013). Comparative marker analysis of extracellular vesicles in different human cancer types. *J Extracell Vesicles, 2*. doi:10.3402/jev.v2i0.20424

Zhang, J., and Liu, Q. (2015). Cholesterol metabolism and homeostasis in the brain. *Protein Cell, 6*(4), 254-264. doi:10.1007/s13238-014-0131-3

Zhao, Z. H., Chen, Z. T., Zhou, R. L., Zhang, X., Ye, Q. Y., and Wang, Y. Z. (2018). Increased DJ1 and α-Synuclein in Plasma Neural-Derived Exosomes as Potential Markers for Parkinson's Disease. *Front Aging Neurosci, 10*, 438. doi:10.3389/fnagi.2018.00438